# Last-Iterate Convergence of Smooth Regret Matching$^+$ Variants in Learning Nash Equilibria

**Linjian Meng**[1], **Youzhi Zhang**[2†], **Zhenxing Ge**[1], **Tianyu Ding**[3],
**Shangdong Yang**[4], **Zheng Xu**[1], **Wenbin Li**[1], **Yang Gao**[1†]

[1] National Key Laboratory for Novel Software Technology, Nanjing University
[2] Centre for Artificial Intelligence and Robotics, Hong Kong Institute of Science & Innovation, CAS
[3] Microsoft Corporation
[4] Jiangsu Key Laboratory of Big Data Security and Intelligent Processing, Nanjing
University of Posts and Telecommunications
menglinjian@smail.nju.edu.cn, youzhi.zhang@cair-cas.org.hk, zhenxingge@smail.nju.edu.cn,
tianyuding@microsoft.com, sdyang@njupt.edu.cn, xuzhengcs@smail.nju.edu.cn,
liwenbin@nju.edu.cn, gaoy@nju.edu.cn

## Abstract

Regret Matching$^+$ (RM$^+$) variants are widely used to build superhuman Poker AIs, yet few studies investigate their last-iterate convergence in learning a Nash equilibrium (NE). Although their last-iterate convergence is established for games satisfying the Minty Variational Inequality (MVI), no studies have demonstrated that these algorithms achieve such convergence in the broader class of games satisfying the weak MVI. A key challenge in proving last-iterate convergence for RM$^+$ variants in games satisfying the weak MVI is that even if the game's loss gradient satisfies the weak MVI, RM$^+$ variants operate on a transformed loss feedback which does not satisfy the weak MVI. To provide last-iterate convergence for RM$^+$ variants, we introduce a concise yet novel proof paradigm that involves: (i) transforming an RM$^+$ variant into an Online Mirror Descent (OMD) instance that updates within the original strategy space of the game to recover the weak MVI, and (ii) showing last-iterate convergence by proving the distance between accumulated regrets converges to zero via the recovered weak MVI of the feedback. Inspired by our proof paradigm, we propose Smooth Optimistic Gradient Based RM$^+$ (SOGRM$^+$) and show that it achieves last-iterate and finite-time best-iterate convergence in learning an NE of games satisfying the weak MVI, the weakest condition among all known RM$^+$ variants. Experiments show that SOGRM$^+$ significantly outperforms other algorithms. Our code is available at https://github.com/menglinjian/NeurIPS-2025-SOGRM.

## 1 Introduction

Nash Equilibrium (NE) is a fundamental concept in the field of game theory. Recent advancements in superhuman game AI, are largely attributed to NE learning [Bowling et al., 2015, Moravčík et al., 2017, Brown and Sandholm, 2018, 2019, Pérolat et al., 2022]. Despite these advancements, the most popular algorithms for learning an NE—no-regret algorithms, typically achieve only average-iterate convergence. Even in two-player zero-sum matrix games, they are prone to divergence or cyclic behavior [Bailey and Piliouras, 2018, Mertikopoulos et al., 2018b, Pérolat et al., 2021]. Average-iterate convergence requires strategy averaging, which poses significant challenges in large-scale games where function approximation is used to denote the strategy as a new function has to be trained to represent the average strategy [Liu et al., 2023].

---

[†] Corresponding authors.

39th Conference on Neural Information Processing Systems (NeurIPS 2025).

Table 1: Comparisons between the convergence results of ours and previous studies about RM$^+$ variants. "2p0s Games", "SM", "SN", and "RS" refer to two-player zero-sum matrix games, strong monotonicity, strict NE, and restarting [Cai et al., 2025], respectively. Games with the MVI cover games with strong monotonicity and two-player zero-sum matrix games. Games with the weak MVI is a super set of games with the MVI. The notations "✓" and "∘" refer to last-iterate convergence and finite-time best-iterate convergence, respectively. Note that the finite-time best-iterate convergence serves as a basis for deriving the linear last-iterate convergence rate of Restarting SExRM$^+$ and Restarting SPRM$^+$. A detailed discussion is provided in Section 2. Notably, the convergence of "SExRM$^+$ & SPRM$^+$" in Cai et al. [2025] for two-player zero-sum matrix games refers to convergence to a specific point within the set of NEs. This is a stronger notion than the convergence concept we adopt (which is also used in Cai et al. [2025] for games with the MVI), where the iterates are guaranteed to converge to the set of NEs. See details in Section 3.1.

| | Algorithm | Games with SM | 2p0s Games with SN | 2p0s Games | Games with MVI | Games with Weak MVI |
|---|---|---|---|---|---|---|
| Meng et al. [2023] | RM$^+$ | ✓ ∘ | | | | |
| Cai et al. [2025] | RM$^+$ | | ✓ ∘ | | | |
| | SExRM$^+$ & SPRM$^+$ | ✓ | ✓ ∘ | ✓ ∘ | ✓ | |
| | RS-SExRM$^+$ & RS-SPRM$^+$ | | ✓ ∘ | ✓ ∘ | | |
| **Ours** | SExRM$^+$ & SPRM$^+$ | ✓ ∘ | ✓ ∘ | ✓ ∘ | ✓ ∘ | |
| | SOGPRM$^+$ | ✓ ∘ | ✓ ∘ | ✓ ∘ | ✓ ∘ | ✓ ∘ |

To address the challenges related to averaging, numerous studies consider last-iterate convergence, ensuring iterates converge to NE [Mertikopoulos et al., 2018a, Daskalakis and Panageas, 2019, Tatarenko and Kamgarpour, 2020, Wei et al., 2021, Lee et al., 2021, Cen et al., 2021, Liu et al., 2023, Sokota et al., 2023, Abe et al., 2022a,b, 2023, Pérolat et al., 2021, 2022, Cai and Zheng, 2023]. These algorithms are based on Online Mirror Descent (OMD) or Follow the Regularized Leader (FTRL). Despite their theoretical appeal, Regret Matching$^+$ (RM$^+$) variants [Tammelin, 2014, Bowling et al., 2015, Farina et al., 2021, 2023], are more commonly utilized in solving real-world games, as evidenced by their success in superhuman Poker AIs [Bowling et al., 2015, Moravčík et al., 2017, Brown and Sandholm, 2018]. The key distinction between RM$^+$ variants and FTRL/OMD-based algorithms is that the former updates within the (subset of the) *cone* of the original strategy space of the game, whereas the latter updates within the original strategy space of the game itself.

Unfortunately, few studies investigate last-iterate convergence of RM$^+$ variants. The results on last-iterate convergence of RM$^+$ variants are restricted to games satisfying strong monotonicity [Meng et al., 2023] or a broader class of games—those satisfying the Minty Variation Inequality (MVI) [Cai et al., 2025]. Games satisfying the MVI cover several common game types, such as two-player zero-sum matrix games and convex-concave games, along with significant applications like the training of Large Language Models (LLM) [Munos et al., 2023, Wang et al., 2025]. Unfortunately, last-iterate convergence of RM$^+$ variants is not established in the broader class of games, such as those that satisfy the weak MVI [Diakonikolas, 2020, Cai and Zheng, 2022, Pethick et al., 2023, Cai et al., 2024][1]. The weak MVI is weaker and covers more games than the MVI, including applications like Generative Adversarial Networks (GAN) [Cai and Zheng, 2022]. Therefore, a key question is

*Do RM$^+$ variants achieve last-iterate convergence in*
*learning an NE of games satisfying the weak MVI?*

The main challenge in proving last-iterate convergence of RM$^+$ variants in games satisfying the weak MVI is that even if the game's loss gradient satisfies the weak MVI, RM$^+$ variants operate on a transformed feedback which does not satisfy it. Similar to Cai et al. [2025], we focus on smooth RM$^+$ variants [Farina et al., 2023], as other RM$^+$ variants (*e.g.*, RM$^+$[Tammelin, 2014], PRM$^+$[Farina et al., 2021]) diverge even in two-player zero-sum matrix games [Cai et al., 2025].

**Contributions.** (i) To prove last-iterate convergence for smooth RM$^+$ variants, we introduce a novel proof paradigm: recovering the weak MVI by transforming a smooth RM$^+$ variant into an OMD instance that updates within the original strategy space of the game[2], then showing last-iterate

---

[1]These works and ours, consider on games satisfying the weak MVI with $\rho$ (defined in Section 3) bounded below by a negative constant. No prior work has investigated games satisfying the weak MVI with any $\rho > -\infty$.

[2]This transformation is not the equivalence in Farina et al. [2021], which transforms a RM$^+$ variant into an OMD instance that updates within the cone of the original strategy space, rather than the original strategy space itself. In the OMD instance used in Farina et al. [2021], the feedback does not satisfy the weak MVI.

convergence by proving the distance between accumulated regrets converges to zero via the recovered weak MVI. Our proof paradigm is inspired by the fact: for any smooth $RM^+$ variant, its feedback does not satisfy the weak MVI; however, in this corresponding OMD instance, the feedback is the loss gradient of the original games, satisfying the weak MVI, implying the weak MVI is recovered. By using the recovered weak MVI, we show that the distance of smooth $RM^+$ variants to the set of NEs is related to the distance between accumulated regrets, and the latter distance converges to zero. (ii) To show the practical applicability of our proof paradigm, we utilize this paradigm to establish that two existing smooth $RM^+$ variants, Smooth Extra-gradient $RM^+$ (SExRM$^+$) and Smooth Predictive $RM^+$ (SPRM$^+$), achieve last-iterate in learning an NE of games satisfying the MVI. (iii) Inspired by our proof paradigm, we propose Smooth Optimistic Gradient Based $RM^+$ (SOGRM$^+$), which achieves last-iterate convergence in games satisfying the weak MVI. (iv) Experimental results show that SOGRM$^+$ significantly outperforms other algorithms. (v) Our proof paradigm yields finite-time best-iterate convergence rates for SExRM$^+$, SPRM$^+$, and SOGRM$^+$ without any modifications.

## 2 Related Work

Our work primarily focuses on the last-iterate convergence of $RM^+$ variants in learning an NE. Table 1 shows the comparison between our work and the two most relevant literature [Meng et al., 2023, Cai et al., 2025]. (i) Our proof diverges significantly from theirs as they either analyze the dynamics of limit points [Cai et al., 2025] or use the strong monotonicity [Meng et al., 2023]. (ii) The last-iterate convergence results of Meng et al. [2023] and Cai et al. [2025] cannot be extended to games satisfying the weak MVI. Specifically, the results in Meng et al. [2023] need the strong monotonicity, which is a stronger assumption than the MVI (let alone the weak MVI). In addition, the algorithms in Cai et al. [2025], such as SExRM$^+$ and SPRM$^+$, do not demonstrate last-iterate convergence in games satisfying the weak MVI. (iii) Cai et al. [2025] have to use another approach to prove the finite-time best-iterate convergence while we employ the same proof paradigm. (iv) The finite-time best-iterate convergence results of Cai et al. [2025] only hold in two-player zero-sum matrix games as their results depend on the definition of the duality gap of these games. In contrast, our finite-time best-iterate convergence results hold in games satisfying the weak MVI.

Notably, we establish the finite-time, rather than asymptotic, best convergence rate for both SExRM$^+$ and SPRM$^+$, which is very important for establishing the linear last-iterate convergence rate for smooth $RM^+$ variants. Specifically, as shown in Cai et al. [2025], if either SExRM$^+$ or SPRM$^+$ achieves the finite-time best-iterate convergence rate in a game satisfying monotonicity[3] and the metric subregularity [Wei et al., 2021], then their $RM^+$ variants employing the restarting: Restarting SExRM$^+$ or Restarting SPRM$^+$, exhibit a linear last-iterate convergence rate in that game. As mentioned above, we prove the finite-time best-iterate convergence rate of SExRM$^+$ and SPRM$^+$ in games satisfying the MVI, whereas Cai et al. [2025] establish this result only for two-player zero-sum matrix games. In other words, our results regarding the finite-time best-iterate convergence rate of SExRM$^+$ and SPRM$^+$ extend the linear last-iterate convergence rate of Restarting SExRM$^+$ and Restarting SPRM$^+$ from two-player zero-sum matrix games to games satisfying monotonicity and the metric subregularity. As we can directly apply the proof steps outlined in Cai et al. [2025] (their Appendix F.4 and G.3), we do not reproduce these details in this paper.

In this paper, we focus on smooth $RM^+$ variants. Our proof paradigm transforms a smooth $RM^+$ variant into an OMD instance that updates within the original strategy space of the game. This contrasts with prior work [Liu et al., 2021], which transforms the vanilla $RM^+$ [Tammelin, 2014] rather than a smooth $RM^+$ variant into an OMD instance that updates within the original strategy space. More importantly, our equivalence (transformation) establishes both last-iterate and finite-time best-iterate convergence for smooth $RM^+$ variants. In contrast, Liu et al. [2021] fail to demonstrate even average-iterate convergence for $RM^+$ by using their equivalence.

SOGRM$^+$ is the first $RM^+$ variant to achieve both last-iterate and finite-time best-iterate convergence in games satisfying the weak MVI. To achieve such convergences, it ensures that the weak MVI can be used after the weak MVI is recovered, which is not achievable by other smooth $RM^+$ variants. This arises as SOGRM$^+$ performs one prox-mapping operator (defined in Section 3.2) per iteration, whereas other smooth $RM^+$ variants execute two, as detailed in the second and third paragraphs of

---

[3]Monotonicity is a stronger assumption than the MVI, as the former guarantees $\forall \boldsymbol{x}^* \in \boldsymbol{\mathcal{X}}^*$ and $\boldsymbol{x} \in \boldsymbol{\mathcal{X}}$, $\langle \boldsymbol{\ell}^{\boldsymbol{x}}, \boldsymbol{x} - \boldsymbol{x}^* \rangle \geq 0$, whereas the latter ensures $\exists \boldsymbol{x}^* \in \boldsymbol{\mathcal{X}}^*$, $\forall \boldsymbol{x} \in \boldsymbol{\mathcal{X}}$, $\langle \boldsymbol{\ell}^{\boldsymbol{x}}, \boldsymbol{x} - \boldsymbol{x}^* \rangle \geq 0$ in the games considered in this paper. For more details, see Section 3.

Section 6. Notably, even with our proof paradigm, establishing these convergences for SOGRM$^+$ is significantly more challenging than for OMD-based algorithms, as discussed in Section 6. For the detailed comparison to Cai et al. [2025], see Appendix A. For the comparison between the convergence results of ours and previous studies about OMD variants in games satisfying the weak MVI, refer to Appendix B.

# 3 Preliminaries

## 3.1 Smooth Games

**Smooth games.** In this paper, we consider smooth games whose strategy space is simplex. We use $x_i \in \mathcal{X}_i$ to denote the strategy of player $i$ and $x = \{x_i | i \in \mathcal{N}\}$ to represent the strategy profile, where $\mathcal{X}_i$ is an $(|A_i| - 1)$-dimension simplex $\Delta^{|A_i|}$ and $\mathcal{N}$ is the set of players. The utility of player $i$ if all players follow strategy profile $x$ is $-\infty < u_i(x_i, x_{-i}) < +\infty$, where $-i$ are the players other than $i$. For any $i$ and the fixed $x_{-i} \in \mathcal{X}_{-i}$, $u_i(x_i, x_{-i})$ is a concave function w.r.t. $x_i \in \mathcal{X}_i$. Also, $\ell_i^x = -\nabla_{x_i} u_i(x_i, x_{-i})$ is loss gradient. In smooth games, $\|\ell^x - \ell^{x'}\|_2 \le L\|x - x'\|_2, \forall x, x' \in \mathcal{X}$, where $\ell^x = [\ell_i^x : i \in \mathcal{N}]$, and $L > 0$ is a constant. In addition, we assume $\|\ell_i^x\|_1 \le P$ for each player $i$ and strategy $x$, where $P$ is a positive constant.

**Nash equilibrium (NE).** In NE, for any player, her strategy is the best-response to the strategies of others. No player can benefit by unilaterally deviating from this equilibrium. The notation $\mathcal{X}^*$ denotes the set of NEs. As $u_i(x_i, x_{-i})$ is a concave function w.r.t. $x_i \in \mathcal{X}_i$, then $\forall x^* \in \mathcal{X}^*, x \in \mathcal{X}$, the Stampacchia variational inequality $\langle \ell_i^{x^*}, x_i^* - x_i \rangle \le 0$ holds [Facchinei, 2003].

**Last-iterate convergence.** In this paper, this convergence refers to the behavior where the strategy profiles $x^t$ converge to the set of NEs $\mathcal{X}^*$. Note that Cai et al. [2025] define a stronger concept of convergence, known as last-iterate convergence of the iterates. In contrast to last-iterate convergence discussed in our paper, last-iterate convergence of the iterates implies that $x^t$ converges to a specific point within the set of NEs. Our results and the results in Cai et al. [2025] for games satisfying the MVI do not pertain to last-iterate convergence of the iterates, whereas the results in Cai et al. [2025] regarding SExRM$^+$ and SPRM$^+$ for two-player zero-sum games do.

**Monotonicity.** Smooth games with monotonicity are called smooth monotone games, which include many common and well-studied classes of games, such as two-player zero-sum matrix games and convex-concave games [Rosen, 1965]. The most important property of smooth monotone games is monotonicity $\langle \ell^x - \ell^{x'}, x - x' \rangle \ge 0, \forall x, x' \in \mathcal{X}$. Monotonicity is the most widely used assumption in existing works about last-iterate convergence. From the definition of the NE, in smooth monotone games, $\forall x^* \in \mathcal{X}^*$ and $x \in \mathcal{X}$, $\langle \ell^x, x - x^* \rangle \ge 0$. We do not present the definition of strong monotonicity—a stronger assumption than monotonicity—as few types of games satisfy it.

**Minty variation inequality (MVI).** It is a weaker assumption than the monotonicity. The MVI implies that $\exists x^* \in \mathcal{X}^*$, $\forall x \in \mathcal{X}$, $\langle \ell^x, x - x^* \rangle \ge 0$. Note that the MVI uses $\exists$ rather than $\forall$. We provide an example of games that satisfy the weak MVI in Appendix J.

**Weak MVI.** Some recent works consider a weaker assumption than the MVI called the weak MVI [Diakonikolas, 2020, Cai and Zheng, 2022, Pethick et al., 2023, Cai et al., 2024], which covers more game types. Formally, the weak MVI with $\rho \le 0$ implies there exists $x^* \in \mathcal{X}^*$ that ensures

$$\langle \ell^x + z, x - x^* \rangle \ge \rho\|\ell^x + z\|_2^2, \forall z \in \mathcal{N}_{\mathcal{X}}(x), x \in \mathcal{X},$$

where $\mathcal{N}_{\mathcal{X}}(x) = \{v \in \mathbb{R}^{|\mathcal{X}|} : \langle v, x' - x \rangle \le 0, \forall x' \in \mathcal{X}\}$ is the normal cone of $x$. If $\rho \to -\infty$, intuitively, any smooth games satisfy the weak MVI. In Appendix J, we provide a smooth game that satisfies the weak MVI with $0 > \rho > -\infty$ and does not satisfy the MVI. The relations between monotonicity, the MVI, and the weak MVI, is that monotonicity $\subseteq$ MVI $\subseteq$ weak MVI.

**Tangent residual.** To measure the distance from a strategy profile to the set of NEs, we employ the tangent residual provided by Cai et al. [2022]. Formally, $\forall x \in \mathcal{X}$, its tangent residual is

$$r^{tan}(x) = \min_{z \in \mathcal{N}_{\mathcal{X}}(x)} \|\ell^x + z\|_2.$$

If $r^{tan}(x) = 0$, then $x$ is an NE in smooth games. Also, if $x$ is an NE in smooth games, $r^{tan}(x) = 0$. Therefore, $\lim_{t \to \infty} r^{tan}(x^t) = 0$ implies that $x^t$ converges to the set of NEs. As shown in Cai et al. [2022], the tangent residual is an upper bound for the duality gap, a conventional metric that quantifies the distance between a strategy profile and the set of NEs (details are in Appendix L).

## 3.2 Regret Matching$^+$ and Smooth Regret Matching$^+$ Variants

**Online convex optimization.** Each player $i$ selects a decision $\boldsymbol{x}_i^t$ via the feedback in this framework. Such feedback is the loss gradient $\boldsymbol{\ell}_i^{t-1} = \boldsymbol{\ell}_i^{\boldsymbol{x}^{t-1}}$ in solving smooth games. No-regret algorithms are the algorithms that ensure the regret $R_i^{\mathrm{T}} = \max_{\boldsymbol{x}_i \in \boldsymbol{\mathcal{X}}_i} \sum_{t=1}^{T} \langle \boldsymbol{\ell}_i^t, \boldsymbol{x}_i^t - \boldsymbol{x}_i \rangle$ to grow sublinearly, where $\boldsymbol{x}_i^t$ is the decision at iteration $t$.

**Online mirror descent (OMD).** OMD is a traditional no-regret algorithm [Nemirovskij and Yudin, 1983]. Let $q_i^t(\cdot) : \boldsymbol{\mathcal{X}}_i \to \mathbb{R}, \forall t \geq 1$, OMD generates the decisions via the prox-mapping operator

$$\boldsymbol{x}_i^{t+1} \in \underset{\boldsymbol{x}_i \in \boldsymbol{\mathcal{X}}_i}{\arg\min}\{\langle \boldsymbol{\ell}_i^t, \boldsymbol{x}_i \rangle + q_i^t(\boldsymbol{x}_i) + D_{q_i^{0:t-1}}(\boldsymbol{x}_i, \boldsymbol{x}_i^t)\}, \tag{1}$$

where $q_i^{0:t-1}(\cdot) = q_i^0(\cdot) + q_i^1(\cdot) + \cdots + q_i^{t-1}(\cdot)$, and $D_{q_i^{0:t-1}}(\boldsymbol{x}, \boldsymbol{y}) = q_i^{0:t-1}(\boldsymbol{x}) - q_i^{0:t-1}(\boldsymbol{y}) - \langle \nabla q_i^{0:t-1}(\boldsymbol{y}), \boldsymbol{x} - \boldsymbol{y} \rangle$ is the Bregman divergence associated with $q_i^{0:t-1}(\cdot)$. We employ the definition of OMD in Joulani et al. [2017] and Liu et al. [2021], which is a generalization of the standard OMD, to demonstrate the equivalence between smooth RM$^+$ variants and OMD that updates within the original strategy space of the game. To recover the standard OMD, we can set $q_i^0 = \phi(\cdot)/\eta$ and $q_i^t = 0$ for all $t \geq 1$, where $\phi(\cdot)$ is a 1-strongly convex regularizer w.r.t. some norm in the decision space $\boldsymbol{\mathcal{X}}$, and $\eta > 0$. The use of this specific OMD form in Eq. (1) is necessary as a RM$^+$ variant can be transformed into an OMD instance that updates within the original strategy space of the game only when using the OMD of this form.

**Blackwell approachability framework.** RM$^+$ variants are from this framework whose core insight lies in reframing the problem of regret minimization within the strategy space $\boldsymbol{\mathcal{Z}}$ as regret minimization within $\mathrm{cone}(\boldsymbol{\mathcal{Z}}) = \{\lambda \boldsymbol{z} \mid \boldsymbol{z} \in \boldsymbol{\mathcal{Z}}, \lambda \geq 0\}$ [Blackwell, 1956, Abernethy et al., 2011, Farina et al., 2021]. Specifically, a regret minimization algorithm is instantiated in $\mathrm{cone}(\boldsymbol{\mathcal{Z}})$, where its output at iteration $t$ is $\boldsymbol{\theta}^t$. This corresponds to the strategy $\boldsymbol{z}^t = \boldsymbol{\theta}^t / \langle \boldsymbol{\theta}^t, \mathbf{1} \rangle$ within $\boldsymbol{\mathcal{Z}}$. Given the loss $\boldsymbol{\ell}^t$ at iteration $t$, the algorithm observes the transformed loss $-\boldsymbol{F}(\boldsymbol{\theta}^t) = -\langle \boldsymbol{\ell}^t, \boldsymbol{z}^t \rangle \mathbf{1} + \boldsymbol{\ell}^t$ and subsequently generates $\boldsymbol{\theta}^{t+1}$. The pseudocode of all RM$^+$ variants presented in this paper, *e.g.*, SExRM$^+$, SPRM$^+$, and SOGRM$^+$, is in Appendix K.

**Smooth RM$^+$ variants** [Farina et al., 2023]. These variants are designed to address the instability of PRM$^+$ [Farina et al., 2021], *e.g.*, rapid fluctuations of the strategy $\boldsymbol{x}_i$ across iterations. To do that, Farina et al. [2023] enable the decision $\boldsymbol{\theta}_i^t$ in $\mathbb{R}_{\geq 1}^d$ instead of $\mathbb{R}_{\geq 0}^d$ in other RM$^+$ variants to obtain the smoothness of the strategy $\boldsymbol{x}_i$ w.r.t. $\boldsymbol{\theta}_i$ (Lemma 5.4), where $\mathbb{R}_{\geq 1}^d = \{\boldsymbol{y} \mid \boldsymbol{y} \in \mathbb{R}^d, \boldsymbol{y} \geq \mathbf{0}, \|\boldsymbol{y}\|_1 \geq 1\}$. We consider two existing smooth RM$^+$ variants, Smooth Extra-gradient RM$^+$ (SExRM$^+$) and Smooth Predictive RM$^+$ (SPRM$^+$). They are algorithm grounded in Blackwell approachability framework. Specifically, they are respectively related to instances of two OMD variants, Optimistic Gradient Descent Ascent (OGDA) [Popov, 1980] and Extra-Gradient (EG) [Korpelevich, 1976], which updates within $\mathbb{R}_{\geq 1}^d$, the subset of $\mathrm{cone}(\boldsymbol{\mathcal{X}}_i)$. The update rule of SExRM$^+$ is

$$
\begin{aligned}
\boldsymbol{\theta}_i^{t+\frac{1}{2}} &\in \underset{\boldsymbol{\theta}_i \in \mathbb{R}_{\geq 1}^{|A_i|}}{\arg\min}\{\langle -\boldsymbol{F}_i(\boldsymbol{\theta}^t), \boldsymbol{\theta}_i \rangle + \frac{1}{\eta} D_\psi(\boldsymbol{\theta}_i, \boldsymbol{\theta}_i^t)\}, \ \boldsymbol{x}_i^{t+\frac{1}{2}} = \frac{\boldsymbol{\theta}_i^{t+\frac{1}{2}}}{\|\boldsymbol{\theta}_i^{t+\frac{1}{2}}\|_1}, \\
\boldsymbol{\theta}_i^{t+1} &\in \underset{\boldsymbol{\theta}_i \in \mathbb{R}_{\geq 1}^{|A_i|}}{\arg\min}\{\langle -\boldsymbol{F}_i(\boldsymbol{\theta}^{t+\frac{1}{2}}), \boldsymbol{\theta}_i \rangle + \frac{1}{\eta} D_\psi(\boldsymbol{\theta}_i, \boldsymbol{\theta}_i^t)\}, \ \boldsymbol{x}_i^{t+1} = \frac{\boldsymbol{\theta}_i^{t+1}}{\|\boldsymbol{\theta}_i^{t+1}\|_1},
\end{aligned} \tag{2}
$$

and the update rule of SPRM$^+$ is

$$
\begin{aligned}
\boldsymbol{\theta}_i^{t+\frac{1}{2}} &\in \underset{\boldsymbol{\theta}_i \in \mathbb{R}_{\geq 1}^{|A_i|}}{\arg\min}\{\langle -\boldsymbol{F}_i(\boldsymbol{\theta}^{t-\frac{1}{2}}), \boldsymbol{\theta}_i \rangle + \frac{1}{\eta} D_\psi(\boldsymbol{\theta}_i, \boldsymbol{\theta}_i^t)\}, \ \boldsymbol{x}_i^{t+\frac{1}{2}} = \frac{\boldsymbol{\theta}_i^{t+\frac{1}{2}}}{\|\boldsymbol{\theta}_i^{t+\frac{1}{2}}\|_1}, \\
\boldsymbol{\theta}_i^{t+1} &\in \underset{\boldsymbol{\theta}_i \in \mathbb{R}_{\geq 1}^{|A_i|}}{\arg\min}\{\langle -\boldsymbol{F}_i(\boldsymbol{\theta}^{t+\frac{1}{2}}), \boldsymbol{\theta}_i \rangle + \frac{1}{\eta} D_\psi(\boldsymbol{\theta}_i, \boldsymbol{\theta}_i^t)\}, \ \boldsymbol{x}_i^{t+1} = \frac{\boldsymbol{\theta}_i^{t+1}}{\|\boldsymbol{\theta}_i^{t+1}\|_1},
\end{aligned} \tag{3}
$$

where $\eta > 0$ and $\psi(\cdot)$ is the quadratic regularizer. Unlike RM$^+$, smooth RM$^+$ variants do not enjoy step size invariance [Chakrabarti et al., 2024]. However, empirical evidence demonstrates that RM$^+$ does not achieve last-iterate convergence in two-player zero-sum matrix games [Cai et al., 2025].

# 4 Our Proof Paradigm

We now introduce our proof paradigm. We only consider smooth $RM^+$ variants, since other $RM^+$ variants, *e.g.*, $RM^+$ and $PRM^+$, are experimentally shown to diverge even in two-player zero-sum matrix games [Cai et al., 2025]. Our proof paradigm is inspired by this fact: for any smooth $RM^+$ variant, its feedback does not satisfy the weak MVI; however, in its corresponding OMD instance that updates within the original strategy space of the games, the feedback is the loss gradient of the original games, satisfying the weak MVI. Specifically, our proof paradigm consists of two steps: (i) recovering the weak MVI by transforming a smooth $RM^+$ variant into an OMD instance that updates within the original strategy space of the games, and (ii) showing the last-iterate convergence by proving the distance between accumulated regrets converges to zero via the recovered weak MVI.

**Phase 1.** To recover the weak MVI of a smooth $RM^+$ variant, we transform this smooth $RM^+$ variant into an OMD instance that updates within the original strategy space of the game. To do that, it is sufficient to show the equivalence between the update rule defined in Eq. (4) and (5):

$$
\begin{cases}
\boldsymbol{\theta}_i^{t_2} \in \underset{\boldsymbol{\theta}_i \in \mathbb{R}_{\geq 1}^{|A_i|}}{\arg\min} \{\langle -\boldsymbol{F}_i(\boldsymbol{\theta}^{t_1}), \boldsymbol{\theta}_i \rangle + \frac{1}{\eta} D_\psi(\boldsymbol{\theta}_i, \boldsymbol{\theta}_i^{t_0})\}, \\
\boldsymbol{x}_i^{t_2} = \dfrac{\boldsymbol{\theta}_i^{t_2}}{\|\boldsymbol{\theta}_i^{t_2}\|_1}, \quad \boldsymbol{F}_i(\boldsymbol{\theta}^{t_1}) = \langle \dfrac{\boldsymbol{\theta}_i^{t_1}}{\|\boldsymbol{\theta}_i^{t_1}\|_1}, \boldsymbol{\ell}_i^{t_1}\rangle \mathbf{1} - \boldsymbol{\ell}_i^{t_1},
\end{cases}
\tag{4}
$$

$$
\begin{cases}
\boldsymbol{x}_i^{t_2} \in \underset{\boldsymbol{x}_i \in \boldsymbol{\mathcal{X}}_i}{\arg\min} \{\langle \boldsymbol{\ell}_i^{t_1}, \boldsymbol{x}_i \rangle + f_i(\boldsymbol{x}_i) + D_{h_i}(\boldsymbol{x}_i, \boldsymbol{x}_i^{t_0})\}, \\
h_i(\boldsymbol{x}_i) + f_i(\boldsymbol{x}_i) = \dfrac{\|\boldsymbol{\theta}_i^{t_2}\|_1}{\eta} \psi(\boldsymbol{x}_i), \quad h_i(\boldsymbol{x}_i) = \dfrac{\|\boldsymbol{\theta}_i^{t_0}\|_1}{\eta} \psi(\boldsymbol{x}_i),
\end{cases}
\tag{5}
$$

where $t_0$, $t_1$, $t_2$ refer to different iterations, $\eta > 0$, $\boldsymbol{x}_i^{t_0} = \boldsymbol{\theta}_i^{t_0}/\|\boldsymbol{\theta}_i^{t_0}\|_1$, $\boldsymbol{\theta}_i^{t_0}$ with $\boldsymbol{\theta}_i^{t_1} \in \mathbb{R}_{\geq 1}^{|A_i|}$, $\boldsymbol{\ell}_i^{t_1}$ is the loss gradient of player $i$ induced by $\boldsymbol{x}^{t_1} = [\boldsymbol{x}_i^{t_1} = \boldsymbol{\theta}_i^{t_1}/\|\boldsymbol{\theta}_i^{t_1}\|_1 : i \in \mathcal{N}]$, and $\psi(\cdot)$ is the quadratic regularizer.

The intuitive explanation of this equivalence between the update rule defined in Eq. (4) and (5) is that both projecting onto the truncated positive orthant and projecting onto the simplex can be written in a unified way using the max operator, *e.g.*, both update rules in Eq. (4) and Eq. (5) can be expressed as $[\cdot]^+ = \max(\cdot, \mathbf{0})$. Due to page limits, the detailed proof is in Appendix C.

This equivalence is the inherent property of smooth $RM^+$ variants and does not involve the game types. Notably, this equivalence is not the equivalence in Farina et al. [2021] as the OMD in Farina et al. [2021] updates within $\text{cone}(\boldsymbol{\mathcal{X}}_i)$ rather than $\boldsymbol{\mathcal{X}}_i$, the original strategy space of the game. The weak MVI is recovered as the feedback in Eq. (5) is the loss gradient of the original game that satisfies the weak MVI. Formally, for the feedback in Eq. (5), there exists an $\boldsymbol{x}^* \in \boldsymbol{\mathcal{X}}^*$ ensures that $\langle \boldsymbol{\ell}^{t_1} + \boldsymbol{z}, \boldsymbol{x}^{t_1} - \boldsymbol{x}^* \rangle \geq \rho \|\boldsymbol{\ell}^{t_1} + \boldsymbol{z}\|_2^2, \forall \boldsymbol{z} \in \mathcal{N}_{\boldsymbol{\mathcal{X}}}(\boldsymbol{x}^{t_1})$ with $\boldsymbol{\ell}^{t_1} = [\boldsymbol{\ell}_i^{t_1} | i \in \mathcal{N}]$, as $\boldsymbol{\ell}_i^{t_1}$ is the loss gradient of the original game that satisfies the weak MVI.

**Phase 2.** The first step in this phase is: establishing a relationship between the distance of the strategy profile induced by $RM^+$ variants to the set of NEs and the distance between accumulated regrets. Formally, from the first-order optimality of the prox-mapping operator in Eq. (5), $\forall \boldsymbol{x}_i \in \mathcal{X}_i$ we have

$$
\langle \boldsymbol{\ell}_i^{t_1} + \nabla_{\boldsymbol{x}_i^{t_2}} f_i(\boldsymbol{x}_i^{t_2}) + \nabla_{\boldsymbol{x}_i^{t_2}} D_{h_i}(\boldsymbol{x}_i^{t_2}, \boldsymbol{x}_i^{t_0}), \boldsymbol{x}_i - \boldsymbol{x}_i^{t_2} \rangle \geq 0
$$

$$
\Rightarrow \sum_{i \in \mathcal{N}} \langle \boldsymbol{\ell}_i^{t_1} + \nabla_{\boldsymbol{x}_i^{t_2}} h_i(\boldsymbol{x}_i^{t_2}) + \nabla_{\boldsymbol{x}_i^{t_2}} f_i(\boldsymbol{x}_i^{t_2}) - \nabla_{\boldsymbol{x}_i^{t_0}} h_i(\boldsymbol{x}_i^{t_0}), \boldsymbol{x}_i - \boldsymbol{x}_i^{t_2} \rangle \geq 0
$$

$$
\Rightarrow \sum_{i \in \mathcal{N}} \langle \boldsymbol{\ell}_i^{t_1} - \frac{\boldsymbol{\theta}_i^{t_0} - \boldsymbol{\theta}_i^{t_2}}{\eta}, \boldsymbol{x}_i - \boldsymbol{x}_i^{t_2} \rangle \geq 0 \Leftrightarrow -\boldsymbol{\ell}^{t_1} + \frac{\boldsymbol{\theta}^{t_0} - \boldsymbol{\theta}^{t_2}}{\eta} \in \mathcal{N}_{\boldsymbol{\mathcal{X}}}(\boldsymbol{x}^{t_2})
\tag{6}
$$

$$
\Rightarrow r^{tan}(\boldsymbol{x}^{t_2}) \leq \|\boldsymbol{\ell}^{t_2} - \boldsymbol{\ell}^{t_1} + \frac{\boldsymbol{\theta}^{t_0} - \boldsymbol{\theta}^{t_2}}{\eta}\|_2,
$$

where $\boldsymbol{x}_i^{t_2} = \boldsymbol{\theta}_i^{t_2}/\|\boldsymbol{\theta}_i^{t_2}\|_1$, the third line is from $\nabla_{\boldsymbol{x}_i^{t_2}} h_i(\boldsymbol{x}_i^{t_2}) + \nabla_{\boldsymbol{x}_i^{t_2}} f_i(\boldsymbol{x}_i^{t_2}) = \boldsymbol{\theta}_i^{t_2}/\eta$ and $\nabla_{\boldsymbol{x}_i^{t_0}} h_i(\boldsymbol{x}_i^{t_0}) = \boldsymbol{\theta}_i^{t_0}/\eta$, as well as the last line is from the definition of the tangent residual. Thus, if we prove $\|\boldsymbol{\ell}^{t_2} - \boldsymbol{\ell}^{t_1}\|_2 \to 0$ and $\|\boldsymbol{\theta}^{t_2} - \boldsymbol{\theta}^{t_0}\|_2 \to 0$, we can get that $r^{tan}(\boldsymbol{x}^{t_2}) \to 0$, implying $\boldsymbol{x}^{t_2}$ is an NE. For smooth $RM^+$ variants, we have that $\|\boldsymbol{\ell}^{t_2} - \boldsymbol{\ell}^{t_1}\|_2 \leq O(\|\boldsymbol{\theta}^{t_2} - \boldsymbol{\theta}^{t_1}\|_2)$,

which does not hold for other RM$^+$ variants, where $\ell_i^{t_2}$ is the loss gradient of player $i$ induced by $\boldsymbol{x}^{t_2} = [\boldsymbol{x}_i^{t_2} = \boldsymbol{\theta}_i^{t_2}/\|\boldsymbol{\theta}_i^{t_2}\|_1 : i \in \mathcal{N}]$. For smooth games satisfying the MVI, we show that for existing smooth RM$^+$ variants, such as SExRM$^+$ and SPRM$^+$, $\|\boldsymbol{\theta}^{t_2} - \boldsymbol{\theta}^{t_1}\|_2 \to 0$ and $\|\boldsymbol{\theta}^{t_1} - \boldsymbol{\theta}^{t_0}\|_2 \to 0$ (indicating $\|\boldsymbol{\theta}^{t_2} - \boldsymbol{\theta}^{t_0}\|_2 \to 0$) as $t_0 \to \infty$, implying that $\boldsymbol{x}^{t_2}$ converges to the set of NEs, where $t_1 = t_0 + 1/2$ and $t_2 = t_1 + 1/2$ (detailed in Eq. (22), (23), and (24)). For smooth games satisfying the weak MVI, we apply the recovered weak MVI to SOGRM$^+$ to get $\|\boldsymbol{\theta}^{t_0} - \boldsymbol{\theta}^{t_0+1}\|_2 \to 0$ as $t_0 \to \infty$, proving that $\boldsymbol{x}^{t_2}$ converges to the set of NEs, where $t_2 = t_0 + 1/2$ (detailed in Eq. (11)). Note that $t_0$ can be any iteration $t$, while these convergence results hold only as $t_0 \to \infty$.

# 5 Application of Our Proof Paradigm: Convergence of SExRM$^+$ and SPRM$^+$

To demonstrate the practical applicability of our paradigm, we employ it to establish both last-iterate and finite-time best-iterate convergence of two existing smooth RM$^+$ variants, namely SExRM$^+$ and SPRM$^+$. It is important to note that while Cai et al. [2025] provide last-iterate convergence guarantees for SExRM$^+$ and SPRM$^+$ in learning an NE of games satisfying the MVI, they do not address the finite-time best-iterate convergence.

**Theorem 5.1.** *SExRM$^+$ with $0 < \eta < \frac{1}{DL_u}$ or SPRM$^+$ with $0 < \eta < \frac{1}{8DL_u}$ achieves the asymptotic last-iterate convergence and $O(\frac{1}{\sqrt{t}})$ best-iterate convergence rate in learning an NE of games satisfying the MVI, where $D = \max_{i\in\mathcal{N}} |A_i|$ and $L_u = \sqrt{2P^2 + 4L^2}$. Specifically, $r^{tan}(\boldsymbol{x}^{t+\frac{1}{2}}) \to 0$ and $\min_{\tau\in[t]} r^{tan}(\boldsymbol{x}^{\tau+\frac{1}{2}}) \le O(\frac{1}{\sqrt{t}})$ as $t \to \infty$.*

To prove Theorem 5.1, we introduce the Theorem 5.2, Theorem 5.3, and Lemma 5.4 (the proof of Theorems 5.2 and 5.3 are in Appendix E and F, respectively).

**Theorem 5.2.** *SExRM$^+$ with $0 < \eta < \frac{1}{DL_u}$ ensures $\|\boldsymbol{\theta}^{t+\frac{1}{2}} - \boldsymbol{\theta}^t\|_2 \to 0$ and $\|\boldsymbol{\theta}^{t+1} - \boldsymbol{\theta}^{t+\frac{1}{2}}\|_2 \to 0$ as $t \to \infty$, and $\min_{\tau\in[t]} \left( \|\boldsymbol{\theta}^{\tau+\frac{1}{2}} - \boldsymbol{\theta}^\tau\|_2^2 + \|\boldsymbol{\theta}^{\tau+1} - \boldsymbol{\theta}^{\tau+\frac{1}{2}}\|_2^2 \right) \le O(\frac{1}{t}), \forall t \ge 1$.*

**Theorem 5.3.** *SPRM$^+$ with $0 < \eta < \frac{1}{8DL_u}$ ensures $\|\boldsymbol{\theta}^{t+\frac{1}{2}} - \boldsymbol{\theta}^t\|_2 \to 0$ and $\|\boldsymbol{\theta}^{t+1} - \boldsymbol{\theta}^{t+\frac{1}{2}}\|_2 \to 0$ as $t \to \infty$, and $\min_{\tau\in[t]} \left( \|\boldsymbol{\theta}^{\tau+\frac{1}{2}} - \boldsymbol{\theta}^\tau\|_2^2 + \|\boldsymbol{\theta}^{\tau+1} - \boldsymbol{\theta}^{\tau+\frac{1}{2}}\|_2^2 \right) \le O(\frac{1}{t}), \forall t \ge 1$.*

**Lemma 5.4.** *(Proposition 1 in Farina et al. [2023]) $\forall \boldsymbol{a}, \boldsymbol{b} \in \mathbb{R}_{\ge 1}^d, \|\frac{\boldsymbol{a}}{\|\boldsymbol{a}\|_1} - \frac{\boldsymbol{b}}{\|\boldsymbol{b}\|_1}\|_2 \le \sqrt{d}\|\boldsymbol{a} - \boldsymbol{b}\|_2$.*

**Proof Sketch of Theorem 5.1.** From the analysis in Section 4, for SExRM$^+$ and SPRM$^+$, the tangent residual of the strategy profile $\boldsymbol{x}^{t+1}$ is

$$r^{tan}(\boldsymbol{x}^{t+1}) \le L\|\boldsymbol{x}^{t+1} - \boldsymbol{x}^{t+\frac{1}{2}}\|_2 + \frac{1}{\eta}\|\boldsymbol{\theta}^t - \boldsymbol{\theta}^{t+\frac{1}{2}}\|_2 + \frac{1}{\eta}\|\boldsymbol{\theta}^{t+\frac{1}{2}} - \boldsymbol{\theta}^{t+1}\|_2,$$

Then, using Lemma 5.4 with $\boldsymbol{a} = \boldsymbol{\theta}^{t+1}$ and $\boldsymbol{b} = \boldsymbol{\theta}^{t+\frac{1}{2}}$, we have

$$r^{tan}(\boldsymbol{x}^{t+1}) \le L\sqrt{D}\|\boldsymbol{\theta}^{t+1} - \boldsymbol{\theta}^{t+\frac{1}{2}}\|_2 + \frac{1}{\eta}\|\boldsymbol{\theta}^t - \boldsymbol{\theta}^{t+\frac{1}{2}}\|_2 + \frac{1}{\eta}\|\boldsymbol{\theta}^{t+\frac{1}{2}} - \boldsymbol{\theta}^{t+1}\|_2.$$

From Theorem 5.2 and 5.3 ($\|\boldsymbol{\theta}^t - \boldsymbol{\theta}^{t+\frac{1}{2}}\|_2 \to 0$ and $\|\boldsymbol{\theta}^{t+\frac{1}{2}} - \boldsymbol{\theta}^{t+1}\|_2 \to 0$), we get $r^{tan}(\boldsymbol{x}^t) \to 0$ as $t \to \infty$. Similarly, we get

$$(r^{tan}(\boldsymbol{x}^{t+1}))^2 \le 2L^2\|\boldsymbol{x}^{t+1} - \boldsymbol{x}^{t+\frac{1}{2}}\|_2 + \frac{2}{\eta^2}\|\boldsymbol{\theta}^t - \boldsymbol{\theta}^{t+1}\|_2^2.$$

By using Lemma 5.4, we get

$$(r^{tan}(\boldsymbol{x}^{t+1}))^2 \le \left( 2L^2D^2 + \frac{4}{\eta^2} \right) \left( \|\boldsymbol{\theta}^{t+1} - \boldsymbol{\theta}^{t+\frac{1}{2}}\|_2^2 + \|\boldsymbol{\theta}^t - \boldsymbol{\theta}^{t+\frac{1}{2}}\|_2^2 \right).$$

Thus, from Theorem 5.2 and 5.3 ($\min_{\tau\in[t]}(\|\boldsymbol{\theta}^{\tau+\frac{1}{2}} - \boldsymbol{\theta}^\tau\|_2^2 + \|\boldsymbol{\theta}^{\tau+1} - \boldsymbol{\theta}^{\tau+\frac{1}{2}}\|_2^2) \le O(\frac{1}{t})$), we get that for $\tau = \arg\min_{\tau\in[t]}(\|\boldsymbol{\theta}^{\tau+\frac{1}{2}} - \boldsymbol{\theta}^\tau\|_2^2 + \|\boldsymbol{\theta}^{\tau+1} - \boldsymbol{\theta}^{\tau+\frac{1}{2}}\|_2^2)$,

$$r^{tan}(\boldsymbol{x}^{\tau+1}) \le \sqrt{O\left( \left( \|\boldsymbol{\theta}^{\tau+\frac{1}{2}} - \boldsymbol{\theta}^\tau\|_2^2 + \|\boldsymbol{\theta}^{\tau+1} - \boldsymbol{\theta}^{\tau+\frac{1}{2}}\|_2^2 \right) \right)} \le O(\frac{1}{\sqrt{t}}).$$

For more details, see Appendix D.

# 6  Our Algorithm: SOGRM$^+$

We prove that SExRM$^+$ and SPRM$^+$ achieve last-iterate convergence and finite-time best-iterate convergence in games satisfying the MVI. However, they do not achieve such convergences in games satisfying the weak MVI, covering games satisfying the MVI. Inspired by our paradigm, we propose a new smooth RM$^+$ variant called Smooth Optimistic Gradient Based Regret Matching$^+$ (SOGRM$^+$), which achieves last-iterate and finite-time best-iterate convergence in games satisfying the weak MVI.

The key insight of SOGRM$^+$ is to transform the term related to $\boldsymbol{x}^*$ (an NE) to a term that only depends on the gap between accumulated regrets across different iterations via the recovered weak MVI, which is pivotal for establishing last-iterate convergence. Specifically, it transforms the penultimate line of Eq. (6) from $-\boldsymbol{\ell}^{t_1} + (\boldsymbol{\theta}^{t_0} - \boldsymbol{\theta}^{t_2})/\eta \in \mathcal{N}_{\boldsymbol{\mathcal{X}}}(\boldsymbol{x}^{t_2})$ to $-\boldsymbol{\ell}^{t_2} + (\boldsymbol{\theta}^{t_0} - \boldsymbol{\theta}^{t_0+1})/\eta \in \mathcal{N}_{\boldsymbol{\mathcal{X}}}(\boldsymbol{x}^{t_2})$ by using the second line of Eq. (7) (e.g., $\boldsymbol{\theta}_i^{t+1} = \boldsymbol{\theta}_i^{t+\frac{1}{2}} - \eta \boldsymbol{F}_i(\boldsymbol{\theta}^{t-\frac{1}{2}}) + \eta \boldsymbol{F}_i(\boldsymbol{\theta}^{t+\frac{1}{2}})$), enabling the derivation of $-\langle \boldsymbol{\ell}^{t_2} - \boldsymbol{\ell}^{t_2} + (\boldsymbol{\theta}^{t_0} - \boldsymbol{\theta}^{t_0+1})/\eta, \boldsymbol{x}^{t_2} - \boldsymbol{x}^* \rangle \le -\rho \|(\boldsymbol{\theta}^{t_0} - \boldsymbol{\theta}^{t_0+1})/\eta\|_2^2$ via the weak MVI, where $t_0$ can be any iteration $t$, $t_1 = t_0 - 1/2$, and $t_2 = t_0 + 1/2$. See details in Eq. (9), (10), (41) and (42).

In contrast, for SExRM$^+$ and SPRM$^+$, the second line of the update rule in SOGRM$^+$, $\boldsymbol{\theta}_i^{t+1} = \boldsymbol{\theta}_i^{t+\frac{1}{2}} - \eta \boldsymbol{F}_i(\boldsymbol{\theta}^{t-\frac{1}{2}}) + \eta \boldsymbol{F}_i(\boldsymbol{\theta}^{t+\frac{1}{2}})$, corresponds to a prox-mapping operator (first mentioned in the text around Eq. (1)): $\boldsymbol{\theta}_i^{t+1} \in \arg\min_{\boldsymbol{\theta}_i \in \mathbb{R}_{\ge 1}^{|A_i|}} \{\langle -\boldsymbol{F}_i(\boldsymbol{\theta}^{t+\frac{1}{2}}), \boldsymbol{\theta}_i \rangle + D_\psi(\boldsymbol{\theta}_i, \boldsymbol{\theta}_i^t)/\eta\}$. Unfortunately, this update does not yield $-\boldsymbol{\ell}^{t_2} + (\boldsymbol{\theta}^{t_0} - \boldsymbol{\theta}^{t_0+1})/\eta \in \mathcal{N}_{\boldsymbol{\mathcal{X}}}(\boldsymbol{x}^{t_2})$, enabling the weak MVI cannot be used.

Notably, the proof of SOGRM$^+$ needs additional techniques compared to that of OMD-based algorithms, i.e., transforming variables using the definition of the inner product to employ the weak MVI and tangent residual (details are in Eq. (9), (10), (38), (39), (41), and (46)) rather than directly transforming variables using equalities as in OMD-based algorithms. The update rule of SOGRM$^+$ at iteration $t$ is

$$\boldsymbol{\theta}_i^{t+\frac{1}{2}} \in \arg\min_{\boldsymbol{\theta}_i \in \mathbb{R}_{\ge 1}^{|A_i|}} \{\langle -\boldsymbol{F}_i(\boldsymbol{\theta}^{t-\frac{1}{2}}), \boldsymbol{\theta}_i \rangle + \frac{1}{\eta} D_\psi(\boldsymbol{\theta}_i, \boldsymbol{\theta}_i^t)\}, \; \boldsymbol{x}_i^{t+\frac{1}{2}} = \frac{\boldsymbol{\theta}_i^{t+\frac{1}{2}}}{\|\boldsymbol{\theta}_i^{t+\frac{1}{2}}\|_1}, \qquad (7)$$
$$\boldsymbol{\theta}_i^{t+1} = \boldsymbol{\theta}_i^{t+\frac{1}{2}} - \eta \boldsymbol{F}_i(\boldsymbol{\theta}^{t-\frac{1}{2}}) + \eta \boldsymbol{F}_i(\boldsymbol{\theta}^{t+\frac{1}{2}}).$$

The intuition for the update step in Eq. (7) is: SOGRM$^+$, can be viewed as a momentum-based method, conceptually similar to Adam. Specifically, the update in Eq. (7) is equivalent to $\boldsymbol{\theta}_i^{t+\frac{3}{2}} \in \arg\min_{\boldsymbol{\theta}_i \in \mathbb{R}_{\ge 1}^{|A_i|}} \{\langle -\boldsymbol{F}_i(\boldsymbol{\theta}^{t+\frac{1}{2}}) - (\boldsymbol{F}_i(\boldsymbol{\theta}^{t+\frac{1}{2}}) - \boldsymbol{F}_i(\boldsymbol{\theta}^{t-\frac{1}{2}})), \boldsymbol{\theta}_i \rangle + \frac{1}{\eta} D_\psi(\boldsymbol{\theta}_i, \boldsymbol{\theta}_i^{t+\frac{1}{2}})\}$. In this context, $\boldsymbol{F}_i(\boldsymbol{\theta}^{t+\frac{1}{2}}) - \boldsymbol{F}_i(\boldsymbol{\theta}^{t-\frac{1}{2}})$ acts as a momentum term. More precisely, it represents the trend of change in $\boldsymbol{F}_i$ over the two most recent iterations, analogous to the "acceleration" of $\boldsymbol{F}_i$. This can be likened to a ball rolling down a hill; as acceleration accumulates, its velocity increases, allowing it to reach the bottom more swiftly.

SOGRM$^+$ is a smooth RM$^+$ variant since it operates within the $\mathbb{R}_{\ge 1}^{|A_i|}$, the subset of cone($\boldsymbol{\mathcal{X}}_i$), and updates by facing the adjusted loss vector $\boldsymbol{F}_i(\boldsymbol{\theta}^t) = \langle \boldsymbol{\ell}_i^t, \boldsymbol{x}_i^t \rangle \mathbf{1} - \boldsymbol{\ell}_i^t$ instead of directly using $\boldsymbol{\ell}_i^t$. The convergence results of SOGRM$^+$ are shown in Theorem 6.1.

**Theorem 6.1.** *In smooth games satisfying the weak MVI with $\rho > -\frac{1}{12\sqrt{3}DL_u}$, there always exists an $\eta \in \left(0, \frac{1}{2DL_u}\right)$ satisfying $\frac{1}{2} + \frac{2\rho}{\eta} - 2\eta^2 D^2 L_u^2 > 0$ such that ensures SOGRM$^+$ achieves the asymptotic last-iterate convergence and $O(\frac{1}{\sqrt{t}})$ best-iterate convergence rate in learning an NE of these games, where $D = \max_{i \in \mathcal{N}} |A_i|$ and $L_u = \sqrt{2P^2 + 4L^2}$. Specifically, if all players follow the update rule of SOGRM$^+$, then $r^{tan}(\boldsymbol{x}^{t+\frac{1}{2}}) \to 0$ and $\min_{\tau \in [t]} r^{tan}(\boldsymbol{x}^{\tau+\frac{1}{2}}) \le O(\frac{1}{\sqrt{t}})$ as $t \to \infty$.*

To prove last-iterate and finite-time best-iterate convergence of SOGRM$^+$, we introduce Theorem 6.2 and Lemma 6.3, whose proofs are in Appendix G and H, respectively.

**Theorem 6.2.** *If $\rho > -\frac{1}{12\sqrt{3}DL_u}$, there always exists an $\eta \in \left(0, \frac{1}{2DL_u}\right)$ satisfying $\frac{1}{2} + \frac{2\rho}{\eta} - 2\eta^2 D^2 L_u^2 > 0$ such that ensures $\|\boldsymbol{\theta}^{t+1} - \boldsymbol{\theta}^t\|_2 \to 0$ as $t \to \infty$ and $\min_{\tau \in [t]} \|\boldsymbol{\theta}^{\tau+1} - \boldsymbol{\theta}^\tau\|_2^2 \le O(\frac{1}{t}), \forall t \ge 1$.*

**Lemma 6.3.** *If all players follow the update rule of SOGRM$^+$, $\forall \boldsymbol{x} = [\boldsymbol{x}_i | i \in \mathcal{N}] \in \boldsymbol{\mathcal{X}}$,*

$$\sum_{i \in \mathcal{N}} \langle \boldsymbol{\ell}_i^{t-\frac{1}{2}} - \frac{\boldsymbol{\theta}_i^t - \boldsymbol{\theta}_i^{t+\frac{1}{2}}}{\eta}, \boldsymbol{x}_i - \boldsymbol{x}_i^{t+\frac{1}{2}} \rangle = \sum_{i \in \mathcal{N}} \langle \boldsymbol{\ell}_i^{t+\frac{1}{2}} - \frac{\boldsymbol{\theta}_i^t - \boldsymbol{\theta}_i^{t+1}}{\eta}, \boldsymbol{x}_i - \boldsymbol{x}_i^{t+\frac{1}{2}} \rangle.$$

*Proof.* Now, we prove Theorem 6.1 via our proof paradigm. Firstly, from the equivalence in Section 4 (Eq. (4) can be written as the form in Eq. (5)), the update rule of SOGRM+ can be written as (see details in Appendix I)

$$\boldsymbol{x}_i^{t+\frac{1}{2}} \in \arg\min_{\boldsymbol{x}_i \in \boldsymbol{\mathcal{X}}_i} \{ \langle \boldsymbol{\ell}_i^{t-\frac{1}{2}}, \boldsymbol{x}_i \rangle + q_i^{t-\frac{1}{2}}(\boldsymbol{x}_i) + D_{q_i^{0:t-1}}(\boldsymbol{x}_i, \boldsymbol{x}_i^t) \},$$

$$\boldsymbol{\theta}_i^{t+1} = \boldsymbol{\theta}_i^{t+\frac{1}{2}} - \eta \boldsymbol{F}_i(\boldsymbol{\theta}^{t-\frac{1}{2}}) + \eta \boldsymbol{F}_i(\boldsymbol{\theta}^{t+\frac{1}{2}}), \tag{8}$$

$$q_i^{0:t-1}(\boldsymbol{x}_i) = \frac{\|\boldsymbol{\theta}_i^t\|_1}{\eta} \psi(\boldsymbol{x}_i), \ q_i^{0:t-1}(\boldsymbol{x}_i) + q_i^{t-\frac{1}{2}}(\boldsymbol{x}) = \frac{\|\boldsymbol{\theta}_i^{t+\frac{1}{2}}\|_1}{\eta} \psi(\boldsymbol{x}_i),$$

From the first-order optimality of the first prox-mapping operator in Eq. (8), $\forall \boldsymbol{x} \in \boldsymbol{\mathcal{X}}$, we have

$$\langle \boldsymbol{\ell}_i^{t-\frac{1}{2}} + \nabla_{\boldsymbol{x}_i^{t+\frac{1}{2}}} q_i^{t-\frac{1}{2}}(\boldsymbol{x}_i^{t+\frac{1}{2}}) + \nabla_{\boldsymbol{x}_i^{t+\frac{1}{2}}} D_{q_i^{0:t-1}}(\boldsymbol{x}_i^{t+\frac{1}{2}}, \boldsymbol{x}_i^t), \boldsymbol{x}_i - \boldsymbol{x}_i^{t+\frac{1}{2}} \rangle \geq 0$$

$$\Rightarrow \sum_{i \in \mathcal{N}} \langle \boldsymbol{\ell}_i^{t-\frac{1}{2}} + \nabla_{\boldsymbol{x}_i^{t+\frac{1}{2}}} q_i^{0:t-1}(\boldsymbol{x}^{t+\frac{1}{2}}) + \nabla_{\boldsymbol{x}_i^{t+\frac{1}{2}}} q_i^{t-\frac{1}{2}}(\boldsymbol{x}^{t+\frac{1}{2}}) - \nabla_{\boldsymbol{x}_i^t} q_i^{0:t-1}(\boldsymbol{x}^t), \boldsymbol{x}_i - \boldsymbol{x}_i^{t+\frac{1}{2}} \rangle \geq 0.$$

Then, we have

$$\sum_{i \in \mathcal{N}} \langle \boldsymbol{\ell}_i^{t-\frac{1}{2}} - \frac{\boldsymbol{\theta}_i^t - \boldsymbol{\theta}_i^{t+\frac{1}{2}}}{\eta}, \boldsymbol{x}_i - \boldsymbol{x}_i^{t+\frac{1}{2}} \rangle \geq 0, \ \sum_{i \in \mathcal{N}} \langle \boldsymbol{\ell}_i^{t+\frac{1}{2}} - \frac{\boldsymbol{\theta}_i^t - \boldsymbol{\theta}_i^{t+1}}{\eta}, \boldsymbol{x}_i - \boldsymbol{x}_i^{t+\frac{1}{2}} \rangle \geq 0, \tag{9}$$

where the first inequality is from $\nabla_{\boldsymbol{x}_i^{t+\frac{1}{2}}} q_i^{0:t-1}(\boldsymbol{x}^{t+\frac{1}{2}}) + \nabla_{\boldsymbol{x}_i^{t+\frac{1}{2}}} q_i^{t-\frac{1}{2}}(\boldsymbol{x}^{t+\frac{1}{2}}) = \boldsymbol{\theta}_i^{t+\frac{1}{2}}/\eta$ with $\nabla_{\boldsymbol{x}_i^t} q_i^{0:t-1}(\boldsymbol{x}_i^t) = \boldsymbol{\theta}_i^t/\eta$, and the second inequality is from Lemma 6.3. According to Eq. (9) and the definition of the normal cone, we have

$$-\boldsymbol{\ell}^{t+\frac{1}{2}} + \frac{\boldsymbol{\theta}^t - \boldsymbol{\theta}^{t+1}}{\eta} \in \mathcal{N}_{\boldsymbol{\mathcal{X}}}(\boldsymbol{x}^{t+\frac{1}{2}}), \tag{10}$$

where $\boldsymbol{\ell}^{t+\frac{1}{2}} = [\boldsymbol{\ell}_i^{t+\frac{1}{2}} : i \in \mathcal{N}]$. From the definition of the tangent residual, we obtain

$$r^{tan}(\boldsymbol{x}^{t+\frac{1}{2}}) \leq \|\boldsymbol{\ell}^{t+\frac{1}{2}} - \boldsymbol{\ell}^{t+\frac{1}{2}} + \frac{\boldsymbol{\theta}^t - \boldsymbol{\theta}^{t+1}}{\eta}\|_2 \leq \frac{1}{\eta} \|\boldsymbol{\theta}^t - \boldsymbol{\theta}^{t+1}\|_2. \tag{11}$$

Combining Eq. (11) and Theorem 6.2 ($\|\boldsymbol{\theta}^t - \boldsymbol{\theta}^{t+1}\|_2 \to 0$), we get $r^{tan}(\boldsymbol{x}^{t+\frac{1}{2}}) \to 0$ as $t \to \infty$. Similarly, from Theorem 6.2 ($\min_{\tau \in [t]} \|\boldsymbol{\theta}^{\tau+1} - \boldsymbol{\theta}^\tau\|_2^2 \leq O(\frac{1}{t})$), we get that for $\tau = \arg\min_{\tau \in [t]} \|\boldsymbol{\theta}^\tau - \boldsymbol{\theta}^{\tau+1}\|_2^2$,

$$r^{tan}(\boldsymbol{x}^{\tau+\frac{1}{2}}) \leq \frac{1}{\eta} \|\boldsymbol{\theta}^\tau - \boldsymbol{\theta}^{\tau+1}\|_2 \leq O(\frac{1}{\sqrt{t}}).$$

These complete the proof. □

## 7 Conclusions

We study last-iterate convergence of RM$^+$ variants in learning an NE of games satisfying the weak MVI. We introduce a concise yet novel proof paradigm to analyze last-iterate convergence of RM$^+$ variants. Using this paradigm, we show that two existing smooth RM$^+$ variants, such as SExRM$^+$ and SPRM$^+$, exhibit last-iterate and finite-time best-iterate convergence in games satisfying the MVI. Building on our proof paradigm, we propose SOGRM$^+$, achieving last-iterate and finite-time best-iterate convergence in games satisfying the weak MVI. To the best of our knowledge, this is (i) the first last-iterate convergence for RM$^+$ variants in games satisfying the weak MVI, and (ii) the

first finite-time best-iterate convergence for $RM^+$ variants in games satisfying the MVI or even only the weak MVI. Future directions involve integrating our proof paradigm with other OMD algorithms, such as Reflected Gradient [Hsieh et al., 2019] and composite Fast Extra-Gradient [Cai et al., 2024], to develop novel $RM^+$ variants that achieve stronger last-iterate convergence.

**Limitations.** The primary limitation of our paper is the lack of proof for the convergence of RM+ variants within extensive-form games. RM+ variants are typically integrated with the counterfactual regret minimization (CFR) framework [Zinkevich et al., 2007, Farina et al., 2019] to tackle extensive-form games. Establishing convergence in this context is particularly challenging, and addressing this issue constitutes one of our future research directions. The second limitation lies in the fact that our $SOGRM^+$ requires a larger minimum $\rho$ for convergence compared to OMD variants, as shown in Appendix B. Nonetheless, it's important to emphasize that we are the first to explore the last-iterate convergence of $RM^+$ variants in games that satisfy the weak MVI.

# Acknowledgements

This work is supported in part by the National Natural Science Foundation of China under Grant 62192783, the Jiangsu Science and Technology Major Project BG2024031, the Fundamental Research Funds for the Central Universities (14380128), the Collaborative Innovation Center of Novel Software Technology and Industrialization, and the InnoHK funding.

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

Table 2: Comparisons between the minimum value of $\rho$ of ours and previous studies about OMD variants in games satisfying the weak MVI.

| | Minimum $\rho$ | Algorithms |
|---|---|---|
| Cai and Zheng [2022] | $-\frac{1}{12\sqrt{3}L}$ | OMD Variants |
| Pethick et al. [2023] | $-\frac{1}{2L}$ | OMD Variants |
| Alacaoglu et al. [2024] | $-\frac{1}{L}$ | OMD Variants |
| Cai et al. [2024] | $-\frac{1}{2L}$ | OMD Variants |
| Pethick et al. [2025] | $-\frac{1}{L}$ | OMD Variants |
| **Ours** | $-\frac{1}{12\sqrt{3}D\sqrt{2P^2+4L^2}}$ | RM$^+$ Variants |

## A  Detailed Comparison to Cai et al. [2025]

Now, we provide a detailed comparison to Cai et al. [2025], including the proof techniques, the best-iterate convergence results, the non-finite-time last-iterate convergence results, and the finite-time last-iterate convergence results.

In terms of the proof techniques, our proof paradigm centers on recovering the weak MVI, whereas the methods proposed by Cai et al. [2025] focus on analyzing the limit points of iterates or leveraging the definition of the duality gap in two-player zero-sum games. Notably, our paradigm is applicable to both best-iterate convergence and last-iterate convergence, in contrast to Cai et al. [2025], who require separate proofs for these two types of convergence.

Regarding the best-iterate convergence results, our results apply to games satisfying the weak MVI, whereas those of Cai et al. [2025] are restricted to two-player zero-sum matrix games. The applicability of our best-iterate convergence results is significantly broader than that of Cai et al. [2025].

Concerning the non-finite-time last-iterate convergence results, our findings hold for games satisfying the weak MVI. Cai et al. [2025] divide their results into two sections: the first pertains to two-player zero-sum matrix games, while the second addresses broader games satisfying the MVI. Their results exhibit pointwise convergence for two-player zero-sum matrix games, surpassing our own in strength. However, in games satisfying the MVI, our results align with theirs, yet ours extend further to include games satisfying the weak MVI.

With respect to the finite-time last-iterate convergence results, as mentioned in Section 2, our best-iterate convergence results enable us to extend the finite-time last-iterate convergence findings of Cai et al. [2025] from two-player zero-sum matrix games to games that concurrently satisfy monotonicity and metric subregularity. This exemplifies one of the advantages of our best-iterate convergence results.

## B  Comparison between the Minimum Value of $\rho$ of Ours and OMD variants

We also compare the minimum value of $\rho$ between our SOGRM$^+$ and previous studies on OMD variants in games satisfying the weak MVI, as shown in Appendix B. Focusing on the constrained setting, we present this comparison specifically within this context. Although other algorithms accommodate a broader range of $\rho$, existing work primarily concentrates on OMD variants. We are the first to explore the last-iterate convergence of RM$^+$ variants in games that satisfy the weak MVI.

## C  Detailed Proof of the Phase 1 of Section 4

In this section, we provide a detailed proof for the Phase 1 of Section 4. Before starting our proof, we first present the following lemma.

**Lemma C.1** (Adapted from Theorem 2.2 of Chen and Ye [2011]). *For any $\boldsymbol{y} \in \mathbb{R}^d$, the projection of $\boldsymbol{y}$ onto the simplex $\Delta^d$ is obtained by $[\boldsymbol{y} - \gamma\mathbf{1}]^+$, where $\gamma$ exists and is unique.*

Now, we start our proof. Firstly, considering Eq. (4), from the fact that $\psi(\cdot)$ is the quadratic regularizer (in other words, $\forall \boldsymbol{a}, \boldsymbol{b} \in \mathbb{R}^d, c \in \mathbb{R}, c\psi(\boldsymbol{a}) = c\|\boldsymbol{a}\|_2^2/2, D_{c\psi}(\boldsymbol{a}, \boldsymbol{b}) = c\|\boldsymbol{a} - \boldsymbol{b}\|_2^2/2)$, we have

$$\boldsymbol{\theta}_i^{t_2} \in \underset{\boldsymbol{\theta}_i \in \mathbb{R}_{\geq 1}^{|A_i|}}{\arg\min}\{\langle -\boldsymbol{F}_i(\boldsymbol{\theta}^{t_1}), \boldsymbol{\theta}_i \rangle + \frac{1}{\eta}D_\psi(\boldsymbol{\theta}_i, \boldsymbol{\theta}_i^{t_0})\}$$

$$\Leftrightarrow \boldsymbol{\theta}_i^{t_2} \in \underset{\boldsymbol{\theta}_i \in \mathbb{R}_{\geq 1}^{|A_i|}}{\arg\min}\{\langle -\boldsymbol{F}_i(\boldsymbol{\theta}^{t_1}), \boldsymbol{\theta}_i \rangle + \frac{1}{2\eta}\|\boldsymbol{\theta}_i - \boldsymbol{\theta}_i^{t_0}\|_2^2\}$$

$$\Leftrightarrow \boldsymbol{\theta}_i^{t_2} \in \underset{\boldsymbol{\theta}_i \in \mathbb{R}_{\geq 1}^{|A_i|}}{\arg\min}\{\langle -\boldsymbol{F}_i(\boldsymbol{\theta}^{t_1}), \boldsymbol{\theta}_i \rangle + \frac{1}{2\eta}\|\boldsymbol{\theta}_i\|_2^2 + \frac{1}{2\eta}\|\boldsymbol{\theta}_i^{t_0}\|_2^2 - 2\frac{1}{2\eta}\langle \boldsymbol{\theta}_i, \boldsymbol{\theta}_i^{t_0} \rangle\}$$

Then, we have

$$\boldsymbol{\theta}_i^{t_2} \in \underset{\boldsymbol{\theta}_i \in \mathbb{R}_{\geq 1}^{|A_i|}}{\arg\min}\{\langle -\boldsymbol{F}_i(\boldsymbol{\theta}^{t_1}), \boldsymbol{\theta}_i \rangle + \frac{1}{\eta}D_\psi(\boldsymbol{\theta}_i, \boldsymbol{\theta}_i^{t_0})\}$$

$$\Leftrightarrow \boldsymbol{\theta}_i^{t_2} \in \underset{\boldsymbol{\theta}_i \in \mathbb{R}_{\geq 1}^{|A_i|}}{\arg\min}\{\langle -2\eta\boldsymbol{F}_i(\boldsymbol{\theta}^{t_1}), \boldsymbol{\theta}_i \rangle + \|\boldsymbol{\theta}_i\|_2^2 + \|\boldsymbol{\theta}_i^{t_0}\|_2^2 - 2\langle \boldsymbol{\theta}_i, \boldsymbol{\theta}_i^{t_0} \rangle\}$$

$$\Leftrightarrow \boldsymbol{\theta}_i^{t_2} \in \underset{\boldsymbol{\theta}_i \in \mathbb{R}_{\geq 1}^{|A_i|}}{\arg\min}\{\langle -2\eta\boldsymbol{F}_i(\boldsymbol{\theta}^{t_1}), \boldsymbol{\theta}_i \rangle + \|\boldsymbol{\theta}_i\|_2^2 - 2\langle \boldsymbol{\theta}_i, \boldsymbol{\theta}_i^{t_0} \rangle\}$$

$$\Leftrightarrow \boldsymbol{\theta}_i^{t_2} \in \underset{\boldsymbol{\theta}_i \in \mathbb{R}_{\geq 1}^{|A_i|}}{\arg\min}\{\langle -2\eta\boldsymbol{F}_i(\boldsymbol{\theta}^{t_1}) - 2\boldsymbol{\theta}_i^{t_0}, \boldsymbol{\theta}_i \rangle + \|\boldsymbol{\theta}_i\|_2^2\}$$

$$\Leftrightarrow \boldsymbol{\theta}_i^{t_2} \in \underset{\boldsymbol{\theta}_i \in \mathbb{R}_{\geq 1}^{|A_i|}}{\arg\min}\{\langle -2\eta\boldsymbol{F}_i(\boldsymbol{\theta}^{t_1}) - 2\boldsymbol{\theta}_i^{t_0}, \boldsymbol{\theta}_i \rangle + \|\boldsymbol{\theta}_i\|_2^2 + \|\eta\boldsymbol{F}_i(\boldsymbol{\theta}^{t_1}) + \boldsymbol{\theta}_i^{t_0}\|_2^2\}$$

$$\Leftrightarrow \boldsymbol{\theta}_i^{t_2} \in \underset{\boldsymbol{\theta}_i \in \mathbb{R}_{\geq 1}^{|A_i|}}{\arg\min} \|\eta\boldsymbol{F}_i(\boldsymbol{\theta}^{t_1}) + \boldsymbol{\theta}_i^{t_0} - \boldsymbol{\theta}_i\|_2^2.$$

Since $\boldsymbol{F}_i(\boldsymbol{\theta}^{t_1}) = \langle \frac{\boldsymbol{\theta}_i^{t_1}}{\|\boldsymbol{\theta}_i^{t_1}\|_1}, \boldsymbol{\ell}_i^{t_1} \rangle \mathbf{1} - \boldsymbol{\ell}_i^{t_1}$, we have

$$\boldsymbol{\theta}_i^{t_2} \in \underset{\boldsymbol{\theta}_i \in \mathbb{R}_{\geq 1}^{|A_i|}}{\arg\min}\{\langle -\boldsymbol{F}_i(\boldsymbol{\theta}^{t_1}), \boldsymbol{\theta}_i \rangle + \frac{1}{\eta}D_\psi(\boldsymbol{\theta}_i, \boldsymbol{\theta}_i^{t_0})\}$$

$$\Leftrightarrow \boldsymbol{\theta}_i^{t_2} \in \underset{\boldsymbol{\theta}_i \in \mathbb{R}_{\geq 1}^{|A_i|}}{\arg\min} \|\boldsymbol{\theta}_i^{t_0} + \eta\langle \frac{\boldsymbol{\theta}_i^{t_1}}{\|\boldsymbol{\theta}_i^{t_1}\|_1}, \boldsymbol{\ell}_i^{t_1} \rangle \mathbf{1} - \eta\boldsymbol{\ell}_i^{t_1} - \boldsymbol{\theta}_i\|_2^2. \tag{12}$$

Eq. (12) indicates getting the projection of $\boldsymbol{\theta}_i^{t_0} + \eta\langle \frac{\boldsymbol{\theta}_i^{t_1}}{\|\boldsymbol{\theta}_i^{t_1}\|_1}, \boldsymbol{\ell}_i^{t_1} \rangle \mathbf{1} - \eta\boldsymbol{\ell}_i^{t_1}$ on $\mathbb{R}_{\geq 1}^{|A_i|}$. As analyzed in Section K of Farina et al. [2023], $\forall \boldsymbol{y} \in \mathbb{R}^{|d|}$, projecting $\boldsymbol{y}$ to $\mathbb{R}_{\geq 1}^{|d|}$, if $\|[\boldsymbol{y}]^+\|_1 \geq 1$, then the solution of the projection is $[\boldsymbol{x}]^+$. If $\|[\boldsymbol{y}]^+\|_1 < 1$, return the projection of $\boldsymbol{y}$ on simplex. Let $\boldsymbol{y} = \boldsymbol{\theta}_i^{t_0} + \eta\langle \frac{\boldsymbol{\theta}_i^{t_1}}{\|\boldsymbol{\theta}_i^{t_1}\|_1}, \boldsymbol{\ell}_i^{t_1} \rangle \mathbf{1} - \eta\boldsymbol{\ell}_i^{t_1}$, we have

$$\boldsymbol{\theta}_i^{t_2} = \begin{cases} [\boldsymbol{\theta}_i^{t_0} + \eta\langle \frac{\boldsymbol{\theta}_i^{t_1}}{\|\boldsymbol{\theta}_i^{t_1}\|_1}, \boldsymbol{\ell}_i^{t_1} \rangle \mathbf{1} - \eta\boldsymbol{\ell}_i^{t_1}]^+, & \text{if} \|[\boldsymbol{\theta}_i^{t_0} + \eta\langle \frac{\boldsymbol{\theta}_i^{t_1}}{\|\boldsymbol{\theta}_i^{t_1}\|_1}, \boldsymbol{\ell}_i^{t_1} \rangle \mathbf{1} - \eta\boldsymbol{\ell}_i^{t_1}]^+\|_1 \geq 1, \\ [\boldsymbol{\theta}_i^{t_0} + \eta\langle \frac{\boldsymbol{\theta}_i^{t_1}}{\|\boldsymbol{\theta}_i^{t_1}\|_1}, \boldsymbol{\ell}_i^{t_1} \rangle \mathbf{1} - \eta\boldsymbol{\ell}_i^{t_1} - \beta\mathbf{1}]^+, & \text{if} \|[\boldsymbol{\theta}_i^{t_0} + \eta\langle \frac{\boldsymbol{\theta}_i^{t_1}}{\|\boldsymbol{\theta}_i^{t_1}\|_1}, \boldsymbol{\ell}_i^{t_1} \rangle \mathbf{1} - \eta\boldsymbol{\ell}_i^{t_1}]^+\|_1 < 1, \end{cases} \tag{13}$$

where the top means if $\|[\boldsymbol{\theta}_i^{t_0} + \eta\langle \frac{\boldsymbol{\theta}_i^{t_1}}{\|\boldsymbol{\theta}_i^{t_1}\|_1}, \boldsymbol{\ell}_i^{t_1} \rangle \mathbf{1} - \eta\boldsymbol{\ell}_i^{t_1}]^+\|_1 \geq 1$, the solution is $[\boldsymbol{\theta}_i^{t_0} + \eta\langle \frac{\boldsymbol{\theta}_i^{t_1}}{\|\boldsymbol{\theta}_i^{t_1}\|_1}, \boldsymbol{\ell}_i^{t_1} \rangle \mathbf{1} - \eta\boldsymbol{\ell}_i^{t_1}]^+$, and the bottom implies if $\|[\boldsymbol{\theta}_i^{t_0} + \eta\langle \frac{\boldsymbol{\theta}_i^{t_1}}{\|\boldsymbol{\theta}_i^{t_1}\|_1}, \boldsymbol{\ell}_i^{t_1} \rangle \mathbf{1} - \eta\boldsymbol{\ell}_i^{t_1}]^+\|_1 < 1$, the solution is the projection of $\boldsymbol{\theta}_i^{t_0} + \eta\langle \frac{\boldsymbol{\theta}_i^{t_1}}{\|\boldsymbol{\theta}_i^{t_1}\|_1}, \boldsymbol{\ell}_i^{t_1} \rangle \mathbf{1} - \eta\boldsymbol{\ell}_i^{t_1}$ on simplex. Hence, as shown in Lemma C.1, the closed-form solution in the case where $\|[\boldsymbol{\theta}_i^{t_0} + \eta\langle \frac{\boldsymbol{\theta}_i^{t_1}}{\|\boldsymbol{\theta}_i^{t_1}\|_1}, \boldsymbol{\ell}_i^{t_1} \rangle \mathbf{1} - \eta\boldsymbol{\ell}_i^{t_1}]^+\|_1 < 1$ is $[\boldsymbol{\theta}_i^{t_0} + \eta\langle \frac{\boldsymbol{\theta}_i^{t_1}}{\|\boldsymbol{\theta}_i^{t_1}\|_1}, \boldsymbol{\ell}_i^{t_1} \rangle \mathbf{1} - \eta\boldsymbol{\ell}_i^{t_1} - \beta\mathbf{1}]^+$, where $\beta$ exists and is unique to ensure

$[\boldsymbol{\theta}_i^{t_0} + \eta\langle\frac{\boldsymbol{\theta}_i^{t_1}}{\|\boldsymbol{\theta}_i^{t_1}\|_1}, \boldsymbol{\ell}_i^{t_1}\rangle\mathbf{1} - \eta\boldsymbol{\ell}_i^{t_1} - \beta\mathbf{1}]^+$ in simplex, because (i) $\beta$ is the term $\gamma$ in Lemma C.1, as well as (ii) $\gamma$ exists and is unique. In other words, $\boldsymbol{\theta}_i^{t_2}$ in Eq. (4) can be obtained via

$$\boldsymbol{\theta}_i^{t_2} = \begin{cases} [\boldsymbol{\theta}_i^{t_0} + \eta\langle\frac{\boldsymbol{\theta}_i^{t_1}}{\|\boldsymbol{\theta}_i^{t_1}\|_1}, \boldsymbol{\ell}_i^{t_1}\rangle\mathbf{1} - \eta\boldsymbol{\ell}_i^{t_1}]^+, & \text{if } \|[\boldsymbol{\theta}_i^{t_0} + \eta\langle\frac{\boldsymbol{\theta}_i^{t_1}}{\|\boldsymbol{\theta}_i^{t_1}\|_1}, \boldsymbol{\ell}_i^{t_1}\rangle\mathbf{1} - \eta\boldsymbol{\ell}_i^{t_1}]^+\|_1 \geq 1, \\ [\boldsymbol{\theta}_i^{t_0} + \eta\langle\frac{\boldsymbol{\theta}_i^{t_1}}{\|\boldsymbol{\theta}_i^{t_1}\|_1}, \boldsymbol{\ell}_i^{t_1}\rangle\mathbf{1} - \eta\boldsymbol{\ell}_i^{t_1} - \beta\mathbf{1}]^+, & \text{if } \|[\boldsymbol{\theta}_i^{t_0} + \eta\langle\frac{\boldsymbol{\theta}_i^{t_1}}{\|\boldsymbol{\theta}_i^{t_1}\|_1}, \boldsymbol{\ell}_i^{t_1}\rangle\mathbf{1} - \eta\boldsymbol{\ell}_i^{t_1}]^+\|_1 < 1, \end{cases}$$

Secondly, for Eq. (5), we have

$$\boldsymbol{x}_i^{t_2} \in \underset{\boldsymbol{x}_i\in\boldsymbol{\mathcal{X}}_i}{\arg\min}\{\langle\boldsymbol{\ell}_i^{t_1}, \boldsymbol{x}_i\rangle + f_i(\boldsymbol{x}_i) + D_{h_i}(\boldsymbol{x}_i, \boldsymbol{x}_i^{t_0})\}$$

$$\Leftrightarrow \boldsymbol{x}_i^{t_2} \in \underset{\boldsymbol{x}_i\in\boldsymbol{\mathcal{X}}_i}{\arg\min}\{\langle\boldsymbol{\ell}_i^{t_1}, \boldsymbol{x}_i\rangle + \frac{\|\boldsymbol{\theta}_i^{t_2}\|_1}{\eta}\psi(\boldsymbol{x}_i) - \frac{\|\boldsymbol{\theta}_i^{t_0}\|_1}{\eta}\psi(\boldsymbol{x}_i) + D_{\frac{\|\boldsymbol{\theta}_i^{t_0}\|_1}{\eta}\psi}(\boldsymbol{x}_i, \boldsymbol{x}_i^{t_0})\}.$$

Since $\psi(\cdot)$ is the quadratic regularizer (in other words, $\forall\boldsymbol{a}, \boldsymbol{b} \in \mathbb{R}^d, c \in \mathbb{R}, c\psi(\boldsymbol{a}) = c\|\boldsymbol{a}\|_2^2/2$, $D_{c\psi}(\boldsymbol{a}, \boldsymbol{b}) = c\|\boldsymbol{a}-\boldsymbol{b}\|_2^2/2)$, we have

$$\boldsymbol{x}_i^{t_2} \in \underset{\boldsymbol{x}_i\in\boldsymbol{\mathcal{X}}_i}{\arg\min}\{\langle\boldsymbol{\ell}_i^{t_1}, \boldsymbol{x}_i\rangle + f_i(\boldsymbol{x}_i) + D_{h_i}(\boldsymbol{x}_i, \boldsymbol{x}_i^{t_0})\}$$

$$\Leftrightarrow \boldsymbol{x}_i^{t_2} \in \underset{\boldsymbol{x}_i\in\boldsymbol{\mathcal{X}}_i}{\arg\min}\{\langle\boldsymbol{\ell}_i^{t_1}, \boldsymbol{x}_i\rangle + \frac{\|\boldsymbol{\theta}_i^{t_2}\|_1}{2\eta}\|\boldsymbol{x}_i\|_2^2 - \frac{\|\boldsymbol{\theta}_i^{t_0}\|_1}{2\eta}\|\boldsymbol{x}_i\|_2^2 + \frac{\|\boldsymbol{\theta}_i^{t_0}\|_1}{2\eta}\|\boldsymbol{x}_i - \boldsymbol{x}_i^{t_0}\|_2^2\}$$

$$\Leftrightarrow \boldsymbol{x}_i^{t_2} \in \underset{\boldsymbol{x}_i\in\boldsymbol{\mathcal{X}}_i}{\arg\min}\{\langle 2\eta\boldsymbol{\ell}_i^{t_1}, \boldsymbol{x}_i\rangle + \|\boldsymbol{\theta}_i^{t_2}\|_1\|\boldsymbol{x}_i\|_2^2 - \|\boldsymbol{\theta}_i^{t_0}\|_1\|\boldsymbol{x}_i\|_2^2 + \|\boldsymbol{\theta}_i^{t_0}\|_1\|\boldsymbol{x}_i - \boldsymbol{x}_i^{t_0}\|_2^2\}$$

$$\Leftrightarrow \boldsymbol{x}_i^{t_2} \in \underset{\boldsymbol{x}_i\in\boldsymbol{\mathcal{X}}_i}{\arg\min}\{\langle 2\eta\boldsymbol{\ell}_i^{t_1}, \boldsymbol{x}_i\rangle + \|\boldsymbol{\theta}_i^{t_2}\|_1\|\boldsymbol{x}_i\|_2^2 - \|\boldsymbol{\theta}_i^{t_0}\|_1\|\boldsymbol{x}_i\|_2^2 + \|\boldsymbol{\theta}_i^{t_0}\|_1\|\boldsymbol{x}_i\|_2^2 +$$
$$\|\boldsymbol{\theta}_i^{t_0}\|_1\|\boldsymbol{x}_i^{t_0}\|_2^2 - 2\|\boldsymbol{\theta}_i^{t_0}\|_1\langle\boldsymbol{x}_i, \boldsymbol{x}_i^{t_0}\rangle\}$$

$$\Leftrightarrow \boldsymbol{x}_i^{t_2} \in \underset{\boldsymbol{x}_i\in\boldsymbol{\mathcal{X}}_i}{\arg\min}\{\langle 2\eta\boldsymbol{\ell}_i^{t_1}, \boldsymbol{x}_i\rangle + \|\boldsymbol{\theta}_i^{t_2}\|_1\|\boldsymbol{x}_i\|_2^2 - 2\|\boldsymbol{\theta}_i^{t_0}\|_1\langle\boldsymbol{x}_i, \boldsymbol{x}_i^{t_0}\rangle\}$$

$$\Leftrightarrow \boldsymbol{x}_i^{t_2} \in \underset{\boldsymbol{x}_i\in\boldsymbol{\mathcal{X}}_i}{\arg\min}\{\langle 2\eta\boldsymbol{\ell}_i^{t_1} - 2\|\boldsymbol{\theta}_i^{t_0}\|_1\boldsymbol{x}_i^{t_0}, \boldsymbol{x}_i\rangle + \|\boldsymbol{\theta}_i^{t_2}\|_1\|\boldsymbol{x}_i\|_2^2\} \quad (14)$$

$$\Leftrightarrow \boldsymbol{x}_i^{t_2} \in \underset{\boldsymbol{x}_i\in\boldsymbol{\mathcal{X}}_i}{\arg\min}\{2\frac{\langle\eta\boldsymbol{\ell}_i^{t_1} - \|\boldsymbol{\theta}_i^{t_0}\|_1\boldsymbol{x}_i^{t_0}, \boldsymbol{x}_i\rangle}{\|\boldsymbol{\theta}_i^{t_2}\|_1} + \|\boldsymbol{x}_i\|_2^2\}$$

$$\Leftrightarrow \boldsymbol{x}_i^{t_2} \in \underset{\boldsymbol{x}_i\in\boldsymbol{\mathcal{X}}_i}{\arg\min}\{2\frac{\langle\eta\boldsymbol{\ell}_i^{t_1} - \|\boldsymbol{\theta}_i^{t_0}\|_1\boldsymbol{x}_i^{t_0}, \boldsymbol{x}_i\rangle}{\|\boldsymbol{\theta}_i^{t_2}\|_1} + \|\boldsymbol{x}_i\|_2^2 + \|\frac{\eta\boldsymbol{\ell}_i^{t_1} - \|\boldsymbol{\theta}_i^{t_0}\|_1\boldsymbol{x}_i^{t_0}}{\|\boldsymbol{\theta}_i^{t_2}\|_1}\|_2^2\}$$

$$\Leftrightarrow \boldsymbol{x}_i^{t_2} \in \underset{\boldsymbol{x}_i\in\boldsymbol{\mathcal{X}}_i}{\arg\min}\|\frac{\|\boldsymbol{\theta}_i^{t_0}\|_1\boldsymbol{x}_i^{t_0} - \eta\boldsymbol{\ell}_i^{t_1}}{\|\boldsymbol{\theta}_i^{t_2}\|_1} - \boldsymbol{x}_i\|_2^2$$

$$\Leftrightarrow \boldsymbol{x}_i^{t_2} \in \underset{\boldsymbol{x}_i\in\boldsymbol{\mathcal{X}}_i}{\arg\min}\|\frac{\boldsymbol{\theta}_i^{t_0} - \eta\boldsymbol{\ell}_i^{t_1}}{\|\boldsymbol{\theta}_i^{t_2}\|_1} - \boldsymbol{x}_i\|_2^2,$$

where the last line is from $\boldsymbol{\theta}_i^{t_0} = \|\boldsymbol{\theta}_i^{t_0}\|_1\boldsymbol{x}_i^{t_0}$ ($\boldsymbol{x}_i^{t_0} = \frac{\boldsymbol{\theta}_i^{t_0}}{\|\boldsymbol{\theta}_i^{t_0}\|_1}$). Since $\boldsymbol{\mathcal{X}}_i$ is simplex, Eq. (14) indicates getting the projection of $\frac{\boldsymbol{\theta}_i^{t_0} - \eta\boldsymbol{\ell}_i^{t_1}}{\|\boldsymbol{\theta}_i^{t_2}\|_1}$ on simplex. Therefore, as shown in Lemma C.1, the closed-form solution of Eq. (14) is Eq. (15):

$$\boldsymbol{x}_i^{t_2} = [\frac{\boldsymbol{\theta}_i^{t_0} - \eta\boldsymbol{\ell}_i^{t_1}}{\|\boldsymbol{\theta}_i^{t_2}\|_1} - \alpha'\mathbf{1}]^+ = \frac{[\boldsymbol{\theta}_i^{t_0} - \alpha\mathbf{1} - \eta\boldsymbol{\ell}_i^{t_1}]^+}{\|\boldsymbol{\theta}_i^{t_2}\|_1}, \quad (15)$$

where $\alpha' = \frac{\alpha}{\|\boldsymbol{\theta}_i^{t_2}\|_1}$. Notably, $\alpha$ exists and is unique to ensure $\frac{[\boldsymbol{\theta}_i^{t_0} - \alpha\mathbf{1} - \eta\boldsymbol{\ell}_i^{t_1}]^+}{\|\boldsymbol{\theta}_i^{t_2}\|_1}$ in simplex, because (i) $\alpha'$ is the term $\gamma$ in Lemma C.1, as well as (ii) $\gamma$ exists and is unique. In other words, the update rule in Eq. (5) can be written as

$$\boldsymbol{x}_i^{t_2} = \frac{[\boldsymbol{\theta}_i^{t_0} - \alpha\mathbf{1} - \eta\boldsymbol{\ell}_i^{t_1}]^+}{\|\boldsymbol{\theta}_i^{t_2}\|_1}. \quad (16)$$

Note that $\boldsymbol{\theta}_i^{t_2}$, $\boldsymbol{\theta}_i^{t_0}$, and $\boldsymbol{\ell}_i^{t_1}$ are the same between Eq. (4) and Eq. (5).

Thirdly, assume $\alpha$ in Eq. (16) is

$$
\alpha = \begin{cases} -\eta\langle \frac{\boldsymbol{\theta}_i^{t_1}}{\|\boldsymbol{\theta}_i^{t_1}\|_1}, \boldsymbol{\ell}_i^{t_1}\rangle & \text{if } \|[\boldsymbol{\theta}_i^{t_0} + \eta\langle \frac{\boldsymbol{\theta}_i^{t_1}}{\|\boldsymbol{\theta}_i^{t_1}\|_1}, \boldsymbol{\ell}_i^{t_1}\rangle \mathbf{1} - \eta\boldsymbol{\ell}_i^{t_1}]^+\|_1 \geq 1, \\ -\eta\langle \frac{\boldsymbol{\theta}_i^{t_1}}{\|\boldsymbol{\theta}_i^{t_1}\|_1}, \boldsymbol{\ell}_i^{t_1}\rangle + \beta & \text{if } \|[\boldsymbol{\theta}_i^{t_0} + \eta\langle \frac{\boldsymbol{\theta}_i^{t_1}}{\|\boldsymbol{\theta}_i^{t_1}\|_1}, \boldsymbol{\ell}_i^{t_1}\rangle \mathbf{1} - \eta\boldsymbol{\ell}_i^{t_1}]^+\|_1 < 1, \end{cases}
\tag{17}
$$

By using Eq. (13), we have $[\boldsymbol{\theta}_i^{t_0} - \alpha\mathbf{1} - \eta\boldsymbol{\ell}_i^{t_1}]^+ = \boldsymbol{\theta}_i^{t_2}$. Therefore, substituting Eq. (17) into Eq. (16), we get that the update rule in Eq. (16) (or Eq. (5)) can be written as (since $[\boldsymbol{\theta}_i^{t_0} - \alpha\mathbf{1} - \eta\boldsymbol{\ell}_i^{t_1}]^+ = \boldsymbol{\theta}_i^{t_2}$)

$$
\boldsymbol{x}_i^{t_2} = \frac{[\boldsymbol{\theta}_i^{t_0} - \alpha\mathbf{1} - \eta\boldsymbol{\ell}_i^{t_1}]^+}{\|\boldsymbol{\theta}_i^{t_2}\|_1} = \frac{\boldsymbol{\theta}_i^{t_2}}{\|\boldsymbol{\theta}_i^{t_2}\|_1},
\tag{18}
$$

which enables $\boldsymbol{x}_i^{t_2}$ in simplex since $\|\boldsymbol{x}_i^{t_2}\|_1$ and $\boldsymbol{x}_i^{t_2} \geq \mathbf{0}$ (the $\boldsymbol{x}_i^{t_2}$ in Eq. (5)). In addition, as mention in the text around Eq. (15), we have $\alpha$ is unique to ensure $\boldsymbol{x}_i^{t_2} = \frac{[\boldsymbol{\theta}_i^{t_0} - \alpha\mathbf{1} - \eta\boldsymbol{\ell}_i^{t_1}]^+}{\|\boldsymbol{\theta}_i^{t_2}\|_1}$ in simplex. Hence, the value of $\alpha$ must be same as Eq. (17) shown! Also, note the $\boldsymbol{x}_i^{t_2}$ in Eq. (4) is obtained via

$$
\boldsymbol{x}_i^{t_2} = \frac{\boldsymbol{\theta}_i^{t_2}}{\|\boldsymbol{\theta}_i^{t_2}\|_1}.
\tag{19}
$$

Therefore, combining Eq. (18) with Eq. (19), we get that the update rule in Eq. (4) is the same as Eq. (5). It completes the proof.

## D    Proof of Theorem 5.1

*Proof.* Now, we start to prove last-iterate and finite-time best-iterate convergence of SExRM$^+$ and SPRM$^+$. Firstly, from the analysis in Section 4 that Eq. (4) can be written as the form in Eq. (5), the update rule of SExRM$^+$ can be written as (see details in Appendix I)

$$
\boldsymbol{x}_i^{t+\frac{1}{2}} \in \arg\min_{\boldsymbol{x}_i \in \mathcal{X}_i}\{\langle \boldsymbol{\ell}_i^t, \boldsymbol{x}_i\rangle + q_i^{t-\frac{1}{2}}(\boldsymbol{x}_i) + D_{q_i^{0:t-1}}(\boldsymbol{x}_i, \boldsymbol{x}_i^t)\}, \ q_i^{0:t-1}(\boldsymbol{x}_i) = \frac{\|\boldsymbol{\theta}_i^t\|_1}{\eta}\psi(\boldsymbol{x}_i),
$$

$$
\boldsymbol{x}_i^{t+1} \in \arg\min_{\boldsymbol{x}_i \in \mathcal{X}_i}\{\langle \boldsymbol{\ell}_i^{t+\frac{1}{2}}, \boldsymbol{x}_i\rangle + q_i^t(\boldsymbol{x}_i) + D_{q_i^{0:t-1}}(\boldsymbol{x}_i, \boldsymbol{x}_i^t)\},
\tag{20}
$$

$$
q_i^{0:t-1}(\boldsymbol{x}_i) + q_i^{t-\frac{1}{2}}(\boldsymbol{x}) = \frac{\|\boldsymbol{\theta}_i^{t+\frac{1}{2}}\|_1}{\eta}\psi(\boldsymbol{x}_i), \ q_i^{0:t-1}(\boldsymbol{x}_i) + q_i^t(\boldsymbol{x}_i) = \frac{\|\boldsymbol{\theta}_i^{t+1}\|_1}{\eta}\psi(\boldsymbol{x}_i).
$$

Similarly, the update rule of SPRM$^+$ can be written as

$$
\boldsymbol{x}_i^{t+\frac{1}{2}} \in \arg\min_{\boldsymbol{x}_i \in \mathcal{X}_i}\{\langle \boldsymbol{\ell}_i^{t-\frac{1}{2}}, \boldsymbol{x}_i\rangle + q_i^{t-\frac{1}{2}}(\boldsymbol{x}_i) + D_{q_i^{0:t-1}}(\boldsymbol{x}_i, \boldsymbol{x}_i^t)\}, \ q_i^{0:t-1}(\boldsymbol{x}_i) = \frac{\|\boldsymbol{\theta}_i^t\|_1}{\eta}\psi(\boldsymbol{x}_i),
$$

$$
\boldsymbol{x}_i^{t+1} \in \arg\min_{\boldsymbol{x}_i \in \mathcal{X}_i}\{\langle \boldsymbol{\ell}_i^{t+\frac{1}{2}}, \boldsymbol{x}_i\rangle + q_i^t(\boldsymbol{x}_i) + D_{q_i^{0:t-1}}(\boldsymbol{x}_i, \boldsymbol{x}_i^t)\},
\tag{21}
$$

$$
q_i^{0:t-1}(\boldsymbol{x}_i) + q_i^{t-\frac{1}{2}}(\boldsymbol{x}) = \frac{\|\boldsymbol{\theta}_i^{t+\frac{1}{2}}\|_1}{\eta}\psi(\boldsymbol{x}_i), \ q_i^{0:t-1}(\boldsymbol{x}_i) + q_i^t(\boldsymbol{x}_i) = \frac{\|\boldsymbol{\theta}_i^{t+1}\|_1}{\eta}\psi(\boldsymbol{x}_i).
$$

The MVI is recovered since the feedback $\boldsymbol{\ell}_i^{t-\frac{1}{2}}$, $\boldsymbol{\ell}_i^t$, and $\boldsymbol{\ell}_i^{t+\frac{1}{2}}$ are the loss gradients of the original game that satisfy the MVI. Now, we prove that the tangent residual of the strategy profiles $\boldsymbol{x}^t$ converges to 0. From the analysis in Phase 2 of Section 4, according to the second prox-mapping operator in Eq. (20) and (21), $\forall \boldsymbol{x} \in \mathcal{X}$, we have

$$
\langle \boldsymbol{\ell}^{t+\frac{1}{2}} + \frac{\boldsymbol{\theta}^{t+1} - \boldsymbol{\theta}^t}{\eta}, \boldsymbol{x} - \boldsymbol{x}^{t+1}\rangle \geq 0 \Leftrightarrow -\boldsymbol{\ell}^{t+\frac{1}{2}} + \frac{\boldsymbol{\theta}^t - \boldsymbol{\theta}^{t+1}}{\eta} \in \mathcal{N}_{\mathcal{X}}(\boldsymbol{x}^{t+1}).
\tag{22}
$$

From the definition of the tangent residual, we obtain

$$
\begin{aligned}
r^{tan}(\boldsymbol{x}^{t+1}) &\leq \|\boldsymbol{\ell}^{t+1} - \boldsymbol{\ell}^{t+\frac{1}{2}} + \frac{\boldsymbol{\theta}^t - \boldsymbol{\theta}^{t+1}}{\eta}\|_2 \\
&\leq \|\boldsymbol{\ell}^{t+1} - \boldsymbol{\ell}^{t+\frac{1}{2}}\|_2 + \frac{1}{\eta}\|\boldsymbol{\theta}^t - \boldsymbol{\theta}^{t+1}\|_2 \\
&\leq L\|\boldsymbol{x}^{t+1} - \boldsymbol{x}^{t+\frac{1}{2}}\|_2 + \frac{1}{\eta}\|\boldsymbol{\theta}^t - \boldsymbol{\theta}^{t+\frac{1}{2}}\|_2 + \frac{1}{\eta}\|\boldsymbol{\theta}^{t+\frac{1}{2}} - \boldsymbol{\theta}^{t+1}\|_2,
\end{aligned}
\tag{23}
$$

where the last inequality is from the smoothness of the smooth games. Then, using Lemma 5.4 with $\boldsymbol{a} = \boldsymbol{\theta}^{t+1}$ and $\boldsymbol{b} = \boldsymbol{\theta}^{t+\frac{1}{2}}$, we have

$$r^{tan}(\boldsymbol{x}^{t+1}) \leq L\sqrt{D}\|\boldsymbol{\theta}^{t+1} - \boldsymbol{\theta}^{t+\frac{1}{2}}\|_2 + \frac{1}{\eta}\|\boldsymbol{\theta}^t - \boldsymbol{\theta}^{t+\frac{1}{2}}\|_2 + \frac{1}{\eta}\|\boldsymbol{\theta}^{t+\frac{1}{2}} - \boldsymbol{\theta}^{t+1}\|_2. \tag{24}$$

From Theorem 5.2 and 5.3 ($\|\boldsymbol{\theta}^t - \boldsymbol{\theta}^{t+\frac{1}{2}}\|_2 \to 0$ and $\|\boldsymbol{\theta}^{t+\frac{1}{2}} - \boldsymbol{\theta}^{t+1}\|_2 \to 0$), we get $r^{tan}(\boldsymbol{x}^t) \to 0$ as $t \to \infty$. Similarly, we get

$$(r^{tan}(\boldsymbol{x}^{t+1}))^2 \leq \|\boldsymbol{\ell}^{t+1} - \boldsymbol{\ell}^{t+\frac{1}{2}} + \frac{\boldsymbol{\theta}^t - \boldsymbol{\theta}^{t+1}}{\eta}\|_2^2 \leq 2\|\boldsymbol{\ell}^{t+1} - \boldsymbol{\ell}^{t+\frac{1}{2}}\|_2^2 + \frac{2}{\eta^2}\|\boldsymbol{\theta}^t - \boldsymbol{\theta}^{t+1}\|_2^2$$

$$\leq 2L^2\|\boldsymbol{x}^{t+1} - \boldsymbol{x}^{t+\frac{1}{2}}\|_2 + \frac{2}{\eta^2}\|\boldsymbol{\theta}^t - \boldsymbol{\theta}^{t+1}\|_2^2.$$

By using Lemma 5.4, we get

$$(r^{tan}(\boldsymbol{x}^{t+1}))^2 \leq 2L^2D^2\|\boldsymbol{\theta}^{t+1} - \boldsymbol{\theta}^{t+\frac{1}{2}}\|_2^2 + \frac{4}{\eta^2}\|\boldsymbol{\theta}^t - \boldsymbol{\theta}^{t+\frac{1}{2}}\|_2^2 + \frac{4}{\eta^2}\|\boldsymbol{\theta}^{t+\frac{1}{2}} - \boldsymbol{\theta}^{t+1}\|_2^2$$

$$\leq \left(2L^2D^2 + \frac{4}{\eta^2}\right)\left(\|\boldsymbol{\theta}^{t+1} - \boldsymbol{\theta}^{t+\frac{1}{2}}\|_2^2 + \|\boldsymbol{\theta}^t - \boldsymbol{\theta}^{t+\frac{1}{2}}\|_2^2\right).$$

Thus, from Theorem 5.2 and 5.3 ($\min_{\tau\in[t]}(\|\boldsymbol{\theta}^{\tau+\frac{1}{2}} - \boldsymbol{\theta}^\tau\|_2^2 + \|\boldsymbol{\theta}^{\tau+1} - \boldsymbol{\theta}^{\tau+\frac{1}{2}}\|_2^2) \leq O(\frac{1}{t})$), we get that for $\tau = \arg\min_{\tau\in[t]}(\|\boldsymbol{\theta}^{\tau+\frac{1}{2}} - \boldsymbol{\theta}^\tau\|_2^2 + \|\boldsymbol{\theta}^{\tau+1} - \boldsymbol{\theta}^{\tau+\frac{1}{2}}\|_2^2)$,

$$r^{tan}(\boldsymbol{x}^{\tau+1}) \leq \sqrt{O\left(\left(\|\boldsymbol{\theta}^{\tau+\frac{1}{2}} - \boldsymbol{\theta}^\tau\|_2^2 + \|\boldsymbol{\theta}^{\tau+1} - \boldsymbol{\theta}^{\tau+\frac{1}{2}}\|_2^2\right)\right)} \leq O(\frac{1}{\sqrt{t}}).$$

These complete the proof. $\qquad\square$

# E Proof of Theorem 5.2

**Lemma E.1.** *(Proof is in Appendix E.1) Let $\boldsymbol{x}^* \in \mathcal{X}^*$ that satisfies the MVI and assume all players follow the update rule of SExRM$^+$, then for every iteration $t \geq 1$, it holds that $\|\boldsymbol{\theta}^{t+1} - \boldsymbol{x}^*\|_2^2 \leq \|\boldsymbol{\theta}^t - \boldsymbol{x}^*\|_2^2 - (1 - \eta D L_u)\left(\|\boldsymbol{\theta}^{t+\frac{1}{2}} - \boldsymbol{\theta}^t\|_2^2 + \|\boldsymbol{\theta}^{t+1} - \boldsymbol{\theta}^{t+\frac{1}{2}}\|_2^2\right)$, where $D = \max_{i\in\mathcal{N}}|A_i|$ and $L_u = \sqrt{2P^2 + 4L^2}$.*

From Lemma E.1, we have

$$\|\boldsymbol{\theta}^{t+1} - \boldsymbol{x}^*\|_2^2 - \|\boldsymbol{\theta}^t - \boldsymbol{x}^*\|_2^2 \leq -(1 - \eta D L_u)\left(\|\boldsymbol{\theta}^{t+\frac{1}{2}} - \boldsymbol{\theta}^t\|_2^2 + \|\boldsymbol{\theta}^{t+1} - \boldsymbol{\theta}^{t+\frac{1}{2}}\|_2^2\right). \tag{25}$$

Assume $\left(\|\boldsymbol{\theta}^{t+\frac{1}{2}} - \boldsymbol{\theta}^t\|_2^2 + \|\boldsymbol{\theta}^{t+1} - \boldsymbol{\theta}^{t+\frac{1}{2}}\|_2^2\right)$ does not converge to 0. Then, from Eq. (25), we have

$$\|\boldsymbol{\theta}^{T+1} - \boldsymbol{x}^*\|_2^2 \leq \|\boldsymbol{\theta}^1 - \boldsymbol{x}^*\|_2^2 - \sum_{t=1}^T(1 - \eta D L_u)\left(\|\boldsymbol{\theta}^{t+\frac{1}{2}} - \boldsymbol{\theta}^t\|_2^2 + \|\boldsymbol{\theta}^{t+1} - \boldsymbol{\theta}^{t+\frac{1}{2}}\|_2^2\right)$$

In addition, since $\eta < \frac{1}{DL_u}$, we have $(1 - \eta D L_u) > 0$. Therefore, as $T \to \infty$, $\|\boldsymbol{\theta}^{T+1} - \boldsymbol{x}^*\|_2^2 \leq \|\boldsymbol{\theta}^1 - \boldsymbol{x}^*\|_2^2 - \sum_{t=1}^T\left(\|\boldsymbol{\theta}^{t+\frac{1}{2}} - \boldsymbol{\theta}^t\|_2^2 + \|\boldsymbol{\theta}^{t+1} - \boldsymbol{\theta}^{t+\frac{1}{2}}\|_2^2\right) = -\infty$, which contracts that $\|\boldsymbol{\theta}^{T+1} - \boldsymbol{x}^*\|_2^2 \geq 0$. Therefore, we have $\left(\|\boldsymbol{\theta}^{t+\frac{1}{2}} - \boldsymbol{\theta}^t\|_2^2 + \|\boldsymbol{\theta}^{t+1} - \boldsymbol{\theta}^{t+\frac{1}{2}}\|_2^2\right)$ converge to 0 as $t \to \infty$.

In addition, from $\eta < \frac{1}{DL_u}$ and Eq. (25), we have

$$\sum_{t=1}^T\left(\|\boldsymbol{\theta}^{t+\frac{1}{2}} - \boldsymbol{\theta}^t\|_2^2 + \|\boldsymbol{\theta}^{t+1} - \boldsymbol{\theta}^{t+\frac{1}{2}}\|_2^2\right) \leq \frac{\|\boldsymbol{\theta}^1 - \boldsymbol{x}^*\|_2^2 - \|\boldsymbol{\theta}^{T+1} - \boldsymbol{x}^*\|_2^2}{1 - \eta D L_u} \leq C,$$

where $C$ is a constant which depends on $\boldsymbol{\theta}^1$, $\boldsymbol{x}^*$, $\eta$, $D$, and $L_u$. Therefore, we get

$$T\min_{t\in T}\left(\|\boldsymbol{\theta}^{t+\frac{1}{2}} - \boldsymbol{\theta}^t\|_2^2 + \|\boldsymbol{\theta}^{t+1} - \boldsymbol{\theta}^{t+\frac{1}{2}}\|_2^2\right) \leq \sum_{t=1}^T\left(\|\boldsymbol{\theta}^{t+\frac{1}{2}} - \boldsymbol{\theta}^t\|_2^2 + \|\boldsymbol{\theta}^{t+1} - \boldsymbol{\theta}^{t+\frac{1}{2}}\|_2^2\right) \leq C,$$

which implies

$$\min_{t\in T}\left(\|\boldsymbol{\theta}^{t+\frac{1}{2}} - \boldsymbol{\theta}^t\|_2^2 + \|\boldsymbol{\theta}^{t+1} - \boldsymbol{\theta}^{t+\frac{1}{2}}\|_2^2\right) \leq \frac{C}{T}.$$

## E.1 Proof of Lemma E.1

**Lemma E.2.** *(Proof is in Appendix E.2) Assume all players follow the update rule of SExRM$^+$, then for any $\boldsymbol{\theta} \in \mathbb{R}_{\geq 1}^{|\boldsymbol{\mathcal{X}}|}$, we have*

$$
D_\psi(\boldsymbol{\theta},\boldsymbol{\theta}^{t+1}) - D_\psi(\boldsymbol{\theta},\boldsymbol{\theta}^t)
$$
$$
\leq -\eta\langle \boldsymbol{F}(\boldsymbol{\theta}^{t+\frac{1}{2}}),\boldsymbol{\theta}\rangle + \eta\langle \boldsymbol{F}(\boldsymbol{\theta}^{t+\frac{1}{2}}) - \boldsymbol{F}(\boldsymbol{\theta}^t),\boldsymbol{\theta}^{t+1} - \boldsymbol{\theta}^{t+\frac{1}{2}}\rangle - D_\psi(\boldsymbol{\theta}^{t+1},\boldsymbol{\theta}^{t+\frac{1}{2}}) - D_\psi(\boldsymbol{\theta}^{t+\frac{1}{2}},\boldsymbol{\theta}^t).
$$

Substituting $\boldsymbol{\theta} = \boldsymbol{x}^* \in \boldsymbol{\mathcal{X}}^*$ that satisfie s the MVI into Lemma E.2, we get

$$
D_\psi(\boldsymbol{x}^*,\boldsymbol{\theta}^{t+1}) - D_\psi(\boldsymbol{x}^*,\boldsymbol{\theta}^t)
$$
$$
\leq -\eta\langle \boldsymbol{F}(\boldsymbol{\theta}^{t+\frac{1}{2}}),\boldsymbol{x}^*\rangle - D_\psi(\boldsymbol{\theta}^{t+1},\boldsymbol{\theta}^{t+\frac{1}{2}}) - D_\psi(\boldsymbol{\theta}^{t+\frac{1}{2}},\boldsymbol{\theta}^t) + \eta\langle \boldsymbol{F}(\boldsymbol{\theta}^{t+\frac{1}{2}}) - \boldsymbol{F}(\boldsymbol{\theta}^t),\boldsymbol{\theta}^{t+1} - \boldsymbol{\theta}^{t+\frac{1}{2}}\rangle. \tag{26}
$$

For the first term of the right-hand side of Eq. (26), we have

$$
-\eta\sum_{i\in\mathcal{N}}\langle \boldsymbol{F}_i(\boldsymbol{\theta}^{t+\frac{1}{2}}),\boldsymbol{x}_i^*\rangle = -\eta\sum_{i\in\mathcal{N}}\langle \boldsymbol{\ell}_i^{t+\frac{1}{2}},\boldsymbol{x}_i^{t+\frac{1}{2}} - \boldsymbol{x}_i^*\rangle = -\eta\langle \boldsymbol{\ell}^{t+\frac{1}{2}},\boldsymbol{x}^{t+\frac{1}{2}} - \boldsymbol{x}^*\rangle \leq 0. \tag{27}
$$

where the last inequality is from the definition of MVI (Section 3.1). For the fourth term of the right-hand side of Eq. (26), we have

$$
\eta\langle \boldsymbol{F}(\boldsymbol{\theta}^{t+\frac{1}{2}}) - \boldsymbol{F}(\boldsymbol{\theta}^t),\boldsymbol{\theta}^{t+1} - \boldsymbol{\theta}^{t+\frac{1}{2}}\rangle \leq \eta\|\boldsymbol{F}(\boldsymbol{\theta}^{t+\frac{1}{2}}) - \boldsymbol{F}(\boldsymbol{\theta}^t)\|_2\|\boldsymbol{\theta}^{t+1} - \boldsymbol{\theta}^{t+\frac{1}{2}}\|_2
$$
$$
\leq \eta D L_u\|\boldsymbol{\theta}^{t+\frac{1}{2}} - \boldsymbol{\theta}^t\|_2\|\boldsymbol{\theta}^{t+1} - \boldsymbol{\theta}^{t+\frac{1}{2}}\|_2, \tag{28}
$$

where the last line is from Lemma 5.2 of Farina et al. [2023] ($\|\boldsymbol{F}(\boldsymbol{\theta}) - \boldsymbol{F}(\boldsymbol{\theta}')\|_2 \leq DL_u\|\boldsymbol{\theta} - \boldsymbol{\theta}'\|_2, \forall \boldsymbol{\theta},\boldsymbol{\theta}' \in \mathbb{R}_{\geq 1}^{|\boldsymbol{\mathcal{X}}|}$, where $D, L_u$ are in Theorem 5.1). Combining Eq. (26), (27), and (28), we get

$$
D_\psi(\boldsymbol{x}^*,\boldsymbol{\theta}^{t+1}) - D_\psi(\boldsymbol{x}^*,\boldsymbol{\theta}^t)
$$
$$
\leq -D_\psi(\boldsymbol{\theta}^{t+1},\boldsymbol{\theta}^{t+\frac{1}{2}}) - D_\psi(\boldsymbol{\theta}^{t+\frac{1}{2}},\boldsymbol{\theta}^t) + \eta D L_u\|\boldsymbol{\theta}^{t+\frac{1}{2}} - \boldsymbol{\theta}^t\|_2\|\boldsymbol{\theta}^{t+1} - \boldsymbol{\theta}^{t+\frac{1}{2}}\|_2
$$
$$
\leq -D_\psi(\boldsymbol{\theta}^{t+1},\boldsymbol{\theta}^{t+\frac{1}{2}}) - D_\psi(\boldsymbol{\theta}^{t+\frac{1}{2}},\boldsymbol{\theta}^t) + \eta D L_u\left(D_\psi(\boldsymbol{\theta}^{t+\frac{1}{2}},\boldsymbol{\theta}^t) + D_\psi(\boldsymbol{\theta}^{t+1},\boldsymbol{\theta}^{t+\frac{1}{2}})\right)
$$
$$
\leq -(1 - \eta D L_u)D_\psi\left(D_\psi(\boldsymbol{\theta}^{t+\frac{1}{2}},\boldsymbol{\theta}^t) + D_\psi(\boldsymbol{\theta}^{t+1},\boldsymbol{\theta}^{t+\frac{1}{2}})\right).
$$

where the second inequality is from $\forall a,b \in \mathbb{R}, ab \leq pa^2/2 + b^2/2p, \forall p > 0$ (in this case, $a = \|\boldsymbol{\theta}^{t+\frac{1}{2}} - \boldsymbol{\theta}^t\|_2, b = \|\boldsymbol{\theta}^{t+1} - \boldsymbol{\theta}^{t+\frac{1}{2}}\|_2$, $p = 1$) and $D_\psi(\boldsymbol{a},\boldsymbol{b}) = \|\boldsymbol{a} - \boldsymbol{b}\|_2^2/2, \forall \boldsymbol{a},\boldsymbol{b} \in \mathbb{R}^d$ if $\psi(\cdot)$ is the quadratic regularizer.

## E.2 Proof of Lemma E.2

To prove Lemma E.2, we first introduce the following folk theorem (we drop the terms involved $\boldsymbol{x}$ in Eq. (2) and Eq. (3) since they are not used in the following proofs).

**Theorem E.3.** *The Update rule of SExRM$^+$ can be written as*

$$
\boldsymbol{\theta}^{t+\frac{1}{2}} \in \underset{\boldsymbol{\theta}\in\times_{i\in\mathcal{N}}\mathbb{R}_{\geq 1}^{|A_i|}}{\arg\min}\{\langle -\boldsymbol{F}(\boldsymbol{\theta}^t),\boldsymbol{\theta}\rangle + \frac{1}{\eta}D_\psi(\boldsymbol{\theta},\boldsymbol{\theta}^t)\},
$$
$$
\boldsymbol{\theta}^{t+1} \in \underset{\boldsymbol{\theta}\in\times_{i\in\mathcal{N}}\mathbb{R}_{\geq 1}^{|A_i|}}{\arg\min}\{\langle -\boldsymbol{F}(\boldsymbol{\theta}^{t+\frac{1}{2}}),\boldsymbol{\theta}\rangle + \frac{1}{\eta}D_\psi(\boldsymbol{\theta},\boldsymbol{\theta}^t)\}, \tag{29}
$$

*and the update rule of SPRM$^+$ can be written as*

$$
\boldsymbol{\theta}^{t+\frac{1}{2}} \in \underset{\boldsymbol{\theta}\in\times_{i\in\mathcal{N}}\mathbb{R}_{\geq 1}^{|A_i|}}{\arg\min}\{\langle -\boldsymbol{F}(\boldsymbol{\theta}^{t-\frac{1}{2}}),\boldsymbol{\theta}\rangle + \frac{1}{\eta}D_\psi(\boldsymbol{\theta},\boldsymbol{\theta}^t)\},
$$
$$
\boldsymbol{\theta}^{t+1} \in \underset{\boldsymbol{\theta}\in\times_{i\in\mathcal{N}}\mathbb{R}_{\geq 1}^{|A_i|}}{\arg\min}\{\langle -\boldsymbol{F}(\boldsymbol{\theta}^{t+\frac{1}{2}}),\boldsymbol{\theta}\rangle + \frac{1}{\eta}D_\psi(\boldsymbol{\theta},\boldsymbol{\theta}^t)\}, \tag{30}
$$

*where $\eta > 0$ is the learning rate.*

Considering Eq. (29), and using Lemma G.2 with $\boldsymbol{a} = \boldsymbol{\theta}^t$, $\boldsymbol{a}' = \boldsymbol{\theta}^{t+1}$, $\boldsymbol{a}^* = \boldsymbol{\theta}$ and $\boldsymbol{g} = -\eta \boldsymbol{F}(\boldsymbol{\theta}^{t+\frac{1}{2}})$ (in this case, $\boldsymbol{\mathcal{A}}$ is $\times_{i \in \mathcal{N}} \mathbb{R}_{\geq 1}^{|A_i|}$), we have

$$\eta \langle -\boldsymbol{F}(\boldsymbol{\theta}^{t+\frac{1}{2}}), \boldsymbol{\theta}^{t+1} - \boldsymbol{\theta} \rangle \leq D_\psi(\boldsymbol{\theta}, \boldsymbol{\theta}^t) - D_\psi(\boldsymbol{\theta}, \boldsymbol{\theta}^{t+1}) - D_\psi(\boldsymbol{\theta}^{t+1}, \boldsymbol{\theta}^t). \tag{31}$$

Similarly, with $\boldsymbol{a} = \boldsymbol{\theta}^t$, $\boldsymbol{a}' = \boldsymbol{\theta}^{t+\frac{1}{2}}$, $\boldsymbol{a}^* = \boldsymbol{\theta}^{t+1}$ and $\boldsymbol{g} = -\eta \boldsymbol{F}(\boldsymbol{\theta}^t)$, we get

$$\eta \langle -\boldsymbol{F}(\boldsymbol{\theta}^t), \boldsymbol{\theta}^{t+\frac{1}{2}} - \boldsymbol{\theta}^{t+1} \rangle \leq D_\psi(\boldsymbol{\theta}^{t+1}, \boldsymbol{\theta}^t) - D_\psi(\boldsymbol{\theta}^{t+1}, \boldsymbol{\theta}^{t+\frac{1}{2}}) - D_\psi(\boldsymbol{\theta}^{t+\frac{1}{2}}, \boldsymbol{\theta}^t). \tag{32}$$

Summing up Eq. (31) and (32), and adding $\eta \langle \boldsymbol{F}(\boldsymbol{\theta}^{t+\frac{1}{2}}) - \boldsymbol{F}(\boldsymbol{\theta}^t), \boldsymbol{\theta}^{t+1} - \boldsymbol{\theta}^{t+\frac{1}{2}} \rangle$ to both sides, we get

$$\eta \langle -\boldsymbol{F}(\boldsymbol{\theta}^{t+\frac{1}{2}}), \boldsymbol{\theta}^{t+\frac{1}{2}} - \boldsymbol{\theta} \rangle$$

$$\leq D_\psi(\boldsymbol{\theta}, \boldsymbol{\theta}^t) - D_\psi(\boldsymbol{\theta}, \boldsymbol{\theta}^{t+1}) - D_\psi(\boldsymbol{\theta}^{t+1}, \boldsymbol{\theta}^{t+\frac{1}{2}}) - D_\psi(\boldsymbol{\theta}^{t+\frac{1}{2}}, \boldsymbol{\theta}^t) + \eta \langle \boldsymbol{F}(\boldsymbol{\theta}^{t+\frac{1}{2}}) - \boldsymbol{F}(\boldsymbol{\theta}^t), \boldsymbol{\theta}^{t+1} - \boldsymbol{\theta}^{t+\frac{1}{2}} \rangle.$$

Arranging the terms, we have

$$D_\psi(\boldsymbol{\theta}, \boldsymbol{\theta}^{t+1}) - D_\psi(\boldsymbol{\theta}, \boldsymbol{\theta}^t)$$

$$\leq \eta \langle \boldsymbol{F}(\boldsymbol{\theta}^{t+\frac{1}{2}}), \boldsymbol{\theta}^{t+\frac{1}{2}} - \boldsymbol{\theta} \rangle + \eta \langle \boldsymbol{F}(\boldsymbol{\theta}^{t+\frac{1}{2}}) - \boldsymbol{F}(\boldsymbol{\theta}^t), \boldsymbol{\theta}^{t+1} - \boldsymbol{\theta}^{t+\frac{1}{2}} \rangle - D_\psi(\boldsymbol{\theta}^{t+1}, \boldsymbol{\theta}^{t+\frac{1}{2}}) - D_\psi(\boldsymbol{\theta}^{t+\frac{1}{2}}, \boldsymbol{\theta}^t)$$

$$\leq -\eta \langle \boldsymbol{F}(\boldsymbol{\theta}^{t+\frac{1}{2}}), \boldsymbol{\theta} \rangle + \eta \langle \boldsymbol{F}(\boldsymbol{\theta}^{t+\frac{1}{2}}) - \boldsymbol{F}(\boldsymbol{\theta}^t), \boldsymbol{\theta}^{t+1} - \boldsymbol{\theta}^{t+\frac{1}{2}} \rangle - D_\psi(\boldsymbol{\theta}^{t+1}, \boldsymbol{\theta}^{t+\frac{1}{2}}) - D_\psi(\boldsymbol{\theta}^{t+\frac{1}{2}}, \boldsymbol{\theta}^t),$$

where the last line comes from $\langle \boldsymbol{F}(\boldsymbol{\theta}^{t+\frac{1}{2}}), \boldsymbol{\theta}^{t+\frac{1}{2}} \rangle = \sum_{i \in \mathcal{N}} \langle \boldsymbol{F}_i(\boldsymbol{\theta}^{t+\frac{1}{2}}), \boldsymbol{\theta}_i^{t+\frac{1}{2}} \rangle = \sum_{i \in \mathcal{N}} \langle \langle \boldsymbol{\ell}_i^{t+\frac{1}{2}}, \boldsymbol{x}_i^{t+\frac{1}{2}} \rangle \mathbf{1} - \boldsymbol{\ell}_i^{t+\frac{1}{2}}, \boldsymbol{\theta}_i^{t+\frac{1}{2}} \rangle = \sum_{i \in \mathcal{N}} \langle \boldsymbol{\ell}_i^{t+\frac{1}{2}}, \boldsymbol{x}_i^{t+\frac{1}{2}} \rangle \langle \mathbf{1}, \boldsymbol{\theta}_i^{t+\frac{1}{2}} \rangle - \langle \boldsymbol{\ell}_i^{t+\frac{1}{2}}, \boldsymbol{\theta}_i^{t+\frac{1}{2}} \rangle = \sum_{i \in \mathcal{N}} \langle \boldsymbol{\ell}_i^{t+\frac{1}{2}}, \frac{\boldsymbol{\theta}_i^{t+\frac{1}{2}}}{\|\boldsymbol{\theta}_i^{t+\frac{1}{2}}\|_1} \rangle \|\boldsymbol{\theta}_i^{t+\frac{1}{2}}\|_1 - \langle \boldsymbol{\ell}_i^{t+\frac{1}{2}}, \boldsymbol{\theta}_i^{t+\frac{1}{2}} \rangle = 0$. It completes the proof.

## F  Proof of Theorem 5.3

**Lemma F.1.** *(Proof is in Appendix F.1) Let $\boldsymbol{x}^* \in \mathcal{X}^*$ that satisfies the MVI and $0 < \eta < \frac{1}{8DL_u}$, then for every iteration $t \geq 1$, it holds that*

$$\|\boldsymbol{\theta}^{t+1} - \boldsymbol{x}^*\|_2^2 + \frac{1}{16}\|\boldsymbol{\theta}^{t+1} - \boldsymbol{\theta}^{t+\frac{1}{2}}\|_2^2 \leq \|\boldsymbol{\theta}^t - \boldsymbol{x}^*\|_2^2 + \frac{1}{16}\|\boldsymbol{\theta}^t - \boldsymbol{\theta}^{t-\frac{1}{2}}\|_2^2 - \frac{15}{16}(\|\boldsymbol{\theta}^{t+1} - \boldsymbol{\theta}^{t+\frac{1}{2}}\|_2^2 + \|\boldsymbol{\theta}^t - \boldsymbol{\theta}^{t+\frac{1}{2}}\|_2^2).$$

From Lemma F.1, we have

$$\|\boldsymbol{\theta}^{t+1} - \boldsymbol{x}^*\|_2^2 + \frac{1}{16}\|\boldsymbol{\theta}^{t+1} - \boldsymbol{\theta}^{t+\frac{1}{2}}\|_2^2 \leq \|\boldsymbol{\theta}^t - \boldsymbol{\theta}^*\|_2^2 +$$

$$\frac{1}{16}\|\boldsymbol{\theta}^t - \boldsymbol{\theta}^{t-\frac{1}{2}}\|_2^2 - \frac{15}{16}(\|\boldsymbol{\theta}^{t+1} - \boldsymbol{\theta}^{t+\frac{1}{2}}\|_2^2 + \|\boldsymbol{\theta}^t - \boldsymbol{\theta}^{t+\frac{1}{2}}\|_2^2). \tag{33}$$

Assume $\|\boldsymbol{\theta}^{t+1} - \boldsymbol{\theta}^{t+\frac{1}{2}}\|_2^2 + \|\boldsymbol{\theta}^t - \boldsymbol{\theta}^{t+\frac{1}{2}}\|_2^2$ do not converge to 0. Then, from Eq. (33), we have

$$\|\boldsymbol{\theta}^{T+1} - \boldsymbol{x}^*\|_2^2 + \frac{1}{16}\|\boldsymbol{\theta}^{T+1} - \boldsymbol{\theta}^{T+\frac{1}{2}}\|_2^2$$

$$\leq \|\boldsymbol{\theta}^1 - \boldsymbol{x}^*\|_2^2 + \|\frac{1}{16}\boldsymbol{\theta}^1 - \boldsymbol{\theta}^{1-\frac{1}{2}}\|_2^2 - \frac{15}{16}\sum_{t=1}^T (\|\boldsymbol{\theta}^{t+1} - \boldsymbol{\theta}^{t+\frac{1}{2}}\|_2^2 + \|\boldsymbol{\theta}^t - \boldsymbol{\theta}^{t+\frac{1}{2}}\|_2^2).$$

Therefore, as $T \to \infty$, $\|\boldsymbol{\theta}^{T+1} - \boldsymbol{x}^*\|_2^2 + \frac{1}{16}\|\boldsymbol{\theta}^{T+1} - \boldsymbol{\theta}^{T+\frac{1}{2}}\|_2^2 \leq \|\boldsymbol{\theta}^1 - \boldsymbol{\theta}^*\|_2^2 + \|\frac{1}{16}\boldsymbol{\theta}^1 - \boldsymbol{\theta}^{1-\frac{1}{2}}\|_2^2 - \frac{15}{16}\sum_{t=1}^T(\|\boldsymbol{\theta}^{t+1} - \boldsymbol{\theta}^{t+\frac{1}{2}}\|_2^2 + \|\boldsymbol{\theta}^t - \boldsymbol{\theta}^{t+\frac{1}{2}}\|_2^2) = -\infty$, which contracts that $\|\boldsymbol{\theta}^{T+1} - \boldsymbol{x}^*\|_2^2 + \frac{1}{16}\|\boldsymbol{\theta}^{T+1} - \boldsymbol{\theta}^{T+\frac{1}{2}}\|_2^2 \geq 0$. Therefore, we have $\|\boldsymbol{\theta}^{t+1} - \boldsymbol{\theta}^{t+\frac{1}{2}}\|_2^2 + \|\boldsymbol{\theta}^t - \boldsymbol{\theta}^{t+\frac{1}{2}}\|_2^2 \to 0$ as $t \to \infty$, which implies $\|\boldsymbol{\theta}^{t+1} - \boldsymbol{\theta}^{t+\frac{1}{2}}\|_2^2 \to 0$ and $\|\boldsymbol{\theta}^t - \boldsymbol{\theta}^{t+\frac{1}{2}}\|_2^2 \to 0$ as $t \to \infty$. It completes the proof.

In addition, from $\eta < \frac{1}{8DL_u}$ and Eq. (33), we have

$$\sum_{t=1}^T \left( \|\boldsymbol{\theta}^{t+\frac{1}{2}} - \boldsymbol{\theta}^t\|_2^2 + \|\boldsymbol{\theta}^{t+1} - \boldsymbol{\theta}^{t+\frac{1}{2}}\|_2^2 \right) \leq C,$$

where $C$ is a constant which depends on $\boldsymbol{\theta}^1$, $\boldsymbol{\theta}^{\frac{1}{2}}$, $\boldsymbol{x}^*$, $\eta$, $D$, and $L_u$. Therefore, we get

$$T \min_{t \in T} \left( \|\boldsymbol{\theta}^{t+\frac{1}{2}} - \boldsymbol{\theta}^t\|_2^2 + \|\boldsymbol{\theta}^{t+1} - \boldsymbol{\theta}^{t+\frac{1}{2}}\|_2^2 \right) \leq \sum_{t=1}^T \left( \|\boldsymbol{\theta}^{t+\frac{1}{2}} - \boldsymbol{\theta}^t\|_2^2 + \|\boldsymbol{\theta}^{t+1} - \boldsymbol{\theta}^{t+\frac{1}{2}}\|_2^2 \right) \leq C,$$

which implies

$$\min_{t \in T} \left( \|\boldsymbol{\theta}^{t+\frac{1}{2}} - \boldsymbol{\theta}^t\|_2^2 + \|\boldsymbol{\theta}^{t+1} - \boldsymbol{\theta}^{t+\frac{1}{2}}\|_2^2 \right) \leq \frac{C}{T}.$$

## F.1 Proof of Lemma F.1

**Lemma F.2.** *(Proof is in Appendix F.2) Assume all players follow the update rule of SPRM$^+$, then for any $\boldsymbol{\theta} \in \mathbb{R}_{\geq 1}^{|\boldsymbol{\mathcal{X}}|}$, we have*

$$D_\psi(\boldsymbol{\theta}, \boldsymbol{\theta}^{t+1}) - D_\psi(\boldsymbol{\theta}, \boldsymbol{\theta}^t)$$

$$\leq -\eta \langle \boldsymbol{F}(\boldsymbol{\theta}^{t+\frac{1}{2}}), \boldsymbol{\theta} \rangle + \eta \langle \boldsymbol{F}(\boldsymbol{\theta}^{t+\frac{1}{2}}) - \boldsymbol{F}(\boldsymbol{\theta}^{t-\frac{1}{2}}), \boldsymbol{\theta}^{t+1} - \boldsymbol{\theta}^{t+\frac{1}{2}} \rangle - D_\psi(\boldsymbol{\theta}^{t+1}, \boldsymbol{\theta}^{t+\frac{1}{2}}) - D_\psi(\boldsymbol{\theta}^{t+\frac{1}{2}}, \boldsymbol{\theta}^t).$$

Substituting $\boldsymbol{\theta} = \boldsymbol{x}^* \in \boldsymbol{\mathcal{X}}^*$ that satisfies the MVI into Lemma F.2, we get

$$D_\psi(\boldsymbol{x}^*, \boldsymbol{\theta}^{t+1}) - D_\psi(\boldsymbol{x}^*, \boldsymbol{\theta}^t)$$

$$\leq -\eta \langle \boldsymbol{F}(\boldsymbol{\theta}^{t+\frac{1}{2}}), \boldsymbol{x}^* \rangle - D_\psi(\boldsymbol{\theta}^{t+1}, \boldsymbol{\theta}^{t+\frac{1}{2}}) - D_\psi(\boldsymbol{\theta}^{t+\frac{1}{2}}, \boldsymbol{\theta}^t) + \eta \langle \boldsymbol{F}(\boldsymbol{\theta}^{t+\frac{1}{2}}) - \boldsymbol{F}(\boldsymbol{\theta}^{t-\frac{1}{2}}), \boldsymbol{\theta}^{t+1} - \boldsymbol{\theta}^{t+\frac{1}{2}} \rangle. \tag{34}$$

According to the analysis in Section 5, for the first term of the right-hand side of Eq. (34), from Eq. (27), we have

$$-\eta \sum_{i \in \mathcal{N}} \langle \boldsymbol{F}_i(\boldsymbol{\theta}^{t+\frac{1}{2}}), \boldsymbol{x}_i^* \rangle \leq 0.$$

To simply the fourth term of the right-hand side of Eq. (34), we first introduce Lemma F.3, whose proof is in Appendix F.3.

**Lemma F.3.** *Assume all players follow the update rule of SPRM$^+$, then we have*

$$\|\boldsymbol{\theta}^{t+1} - \boldsymbol{\theta}^{t+\frac{1}{2}}\|_2 \leq \eta \|\boldsymbol{F}(\boldsymbol{\theta}^{t+\frac{1}{2}}) - \boldsymbol{F}(\boldsymbol{\theta}^{t-\frac{1}{2}})\|_2.$$

Therefore, for the fourth term of the right-hand side of Eq. (34), we have

$$\eta \langle \boldsymbol{F}(\boldsymbol{\theta}^{t+\frac{1}{2}}) - \boldsymbol{F}(\boldsymbol{\theta}^{t-\frac{1}{2}}), \boldsymbol{\theta}^{t+1} - \boldsymbol{\theta}^{t+\frac{1}{2}} \rangle$$

$$\leq \eta \|\boldsymbol{F}(\boldsymbol{\theta}^{t+\frac{1}{2}}) - \boldsymbol{F}(\boldsymbol{\theta}^{t-\frac{1}{2}})\|_2 \|\boldsymbol{\theta}^{t+1} - \boldsymbol{\theta}^{t+\frac{1}{2}}\|_2$$

$$\leq \eta^2 \|\boldsymbol{F}(\boldsymbol{\theta}^{t+\frac{1}{2}}) - \boldsymbol{F}(\boldsymbol{\theta}^{t-\frac{1}{2}})\|_2^2.$$

where the third line is from Lemma F.3. Then, from Lemma 5.2 of Farina et al. [2023] ($\|\boldsymbol{F}(\boldsymbol{\theta}) - \boldsymbol{F}(\boldsymbol{\theta}')\|_2 \leq DL_u \|\boldsymbol{\theta} - \boldsymbol{\theta}'\|_2, \forall \boldsymbol{\theta}, \boldsymbol{\theta}' \in \mathbb{R}_{\geq 1}^{|\boldsymbol{\mathcal{X}}|}$, where $D, L_u$ are defined in Theorem 5.1) and the choice of $\eta$, we have

$$\eta^2 \|\boldsymbol{F}(\boldsymbol{\theta}^{t+\frac{1}{2}}) - \boldsymbol{F}(\boldsymbol{\theta}^{t-\frac{1}{2}})\|_2^2 \leq \eta^2 D^2 L_u^2 \|\boldsymbol{\theta}^{t+\frac{1}{2}} - \boldsymbol{\theta}^{t-\frac{1}{2}}\|_2^2 \leq \frac{1}{64} \|\boldsymbol{\theta}^{t+\frac{1}{2}} - \boldsymbol{\theta}^{t-\frac{1}{2}}\|_2^2.$$

Continuing from Eq. (34), we then have

$$D_\psi(\boldsymbol{x}^*, \boldsymbol{\theta}^{t+1}) - D_\psi(\boldsymbol{x}^*, \boldsymbol{\theta}^t)$$

$$\leq -D_\psi(\boldsymbol{\theta}^{t+1}, \boldsymbol{\theta}^{t+\frac{1}{2}}) - D_\psi(\boldsymbol{\theta}^{t+\frac{1}{2}}, \boldsymbol{\theta}^t) + \frac{1}{64} \|\boldsymbol{\theta}^{t+\frac{1}{2}} - \boldsymbol{\theta}^{t-\frac{1}{2}}\|_2^2$$

$$\leq -D_\psi(\boldsymbol{\theta}^{t+1}, \boldsymbol{\theta}^{t+\frac{1}{2}}) - D_\psi(\boldsymbol{\theta}^{t+\frac{1}{2}}, \boldsymbol{\theta}^t) + \frac{1}{32} \|\boldsymbol{\theta}^{t+\frac{1}{2}} - \boldsymbol{\theta}^t\|_2^2 + \frac{1}{32} \|\boldsymbol{\theta}^t - \boldsymbol{\theta}^{t-\frac{1}{2}}\|_2^2$$

$$\Leftrightarrow \|\boldsymbol{\theta}^{t+1} - \boldsymbol{x}^*\|_2^2 + \frac{1}{16} \|\boldsymbol{\theta}^{t+1} - \boldsymbol{\theta}^{t+\frac{1}{2}}\|_2^2$$

$$\leq \|\boldsymbol{\theta}^t - \boldsymbol{x}^*\|_2^2 + \frac{1}{16} \|\boldsymbol{\theta}^t - \boldsymbol{\theta}^{t-\frac{1}{2}}\|_2^2 - \frac{15}{16} (\|\boldsymbol{\theta}^{t+1} - \boldsymbol{\theta}^{t+\frac{1}{2}}\|_2^2 + \|\boldsymbol{\theta}^t - \boldsymbol{\theta}^{t+\frac{1}{2}}\|_2^2),$$

where the last line is from $D_\psi(\boldsymbol{a}, \boldsymbol{b}) = \|\boldsymbol{a} - \boldsymbol{b}\|_2^2 / 2$.

## F.2 Proof of Lemma F.2

Considering Eq. (30), and using Lemma G.2 with $\boldsymbol{a} = \boldsymbol{\theta}^t$, $\boldsymbol{a}' = \boldsymbol{\theta}^{t+1}$, $\boldsymbol{a}^* = \boldsymbol{\theta}$ and $\boldsymbol{g} = -\eta \boldsymbol{F}(\boldsymbol{\theta}^{t+\frac{1}{2}})$ (in this case, $\boldsymbol{\mathcal{A}}$ is $\times_{i \in \mathcal{N}} \mathbb{R}_{\geq 1}^{|A_i|}$), we have

$$\eta \langle -\boldsymbol{F}(\boldsymbol{\theta}^{t+\frac{1}{2}}), \boldsymbol{\theta}^{t+1} - \boldsymbol{\theta} \rangle \leq D_\psi(\boldsymbol{\theta}, \boldsymbol{\theta}^t) - D_\psi(\boldsymbol{\theta}, \boldsymbol{\theta}^{t+1}) - D_\psi(\boldsymbol{\theta}^{t+1}, \boldsymbol{\theta}^t)$$

$$\Leftrightarrow \eta \langle -\boldsymbol{F}(\boldsymbol{\theta}^{t+\frac{1}{2}}), \boldsymbol{\theta}^{t+1} - \boldsymbol{\theta} \rangle \leq D_\psi(\boldsymbol{\theta}, \boldsymbol{\theta}^t) - D_\psi(\boldsymbol{\theta}, \boldsymbol{\theta}^{t+1}) - D_\psi(\boldsymbol{\theta}^{t+1}, \boldsymbol{\theta}^t). \tag{35}$$

Similarly, with $\boldsymbol{a} = \boldsymbol{\theta}^t$, $\boldsymbol{a}' = \boldsymbol{\theta}^{t+\frac{1}{2}}$, $\boldsymbol{a}^* = \boldsymbol{\theta}^{t+1}$ and $\boldsymbol{g} = -\eta\boldsymbol{F}(\boldsymbol{\theta}^{t-\frac{1}{2}})$, we get

$$\eta\langle -\boldsymbol{F}(\boldsymbol{\theta}^{t-\frac{1}{2}}), \boldsymbol{\theta}^{t+\frac{1}{2}} - \boldsymbol{\theta}^{t+1}\rangle \leq D_\psi(\boldsymbol{\theta}^{t+1}, \boldsymbol{\theta}^t) - D_\psi(\boldsymbol{\theta}^{t+1}, \boldsymbol{\theta}^{t+\frac{1}{2}}) - D_\psi(\boldsymbol{\theta}^{t+\frac{1}{2}}, \boldsymbol{\theta}^t)$$
$$\Leftrightarrow \eta\langle -\boldsymbol{F}(\boldsymbol{\theta}^{t-\frac{1}{2}}), \boldsymbol{\theta}^{t+\frac{1}{2}} - \boldsymbol{\theta}^{t+1}\rangle \leq D_\psi(\boldsymbol{\theta}^{t+1}, \boldsymbol{\theta}^t) - D_\psi(\boldsymbol{\theta}^{t+1}, \boldsymbol{\theta}^{t+\frac{1}{2}}) - D_\psi(\boldsymbol{\theta}^{t+\frac{1}{2}}, \boldsymbol{\theta}^t). \quad (36)$$

Summing up Eq. (35) and (36), and adding $\eta\langle \boldsymbol{F}(\boldsymbol{\theta}^{t+\frac{1}{2}}) - \boldsymbol{F}(\boldsymbol{\theta}^{t-\frac{1}{2}}), \boldsymbol{\theta}^{t+1} - \boldsymbol{\theta}^{t+\frac{1}{2}}\rangle$ to both sides, we get

$$\eta\langle -\boldsymbol{F}(\boldsymbol{\theta}^{t+\frac{1}{2}}), \boldsymbol{\theta}^{t+\frac{1}{2}} - \boldsymbol{\theta}\rangle$$
$$\leq D_\psi(\boldsymbol{\theta}, \boldsymbol{\theta}^t) - D_\psi(\boldsymbol{\theta}, \boldsymbol{\theta}^{t+1}) - D_\psi(\boldsymbol{\theta}^{t+1}, \boldsymbol{\theta}^{t+\frac{1}{2}}) - D_\psi(\boldsymbol{\theta}^{t+\frac{1}{2}}, \boldsymbol{\theta}^t) + \eta\langle \boldsymbol{F}(\boldsymbol{\theta}^{t+\frac{1}{2}}) - \boldsymbol{F}(\boldsymbol{\theta}^{t-\frac{1}{2}}), \boldsymbol{\theta}^{t+1} - \boldsymbol{\theta}^{t+\frac{1}{2}}\rangle.$$

Arranging the terms, we have

$$D_\psi(\boldsymbol{\theta}, \boldsymbol{\theta}^{t+1}) - D_\psi(\boldsymbol{\theta}, \boldsymbol{\theta}^t)$$
$$\leq \eta\langle \boldsymbol{F}(\boldsymbol{\theta}^{t+\frac{1}{2}}), \boldsymbol{\theta}^{t+\frac{1}{2}} - \boldsymbol{\theta}\rangle + \eta\langle \boldsymbol{F}(\boldsymbol{\theta}^{t+\frac{1}{2}}) - \boldsymbol{F}(\boldsymbol{\theta}^{t-\frac{1}{2}}), \boldsymbol{\theta}^{t+1} - \boldsymbol{\theta}^{t+\frac{1}{2}}\rangle - D_\psi(\boldsymbol{\theta}^{t+1}, \boldsymbol{\theta}^{t+\frac{1}{2}}) - D_\psi(\boldsymbol{\theta}^{t+\frac{1}{2}}, \boldsymbol{\theta}^t)$$
$$\leq -\eta\langle \boldsymbol{F}(\boldsymbol{\theta}^{t+\frac{1}{2}}), \boldsymbol{\theta}\rangle + \eta\langle \boldsymbol{F}(\boldsymbol{\theta}^{t+\frac{1}{2}}) - \boldsymbol{F}(\boldsymbol{\theta}^{t-\frac{1}{2}}), \boldsymbol{\theta}^{t+1} - \boldsymbol{\theta}^{t+\frac{1}{2}}\rangle - D_\psi(\boldsymbol{\theta}^{t+1}, \boldsymbol{\theta}^{t+\frac{1}{2}}) - D_\psi(\boldsymbol{\theta}^{t+\frac{1}{2}}, \boldsymbol{\theta}^t),$$

where the last line comes from $\langle \boldsymbol{F}(\boldsymbol{\theta}^{t+\frac{1}{2}}), \boldsymbol{\theta}^{t+\frac{1}{2}}\rangle = \sum_{i\in\mathcal{N}}\langle \boldsymbol{F}_i(\boldsymbol{\theta}^{t+\frac{1}{2}}), \boldsymbol{\theta}_i^{t+\frac{1}{2}}\rangle = \sum_{i\in\mathcal{N}}\langle\langle \boldsymbol{\ell}_i^{t+\frac{1}{2}}, \boldsymbol{x}_i^{t+\frac{1}{2}}\rangle\mathbf{1} - \boldsymbol{\ell}_i^{t+\frac{1}{2}}, \boldsymbol{\theta}_i^{t+\frac{1}{2}}\rangle = \sum_{i\in\mathcal{N}}\langle \boldsymbol{\ell}_i^{t+\frac{1}{2}}, \boldsymbol{x}_i^{t+\frac{1}{2}}\rangle\langle \mathbf{1}, \boldsymbol{\theta}_i^{t+\frac{1}{2}}\rangle - \langle \boldsymbol{\ell}_i^{t+\frac{1}{2}}, \boldsymbol{\theta}_i^{t+\frac{1}{2}}\rangle = \sum_{i\in\mathcal{N}}\langle \boldsymbol{\ell}_i^{t+\frac{1}{2}}, \frac{\boldsymbol{\theta}_i^{t+\frac{1}{2}}}{\|\boldsymbol{\theta}_i^{t+\frac{1}{2}}\|_1}\rangle\|\boldsymbol{\theta}_i^{t+\frac{1}{2}}\|_1 - \langle \boldsymbol{\ell}_i^{t+\frac{1}{2}}, \boldsymbol{\theta}_i^{t+\frac{1}{2}}\rangle = 0$. It completes the proof.

## F.3 Proof of Lemma F.3

To prove Lemma F.3, we first introduce Lemma F.4, which is Lemma 11 of Wei et al. [2021]

**Lemma F.4.** *Suppose that $\varphi(\cdot)$ satisfies $D_\varphi(\boldsymbol{b}, \boldsymbol{b}') \geq \frac{1}{2}\|\boldsymbol{b} - \boldsymbol{b}'\|_p^2$ for some $p \geq 1$, and let $\boldsymbol{a}, \boldsymbol{a}_1, \boldsymbol{a}_2 \in \mathcal{A}$ (a convex set) be related by the following:*

$$\boldsymbol{a}_1 \in \underset{\boldsymbol{a}'\in\mathcal{A}}{\arg\min}\{\langle \boldsymbol{a}', \boldsymbol{g}_1\rangle + D_\varphi(\boldsymbol{a}', \boldsymbol{a})\},$$
$$\boldsymbol{a}_2 \in \underset{\boldsymbol{a}'\in\mathcal{A}}{\arg\min}\{\langle \boldsymbol{a}', \boldsymbol{g}_2\rangle + D_\varphi(\boldsymbol{a}', \boldsymbol{a})\}.$$

*Then, we have*

$$\|\boldsymbol{a}_1 - \boldsymbol{a}_2\|_p \leq \|\boldsymbol{g}_1 - \boldsymbol{g}_2\|_q,$$

*where $q \geq 1$ and $\frac{1}{p} + \frac{1}{q} = 1$.*

Considering Eq. (30) and substituting $\boldsymbol{a}_1 = \boldsymbol{\theta}^{t+1}$, $\boldsymbol{a}_2 = \boldsymbol{\theta}^{t+\frac{1}{2}}$, $\boldsymbol{g}_1 = -\eta\boldsymbol{F}(\boldsymbol{\theta}^{t+\frac{1}{2}})$, $\boldsymbol{g}_2 = -\eta\boldsymbol{F}(\boldsymbol{\theta}^{t-\frac{1}{2}})$ and $\varphi(\cdot) = \psi(\cdot)$ ($\psi(\cdot)$ is the quadratic regularizer, which satisfies $D_\psi(\boldsymbol{b}, \boldsymbol{b}') \geq \frac{1}{2}\|\boldsymbol{b} - \boldsymbol{b}'\|_2^2$) into Lemma F.4 (in this case, $\mathcal{A}$ is $\times_{i\in\mathcal{N}}\mathbb{R}_{\geq 1}^{|A_i|}$), we have

$$\|\boldsymbol{\theta}^{t+1} - \boldsymbol{\theta}^{t+\frac{1}{2}}\|_2 \leq \|\eta\boldsymbol{F}(\boldsymbol{\theta}^{t+\frac{1}{2}}) - \eta\boldsymbol{F}(\boldsymbol{\theta}^{t-\frac{1}{2}})\|_2$$
$$\Leftrightarrow \|\boldsymbol{\theta}^{t+1} - \boldsymbol{\theta}^{t+\frac{1}{2}}\|_2 \leq \eta\|\boldsymbol{F}(\boldsymbol{\theta}^{t+\frac{1}{2}}) - \boldsymbol{F}(\boldsymbol{\theta}^{t-\frac{1}{2}})\|_2,$$

which completes the proof.

## G Proof of Theorem 6.2

**Lemma G.1.** *Let $\boldsymbol{x}^* \in \mathcal{X}^*$ that satisfies the weak MVI and $0 < \eta < \frac{1}{2DL_u}$, then for every iteration $t \geq 1$, it holds that*

$$\sum_{t=1}^{T}\left(\frac{1}{2} + \frac{2\rho}{\eta} - 2\eta^2 D^2 L_u^2\right)\|\boldsymbol{\theta}^{t+1} - \boldsymbol{\theta}^t\|_2^2 \leq \|\boldsymbol{\theta}^1 - \boldsymbol{x}^*\|_2^2 + \frac{1}{4}\|\boldsymbol{\theta}^{\frac{1}{2}} - \boldsymbol{\theta}^{-\frac{1}{2}}\|_2^2.$$

*where $D = \max_{i\in\mathcal{N}}|A_i|$ and $L_u = \sqrt{2P^2 + 4L^2}$.*

From Lemma G.1, we have

$$\sum_{t=1}^{T} \left( \frac{1}{2} + \frac{2\rho}{\eta} - 2\eta^2 D^2 L_u^2 \right) \|\boldsymbol{\theta}^{t+1} - \boldsymbol{\theta}^t\|_2^2 \leq \|\boldsymbol{\theta}^1 - \boldsymbol{x}^*\|_2^2 + \frac{1}{4} \|\boldsymbol{\theta}^{\frac{1}{2}} - \boldsymbol{\theta}^{-\frac{1}{2}}\|_2^2. \qquad (37)$$

Now, we first prove that if $\rho > -\frac{1}{12\sqrt{3}DL_u}$, there always exists $0 < \eta < \frac{1}{2DL_u}$ that ensures $\frac{1}{2} + \frac{2\rho}{\eta} - 2\eta^2 D^2 L_u^2 > 0$. Formally, consider this case where $\rho = -\frac{1}{12\sqrt{3}DL_u}$, we can set $\eta = 1/(2\sqrt{3}DL_u)$ that ensures

$$\frac{1}{2} + \frac{2\rho}{\eta} - 2\eta^2 D^2 L_u^2 = \frac{1}{2} - \frac{1}{3} - \frac{1}{6} = 0.$$

It is evident that the expression $\frac{1}{2} + \frac{2\rho}{\eta} - 2\eta^2 D^2 L_u^2$ increases with $\rho$ if $\eta > 0$. We have shown that when $\rho = -\frac{1}{12\sqrt{3}DL_u}$ and $\eta = \frac{1}{2\sqrt{3}DL_u}$, the expression evaluates to zero

$$\frac{1}{2} + \frac{2\rho}{\eta} - 2\eta^2 D^2 L_u^2 = \frac{1}{2} - \frac{1}{3} - \frac{1}{6} = 0.$$

Therefore, if $\rho > -\frac{1}{12\sqrt{3}DL_u}$, with $\eta = \frac{1}{2\sqrt{3}DL_u}$, it must hold that

$$\frac{1}{2} + \frac{2\rho}{\eta} - 2\eta^2 D^2 L_u^2 > 0.$$

In other words, if $\rho > -\frac{1}{12\sqrt{3}DL_u}$, then there always exists some $\eta = \frac{1}{2\sqrt{3}DL_u} \in \left( 0, \frac{1}{2DL_u} \right)$ such that

$$\frac{1}{2} + \frac{2\rho}{\eta} - 2\eta^2 D^2 L_u^2 > 0.$$

Assume $\|\boldsymbol{\theta}^{t+1} - \boldsymbol{\theta}^t\|_2^2$ do not converge to $0$. Then, from Eq. (37), we have

$$\sum_{t=1}^{T} \left( \frac{1}{2} + \frac{2\rho}{\eta} - 2\eta^2 D^2 L_u^2 \right) \|\boldsymbol{\theta}^{t+1} - \boldsymbol{\theta}^t\|_2^2 \geq O(T),$$

which contracts that

$$\sum_{t=1}^{T} \left( \frac{1}{2} + \frac{2\rho}{\eta} - 2\eta^2 D^2 L_u^2 \right) \|\boldsymbol{\theta}^{t+1} - \boldsymbol{\theta}^t\|_2^2 \leq \|\boldsymbol{\theta}^1 - \boldsymbol{x}^*\|_2^2 + \frac{1}{4} \|\boldsymbol{\theta}^{\frac{1}{2}} - \boldsymbol{\theta}^{-\frac{1}{2}}\|_2^2.$$

Therefore, $\|\boldsymbol{\theta}^{t+1} - \boldsymbol{\theta}^t\|_2^2 \to 0$.

In addition, from $\frac{1}{2} + \frac{2\rho}{\eta} - 2\eta^2 D^2 L_u^2 > 0$ and Eq. (37), we have

$$\sum_{t=1}^{T} \|\boldsymbol{\theta}^{t+1} - \boldsymbol{\theta}^t\|_2^2 \leq \frac{\|\boldsymbol{\theta}^1 - \boldsymbol{x}^*\|_2^2 + \frac{1}{4}\|\boldsymbol{\theta}^{\frac{1}{2}} - \boldsymbol{\theta}^{-\frac{1}{2}}\|_2^2}{\left( \frac{1}{2} + \frac{2\rho}{\eta} - 2\eta^2 D^2 L_u^2 \right)} = C.$$

Since $\boldsymbol{\theta}^1, \boldsymbol{\theta}^{\frac{1}{2}}, \boldsymbol{\theta}^{-\frac{1}{2}}, \boldsymbol{x}^*, \eta, \rho, D$, and $L_u$ is fixed, $C$ must be a constant. Therefore, we get

$$T \min_{t \in T} \|\boldsymbol{\theta}^{t+1} - \boldsymbol{\theta}^t\|_2^2 \leq \sum_{t=1}^{T} \|\boldsymbol{\theta}^{t+1} - \boldsymbol{\theta}^t\|_2^2 \leq C,$$

which implies

$$\min_{t \in T} \|\boldsymbol{\theta}^{t+1} - \boldsymbol{\theta}^t\|_2^2 \leq \frac{C}{T}.$$

## G.1 Proof of Lemma G.1

**Lemma G.2.** *(Lemma 10 of Wei et al. [2021]) Let $\mathcal{A}$ as a convex set and $\boldsymbol{a}' \in \arg\min_{\boldsymbol{a}' \in \mathcal{A}}\{\langle \boldsymbol{a}', \boldsymbol{g}\rangle + D_\psi(\boldsymbol{a}', \boldsymbol{a})\}$. Then for any $\boldsymbol{a}^* \in \mathcal{A}$,*

$$\langle \boldsymbol{a}' - \boldsymbol{a}^*, \boldsymbol{g}\rangle \leq D_\psi(\boldsymbol{a}^*, \boldsymbol{a}) - D_\psi(\boldsymbol{a}^*, \boldsymbol{a}') - D_\psi(\boldsymbol{a}', \boldsymbol{a}).$$

**Lemma G.3.** *(Adapted from Lemma A.2 of Hsieh et al. [2019]) Assume all players follow the update rule of SOGRM$^+$, then for any $\boldsymbol{\theta}_i \in \mathbb{R}^{|\mathcal{X}_i|}_{\geq 1}$, we have*

$$D_\psi(\boldsymbol{\theta}_i, \boldsymbol{\theta}_i^{t+1}) - D_\psi(\boldsymbol{\theta}_i, \boldsymbol{\theta}_i^t)$$
$$\leq \langle \boldsymbol{\theta}_i^t - \boldsymbol{\theta}_i^{t+\frac{1}{2}} + \eta \boldsymbol{F}_i(\boldsymbol{\theta}^{t-\frac{1}{2}}) - \eta \boldsymbol{F}_i(\boldsymbol{\theta}^{t+\frac{1}{2}}), \boldsymbol{\theta}_i - \boldsymbol{\theta}_i^{t+\frac{1}{2}}\rangle + D_\psi(\boldsymbol{\theta}_i^{t+1}, \boldsymbol{\theta}_i^{t+\frac{1}{2}}) - D_\psi(\boldsymbol{\theta}_i^{t+\frac{1}{2}}, \boldsymbol{\theta}_i^t).$$

Considering Eq. (7) and using Lemma G.2 with $\boldsymbol{a} = \boldsymbol{\theta}_i^t$, $\boldsymbol{a}' = \boldsymbol{\theta}_i^{t+\frac{1}{2}}$, $\boldsymbol{a}^* = \boldsymbol{x}_i^{t+\frac{1}{2}}$, and $\boldsymbol{g} = -\eta \boldsymbol{F}_i(\boldsymbol{\theta}^{t-\frac{1}{2}})$, we have

$$0 \leq \langle \eta \boldsymbol{F}_i(\boldsymbol{\theta}^{t-\frac{1}{2}}), \boldsymbol{\theta}_i^{t+\frac{1}{2}} - \boldsymbol{x}_i^{t+\frac{1}{2}}\rangle + D_\psi(\boldsymbol{x}_i^{t+\frac{1}{2}}, \boldsymbol{\theta}_i^t) - D_\psi(\boldsymbol{x}_i^{t+\frac{1}{2}}, \boldsymbol{\theta}_i^{t+\frac{1}{2}}) - D_\psi(\boldsymbol{\theta}_i^{t+\frac{1}{2}}, \boldsymbol{\theta}_i^t)$$
$$\Leftrightarrow 0 \leq \langle \eta \boldsymbol{F}_i(\boldsymbol{\theta}^{t-\frac{1}{2}}), \boldsymbol{\theta}_i^{t+\frac{1}{2}} - \boldsymbol{x}_i^{t+\frac{1}{2}}\rangle + \langle \boldsymbol{\theta}_i^t - \boldsymbol{\theta}_i^{t+\frac{1}{2}}, \boldsymbol{\theta}_i^{t+\frac{1}{2}} - \boldsymbol{x}_i^{t+\frac{1}{2}}\rangle, \tag{38}$$

where the second line comes from

$$D_\psi(\boldsymbol{x}_i^{t+\frac{1}{2}}, \boldsymbol{\theta}_i^t) - D_\psi(\boldsymbol{x}_i^{t+\frac{1}{2}}, \boldsymbol{\theta}_i^{t+\frac{1}{2}}) - D_\psi(\boldsymbol{\theta}_i^{t+\frac{1}{2}}, \boldsymbol{\theta}_i^t)$$
$$= \frac{\|\boldsymbol{x}_i^{t+\frac{1}{2}}\|_2^2}{2} - \langle \boldsymbol{x}_i^{t+\frac{1}{2}}, \boldsymbol{\theta}_i^t\rangle + \frac{\|\boldsymbol{\theta}_i^t\|_2^2}{2} - \frac{\|\boldsymbol{x}_i^{t+\frac{1}{2}}\|_2^2}{2} + \langle \boldsymbol{x}_i^{t+\frac{1}{2}}, \boldsymbol{\theta}_i^{t+\frac{1}{2}}\rangle - \frac{\|\boldsymbol{\theta}_i^{t+\frac{1}{2}}\|_2^2}{2} - \frac{\|\boldsymbol{\theta}_i^{t+\frac{1}{2}}\|_2^2}{2} + \langle \boldsymbol{\theta}_i^{t+\frac{1}{2}}, \boldsymbol{\theta}_i^t\rangle - \frac{\|\boldsymbol{\theta}_i^t\|_2^2}{2}$$
$$= \langle \boldsymbol{\theta}_i^t - \boldsymbol{\theta}_i^{t+\frac{1}{2}}, \boldsymbol{\theta}_i^{t+\frac{1}{2}} - \boldsymbol{x}_i^{t+\frac{1}{2}}\rangle.$$

Substituting $\boldsymbol{\theta}_i = \boldsymbol{x}_i^* \in \mathcal{X}^*$ that satisfies the weak MVI and Eq. (38) into Lemma G.3, and using the fact that $\langle \boldsymbol{F}_i(\boldsymbol{\theta}^{t+\frac{1}{2}}), \boldsymbol{\theta}_i^{t+\frac{1}{2}}\rangle = \langle \langle \boldsymbol{\ell}_i^{t+\frac{1}{2}}, \boldsymbol{x}_i^{t+\frac{1}{2}}\rangle \mathbf{1} - \boldsymbol{\ell}_i^{t+\frac{1}{2}}, \boldsymbol{\theta}_i^{t+\frac{1}{2}}\rangle = 0$ ($\boldsymbol{x}_i^{t+\frac{1}{2}} = \boldsymbol{\theta}_i^{t+\frac{1}{2}} / \|\boldsymbol{\theta}_i^{t+\frac{1}{2}}\|_1$) and $\langle \boldsymbol{F}_i(\boldsymbol{\theta}^{t+\frac{1}{2}}), \boldsymbol{x}_i^{t+\frac{1}{2}}\rangle = \langle \langle \boldsymbol{\ell}_i^{t+\frac{1}{2}}, \boldsymbol{x}_i^{t+\frac{1}{2}}\rangle \mathbf{1} - \boldsymbol{\ell}_i^{t+\frac{1}{2}}, \boldsymbol{x}_i^{t+\frac{1}{2}}\rangle = 0$, we get

$$D_\psi(\boldsymbol{x}_i^*, \boldsymbol{\theta}_i^{t+1}) - D_\psi(\boldsymbol{x}_i^*, \boldsymbol{\theta}_i^t)$$
$$\leq \langle \boldsymbol{\theta}_i^t - \boldsymbol{\theta}_i^{t+\frac{1}{2}} + \eta \boldsymbol{F}_i(\boldsymbol{\theta}^{t-\frac{1}{2}}) - \eta \boldsymbol{F}_i(\boldsymbol{\theta}^{t+\frac{1}{2}}), \boldsymbol{x}_i^* - \boldsymbol{x}_i^{t+\frac{1}{2}}\rangle + D_\psi(\boldsymbol{\theta}_i^{t+1}, \boldsymbol{\theta}_i^{t+\frac{1}{2}}) - D_\psi(\boldsymbol{\theta}_i^{t+\frac{1}{2}}, \boldsymbol{\theta}_i^t). \tag{39}$$

Since $\boldsymbol{\theta}_i^{t+1} = \boldsymbol{\theta}_i^{t+\frac{1}{2}} - \eta \boldsymbol{F}_i(\boldsymbol{\theta}^{t-\frac{1}{2}}) + \eta \boldsymbol{F}_i(\boldsymbol{\theta}^{t+\frac{1}{2}})$, we have

$$\frac{\boldsymbol{\theta}_i^t - \boldsymbol{\theta}_i^{t+1}}{\eta} = \frac{\boldsymbol{\theta}_i^t - \boldsymbol{\theta}_i^{t+\frac{1}{2}}}{\eta} + \boldsymbol{F}_i(\boldsymbol{\theta}^{t-\frac{1}{2}}) - \boldsymbol{F}_i(\boldsymbol{\theta}^{t+\frac{1}{2}}). \tag{40}$$

From Eq. (10), we have

$$\frac{\boldsymbol{\theta}^t - \boldsymbol{\theta}^{t+1}}{\eta} - \boldsymbol{\ell}^{t+\frac{1}{2}} \in \mathcal{N}_{\mathcal{X}}(\boldsymbol{x}^{t+\frac{1}{2}}). \tag{41}$$

From the definition of weak MVI ($\langle \boldsymbol{\ell}^{\boldsymbol{x}} + \boldsymbol{z}, \boldsymbol{x} - \boldsymbol{x}^*\rangle \geq \rho\|\boldsymbol{\ell}^{\boldsymbol{x}} + \boldsymbol{z}\|_2^2, \forall \boldsymbol{z} \in \mathcal{N}_{\mathcal{X}}(\boldsymbol{x})$) and setting $\boldsymbol{x} = \boldsymbol{x}^{t+\frac{1}{2}}$ and $\boldsymbol{z} = \frac{\boldsymbol{\theta}^t - \boldsymbol{\theta}^{t+1}}{\eta} - \boldsymbol{\ell}^{t+\frac{1}{2}} \in \mathcal{N}_{\mathcal{X}}(\boldsymbol{x}^{t+\frac{1}{2}})$ (Eq. (41)), we have

$$\langle \boldsymbol{\theta}^t - \boldsymbol{\theta}^{t+1}, \boldsymbol{x}^* - \boldsymbol{x}^{t+\frac{1}{2}}\rangle = \eta\langle \boldsymbol{\ell}^{t+\frac{1}{2}} + \frac{\boldsymbol{\theta}^t - \boldsymbol{\theta}^{t+1}}{\eta} - \boldsymbol{\ell}^{t+\frac{1}{2}}, \boldsymbol{x}^* - \boldsymbol{x}^{t+\frac{1}{2}}\rangle$$
$$\leq -\rho\eta\|\frac{\boldsymbol{\theta}^t - \boldsymbol{\theta}^{t+1}}{\eta}\|_2^2 = -\frac{2\rho}{\eta}D_\psi(\boldsymbol{\theta}^{t+1}, \boldsymbol{\theta}^t). \tag{42}$$

Now, we define $c = \frac{1}{2} - 2\eta^2 D^2 L_u^2 > 0$. Combining Eq. (39), (40) and (42), we have

$$D_\psi(\boldsymbol{x}^*, \boldsymbol{\theta}^{t+1}) - D_\psi(\boldsymbol{x}^*, \boldsymbol{\theta}^t) \leq D_\psi(\boldsymbol{\theta}^{t+1}, \boldsymbol{\theta}^{t+\frac{1}{2}}) - D_\psi(\boldsymbol{\theta}^{t+\frac{1}{2}}, \boldsymbol{\theta}^t) - \frac{2\rho}{\eta}D_\psi(\boldsymbol{\theta}^{t+1}, \boldsymbol{\theta}^t)$$

$$\leq D_\psi(\boldsymbol{\theta}^{t+1}, \boldsymbol{\theta}^{t+\frac{1}{2}}) - D_\psi(\boldsymbol{\theta}^{t+\frac{1}{2}}, \boldsymbol{\theta}^t) + cD_\psi(\boldsymbol{\theta}^{t+1}, \boldsymbol{\theta}^t) - (\frac{2\rho}{\eta} + c)D_\psi(\boldsymbol{\theta}^{t+1}, \boldsymbol{\theta}^t)$$

$$\leq (1+2c)D_\psi(\boldsymbol{\theta}^{t+1}, \boldsymbol{\theta}^{t+\frac{1}{2}}) - (1-2c)D_\psi(\boldsymbol{\theta}^{t+\frac{1}{2}}, \boldsymbol{\theta}^t) - (\frac{2\rho}{\eta} + c)D_\psi(\boldsymbol{\theta}^{t+1}, \boldsymbol{\theta}^t), \tag{43}$$

where the last line comes from $D_\psi(\boldsymbol{\theta}^{t+1}, \boldsymbol{\theta}^t) \leq 2D_\psi(\boldsymbol{\theta}^{t+1}, \boldsymbol{\theta}^{t+\frac{1}{2}}) + 2D_\psi(\boldsymbol{\theta}^{t+\frac{1}{2}}, \boldsymbol{\theta}^t)$.

By using the fact that $\boldsymbol{\theta}_i^{t+1} = \boldsymbol{\theta}_i^{t+\frac{1}{2}} - \eta \boldsymbol{F}_i(\boldsymbol{\theta}^{t-\frac{1}{2}}) + \eta \boldsymbol{F}_i(\boldsymbol{\theta}^{t+\frac{1}{2}})$, Lemma 5.2 of Farina et al. [2023] ($\|\boldsymbol{F}(\boldsymbol{\theta}) - \boldsymbol{F}(\boldsymbol{\theta}')\|_2 \leq DL_u\|\boldsymbol{\theta} - \boldsymbol{\theta}'\|_2, \forall \boldsymbol{\theta}, \boldsymbol{\theta}' \in \mathbb{R}_{\geq 1}^{|\mathcal{X}|}$, where $D, L_u$ are defined in Theorem 6.1), and $D_\psi(\boldsymbol{a}, \boldsymbol{b}) = \|\boldsymbol{a} - \boldsymbol{b}\|_2^2/2$, we have

$$D_\psi(\boldsymbol{\theta}^{t+1}, \boldsymbol{\theta}^{t+\frac{1}{2}}) = D_\psi(\eta \boldsymbol{F}(\boldsymbol{\theta}^{t-\frac{1}{2}}), \eta \boldsymbol{F}(\boldsymbol{\theta}^{t+\frac{1}{2}})) \leq \eta^2 D^2 L_u^2 D_\psi(\boldsymbol{\theta}^{t-\frac{1}{2}}, \boldsymbol{\theta}^{t+\frac{1}{2}}). \tag{44}$$

By using Eq. (44), we get

$$D_\psi(\boldsymbol{\theta}^{t+\frac{1}{2}}, \boldsymbol{\theta}^{t-\frac{1}{2}}) \leq 2D_\psi(\boldsymbol{\theta}^{t+\frac{1}{2}}, \boldsymbol{\theta}^t) + 2D_\psi(\boldsymbol{\theta}^t, \boldsymbol{\theta}^{t-\frac{1}{2}}) \leq 2D_\psi(\boldsymbol{\theta}^{t+\frac{1}{2}}, \boldsymbol{\theta}^t) + 2\eta^2 D^2 L_u^2 D_\psi(\boldsymbol{\theta}^{t-\frac{1}{2}}, \boldsymbol{\theta}^{t-\frac{3}{2}}),$$

which implies

$$D_\psi(\boldsymbol{\theta}^{t+\frac{1}{2}}, \boldsymbol{\theta}^t) \geq \frac{1}{2} D_\psi(\boldsymbol{\theta}^{t+\frac{1}{2}}, \boldsymbol{\theta}^{t-\frac{1}{2}}) - \eta^2 D^2 L_u^2 D_\psi(\boldsymbol{\theta}^{t-\frac{1}{2}}, \boldsymbol{\theta}^{t-\frac{3}{2}}), \tag{45}$$

Combining Eq. (43), (44), and (45), we have

$$D_\psi(\boldsymbol{x}^*, \boldsymbol{\theta}^{t+1}) - D_\psi(\boldsymbol{x}^*, \boldsymbol{\theta}^t)$$
$$\leq (1 + 2c)D_\psi(\boldsymbol{\theta}^{t+1}, \boldsymbol{\theta}^{t+\frac{1}{2}}) - (1 - 2c)D_\psi(\boldsymbol{\theta}^{t+\frac{1}{2}}, \boldsymbol{\theta}^t) - (\frac{2\rho}{\eta} + c)D_\psi(\boldsymbol{\theta}^{t+1}, \boldsymbol{\theta}^t)$$
$$\leq -(\frac{1}{2} - c - (1 + 2c)\eta^2 D^2 L_u^2)D_\psi(\boldsymbol{\theta}^{t+\frac{1}{2}}, \boldsymbol{\theta}^{t-\frac{1}{2}}) + (1 - 2c)\eta^2 D^2 L_u^2 D_\psi(\boldsymbol{\theta}^{t-\frac{1}{2}}, \boldsymbol{\theta}^{t-\frac{3}{2}}) -$$
$$(\frac{2\rho}{\eta} + c)D_\psi(\boldsymbol{\theta}^{t+1}, \boldsymbol{\theta}^t)$$
$$\leq -(\frac{2\rho}{\eta} + c)D_\psi(\boldsymbol{\theta}^{t+1}, \boldsymbol{\theta}^t) + 4\eta^4 D^4 L_u^4 \left( D_\psi(\boldsymbol{\theta}^{t-\frac{1}{2}}, \boldsymbol{\theta}^{t-\frac{3}{2}}) - D_\psi(\boldsymbol{\theta}^{t+\frac{1}{2}}, \boldsymbol{\theta}^{t-\frac{1}{2}}) \right).$$

Telescoping the above inequality, and using $c = \frac{1}{2} - 2\eta^2 D^2 L_u^2$ with $D_\psi(\boldsymbol{a}, \boldsymbol{b}) = \|\boldsymbol{a} - \boldsymbol{b}\|_2^2/2$, we have

$$\sum_{t=1}^T \left( \frac{1}{2} + \frac{2\rho}{\eta} - 2\eta^2 D^2 L_u^2 \right) \|\boldsymbol{\theta}^{t+1} - \boldsymbol{\theta}^t\|_2^2 \leq \|\boldsymbol{\theta}^1 - \boldsymbol{x}^*\|_2^2 + 4\eta^4 D^4 L_u^4 \|\boldsymbol{\theta}^{\frac{1}{2}} - \boldsymbol{\theta}^{-\frac{1}{2}}\|_2^2$$

$$\leq \|\boldsymbol{\theta}^1 - \boldsymbol{x}^*\|_2^2 + \frac{1}{4}\|\boldsymbol{\theta}^{\frac{1}{2}} - \boldsymbol{\theta}^{-\frac{1}{2}}\|_2^2,$$

where the last line comes from $4\eta^4 D^4 L_u^4 \leq \frac{1}{4}$ (note that $c > 0$, which implies $2\eta^2 D^2 L_u^2 < \frac{1}{2}$, thus $4\eta^4 D^4 L_u^4 \leq \frac{1}{4}$).

## H  Proof of Lemma 6.3

From the definition of $\sum_{i\in\mathcal{N}}\langle \boldsymbol{\ell}_i^{t-\frac{1}{2}} - \frac{\boldsymbol{\theta}^t - \boldsymbol{\theta}^{t+\frac{1}{2}}}{\eta}, \boldsymbol{x}_i - \boldsymbol{x}_i^{t+\frac{1}{2}} \rangle$, we have

$$\sum_{i\in\mathcal{N}}\langle \boldsymbol{\ell}_i^{t-\frac{1}{2}} - \frac{\boldsymbol{\theta}_i^t - \boldsymbol{\theta}_i^{t+\frac{1}{2}}}{\eta}, \boldsymbol{x}_i - \boldsymbol{x}_i^{t+\frac{1}{2}} \rangle$$
$$= \sum_{i\in\mathcal{N}}\langle \boldsymbol{\ell}_i^{t-\frac{1}{2}} - \frac{\boldsymbol{\theta}_i^t - \boldsymbol{\theta}_i^{t+1}}{\eta} + \boldsymbol{F}_i(\boldsymbol{\theta}^{t-\frac{1}{2}}) - \boldsymbol{F}_i(\boldsymbol{\theta}^{t+\frac{1}{2}}), \boldsymbol{x}_i - \boldsymbol{x}_i^{t+\frac{1}{2}} \rangle$$
$$= \sum_{i\in\mathcal{N}}\langle \boldsymbol{\ell}_i^{t-\frac{1}{2}} - \frac{\boldsymbol{\theta}_i^t - \boldsymbol{\theta}_i^{t+1}}{\eta} + \langle \boldsymbol{\ell}_i^{t-\frac{1}{2}}, \boldsymbol{x}_i^{t-\frac{1}{2}} \rangle \mathbf{1} - \boldsymbol{\ell}_i^{t-\frac{1}{2}} - \langle \boldsymbol{\ell}_i^{t+\frac{1}{2}}, \boldsymbol{x}_i^{t+\frac{1}{2}} \rangle \mathbf{1} + \boldsymbol{\ell}_i^{t+\frac{1}{2}}, \boldsymbol{x}_i - \boldsymbol{x}_i^{t+\frac{1}{2}} \rangle$$
$$= \sum_{i\in\mathcal{N}}\langle \boldsymbol{\ell}_i^{t+\frac{1}{2}} - \frac{\boldsymbol{\theta}_i^t - \boldsymbol{\theta}_i^{t+1}}{\eta}, \boldsymbol{x}_i - \boldsymbol{x}_i^{t+\frac{1}{2}} \rangle, \tag{46}$$

where the last line is from $\langle \langle \boldsymbol{\ell}_i^{t-\frac{1}{2}}, \boldsymbol{x}_i^{t-\frac{1}{2}} \rangle \mathbf{1}, \boldsymbol{x}_i - \boldsymbol{x}_i^{t+\frac{1}{2}} \rangle = 0$ and $\langle \langle \boldsymbol{\ell}_i^{t+\frac{1}{2}}, \boldsymbol{x}_i^{t+\frac{1}{2}} \rangle \mathbf{1}, \boldsymbol{x}_i - \boldsymbol{x}_i^{t+\frac{1}{2}} \rangle = 0$.

# I  How to Obtain Eq. (8), (20), and (21) via the Analysis in Section 4

We now detail how to derive Eq. (8), (20), and (21) from Eq. (7), (2), and (3), respectively. This derivation follows from the analysis provided in Section 4. Specifically, from the analysis in Section 4, we have that $\forall\, \boldsymbol{\theta}_i^{t_2}, \boldsymbol{\theta}_i^{t_1}, \boldsymbol{\theta}_i^{t_0} \in \mathbb{R}_{\geq 1}^{|A_i|}, \eta > 0$ and $\psi(\cdot)$ as the quadratic regularizer, the update rule in Eq. (47) can be written as the form in Eq. (48).

$$\begin{cases} \boldsymbol{x}_i^{t_2} = \dfrac{\boldsymbol{\theta}_i^{t_2}}{\|\boldsymbol{\theta}_i^{t_2}\|_1}, \ \boldsymbol{\theta}_i^{t_2} \in \underset{\boldsymbol{\theta}_i \in \mathbb{R}_{\geq 1}^{|A_i|}}{\arg\min}\{\langle -\boldsymbol{F}_i(\boldsymbol{\theta}^{t_1}), \boldsymbol{\theta}_i\rangle + \dfrac{1}{\eta}D_\psi(\boldsymbol{\theta}_i, \boldsymbol{\theta}_i^{t_0})\}, \\[3mm] \boldsymbol{F}_i(\boldsymbol{\theta}^{t_1}) = \langle \dfrac{\boldsymbol{\theta}_i^{t_1}}{\|\boldsymbol{\theta}_i^{t_1}\|_1}, \boldsymbol{\ell}_i^{t_1}\rangle \mathbf{1} - \boldsymbol{\ell}_i^{t_1}, \end{cases} \tag{47}$$

$$\begin{cases} \boldsymbol{x}_i^{t_2} \in \underset{\boldsymbol{x}_i \in \mathcal{X}_i}{\arg\min}\{\langle \boldsymbol{\ell}_i^{t_1}, \boldsymbol{x}_i\rangle + f_i(\boldsymbol{x}_i) + D_{h_i}(\boldsymbol{x}_i, \boldsymbol{x}_i^{t_0})\}, \\[3mm] h_i(\boldsymbol{x}_i) + f_i(\boldsymbol{x}_i) = \dfrac{\|\boldsymbol{\theta}_i^{t_2}\|_1}{\eta}\psi(\boldsymbol{x}_i), \ h_i(\boldsymbol{x}_i) = \dfrac{\|\boldsymbol{\theta}_i^{t_0}\|_1}{\eta}\psi(\boldsymbol{x}_i), \end{cases} \tag{48}$$

where $t_0$, $t_1$, $t_2$ refer to different iterations, $\boldsymbol{x}_i^{t_2} = \boldsymbol{\theta}_i^{t_2}/\|\boldsymbol{\theta}_i^{t_2}\|_1$, $\boldsymbol{x}_i^{t_0} = \boldsymbol{\theta}_i^{t_0}/\|\boldsymbol{\theta}_i^{t_0}\|_1$.

Consider the update rule of SOGRM$^+$ as shown in the following:

$$\boldsymbol{\theta}_i^{t+\frac{1}{2}} \in \underset{\boldsymbol{\theta}_i \in \mathbb{R}_{\geq 1}^{|A_i|}}{\arg\min}\{\langle -\boldsymbol{F}_i(\boldsymbol{\theta}^{t-\frac{1}{2}}), \boldsymbol{\theta}_i\rangle + \dfrac{1}{\eta}D_\psi(\boldsymbol{\theta}_i, \boldsymbol{\theta}_i^t)\}, \ \boldsymbol{x}_i^{t+\frac{1}{2}} = \dfrac{\boldsymbol{\theta}_i^{t+\frac{1}{2}}}{\|\boldsymbol{\theta}_i^{t+\frac{1}{2}}\|_1},$$
$$\boldsymbol{\theta}_i^{t+1} = \boldsymbol{\theta}_i^{t+\frac{1}{2}} - \eta\boldsymbol{F}_i(\boldsymbol{\theta}^{t-\frac{1}{2}}) + \eta\boldsymbol{F}_i(\boldsymbol{\theta}^{t+\frac{1}{2}}). \tag{49}$$

Substituting $\boldsymbol{\theta}_i^{t_2} = \boldsymbol{\theta}_i^{t+\frac{1}{2}}, \boldsymbol{\theta}_i^{t_1} = \boldsymbol{\theta}_i^{t-\frac{1}{2}}, \boldsymbol{\theta}_i^{t_0} = \boldsymbol{\theta}_i^t$ into Eq. (47), we have that

$$\boldsymbol{\theta}_i^{t+\frac{1}{2}} \in \underset{\boldsymbol{\theta}_i \in \mathbb{R}_{\geq 1}^{|A_i|}}{\arg\min}\{\langle -\boldsymbol{F}_i(\boldsymbol{\theta}^{t-\frac{1}{2}}), \boldsymbol{\theta}_i\rangle + \dfrac{1}{\eta}D_\psi(\boldsymbol{\theta}_i, \boldsymbol{\theta}_i^t)\}, \ \boldsymbol{x}_i^{t+\frac{1}{2}} = \dfrac{\boldsymbol{\theta}_i^{t+\frac{1}{2}}}{\|\boldsymbol{\theta}_i^{t+\frac{1}{2}}\|_1},$$
$$\boldsymbol{F}_i(\boldsymbol{\theta}^{t-\frac{1}{2}}) = \langle \dfrac{\boldsymbol{\theta}_i^{t-\frac{1}{2}}}{\|\boldsymbol{\theta}_i^{t-\frac{1}{2}}\|_1}, \boldsymbol{\ell}_i^{t-\frac{1}{2}}\rangle \mathbf{1} - \boldsymbol{\ell}_i^{t-\frac{1}{2}},$$

which is consistent with the first prox-mapping operator in Eq. (49). Therefore, according to the relationship between Eq. (47) and Eq. (48), we have that the first prox-mapping operator in Eq. (49) and $\boldsymbol{x}_i^{t+\frac{1}{2}} = \boldsymbol{\theta}_i^{t+\frac{1}{2}}/\|\boldsymbol{\theta}_i^{t+\frac{1}{2}}\|_1$ can be rewritten as

$$\boldsymbol{x}_i^{t+\frac{1}{2}} \in \underset{\boldsymbol{x}_i \in \mathcal{X}_i}{\arg\min}\{\langle \boldsymbol{\ell}_i^{t-\frac{1}{2}}, \boldsymbol{x}_i\rangle + q_i^{t-\frac{1}{2}}(\boldsymbol{x}_i) + D_{q_i^{0:t-1}}(\boldsymbol{x}_i, \boldsymbol{x}_i^t)\},$$

$$q_i^{0:t-1}(\boldsymbol{x}_i) = \dfrac{\|\boldsymbol{\theta}_i^t\|_1}{\eta}\psi(\boldsymbol{x}_i), \quad q_i^{0:t-1}(\boldsymbol{x}_i) + q_i^{t-\frac{1}{2}}(\boldsymbol{x}_i) = \dfrac{\|\boldsymbol{\theta}_i^{t+\frac{1}{2}}\|_1}{\eta}\psi(\boldsymbol{x}_i).$$

In this case, $h_i(\boldsymbol{x}_i), f_i(\boldsymbol{x}_i)$ in Eq. (48) are $q_i^{0:t-1}(\boldsymbol{x}_i)$ and $q_i^{t-\frac{1}{2}}(\boldsymbol{x}_i)$, respectively. Therefore, we get Eq. (8).

Consider the update rule of SExRM$^+$ as shown in the following

$$\boldsymbol{\theta}_i^{t+\frac{1}{2}} \in \underset{\boldsymbol{\theta}_i \in \mathbb{R}_{\geq 1}^{|A_i|}}{\arg\min}\{\langle -\boldsymbol{F}_i(\boldsymbol{\theta}^{t-\frac{1}{2}}), \boldsymbol{\theta}_i\rangle + \dfrac{1}{\eta}D_\psi(\boldsymbol{\theta}_i, \boldsymbol{\theta}_i^t)\}, \ \boldsymbol{x}_i^{t+\frac{1}{2}} = \dfrac{\boldsymbol{\theta}_i^{t+\frac{1}{2}}}{\|\boldsymbol{\theta}_i^{t+\frac{1}{2}}\|_1},$$
$$\boldsymbol{\theta}_i^{t+1} \in \underset{\boldsymbol{\theta}_i \in \mathbb{R}_{\geq 1}^{|A_i|}}{\arg\min}\{\langle -\boldsymbol{F}_i(\boldsymbol{\theta}^{t+\frac{1}{2}}), \boldsymbol{\theta}_i\rangle + \dfrac{1}{\eta}D_\psi(\boldsymbol{\theta}_i, \boldsymbol{\theta}_i^t)\}, \ \boldsymbol{x}_i^{t+1} = \dfrac{\boldsymbol{\theta}_i^{t+1}}{\|\boldsymbol{\theta}_i^{t+1}\|_1}. \tag{50}$$

Substituting $\boldsymbol{\theta}_i^{t_2} = \boldsymbol{\theta}_i^{t+\frac{1}{2}}, \boldsymbol{\theta}_i^{t_1} = \boldsymbol{\theta}_i^{t-\frac{1}{2}}, \boldsymbol{\theta}_i^{t_0} = \boldsymbol{\theta}_i^t$ into Eq. (47), we have that

$$\boldsymbol{\theta}_i^{t+\frac{1}{2}} \in \underset{\boldsymbol{\theta}_i \in \mathbb{R}_{\geq 1}^{|A_i|}}{\arg\min} \{\langle -\boldsymbol{F}_i(\boldsymbol{\theta}^t), \boldsymbol{\theta}_i \rangle + \frac{1}{\eta} D_\psi(\boldsymbol{\theta}_i, \boldsymbol{\theta}_i^t)\}, \ \boldsymbol{x}_i^{t+\frac{1}{2}} = \frac{\boldsymbol{\theta}_i^{t+\frac{1}{2}}}{\|\boldsymbol{\theta}_i^{t+\frac{1}{2}}\|_1},$$

$$\boldsymbol{F}_i(\boldsymbol{\theta}^t) = \langle \frac{\boldsymbol{\theta}_i^t}{\|\boldsymbol{\theta}_i^t\|_1}, \boldsymbol{\ell}_i^t \rangle \mathbf{1} - \boldsymbol{\ell}_i^t,$$

which is consistent with the first prox-mapping operator in Eq. (50). Therefore, according to the relationship between Eq. (47) and Eq. (48), we have that the first prox-mapping operator in Eq. (50) and $\boldsymbol{x}_i^{t+\frac{1}{2}} = \boldsymbol{\theta}_i^{t+\frac{1}{2}}/\|\boldsymbol{\theta}_i^{t+\frac{1}{2}}\|_1$ can be rewritten as

$$\boldsymbol{x}_i^{t+\frac{1}{2}} \in \underset{\boldsymbol{x}_i \in \mathcal{X}_i}{\arg\min} \{\langle \boldsymbol{\ell}_i^{t-\frac{1}{2}}, \boldsymbol{x}_i \rangle + q_i^{t-\frac{1}{2}}(\boldsymbol{x}_i) + D_{q_i^{0:t-1}}(\boldsymbol{x}_i, \boldsymbol{x}_i^t)\},$$

$$q_i^{0:t-1}(\boldsymbol{x}_i) = \frac{\|\boldsymbol{\theta}_i^t\|_1}{\eta} \psi(\boldsymbol{x}_i), \quad q_i^{0:t-1}(\boldsymbol{x}_i) + q_i^{t-\frac{1}{2}}(\boldsymbol{x}_i) = \frac{\|\boldsymbol{\theta}_i^{t+\frac{1}{2}}\|_1}{\eta} \psi(\boldsymbol{x}_i). \tag{51}$$

In this case, $h_i(\boldsymbol{x}_i), f_i(\boldsymbol{x}_i)$ in Eq. (48) are $q_i^{0:t-1}(\boldsymbol{x}_i)$ and $q_i^{t-\frac{1}{2}}(\boldsymbol{x}_i)$, respectively. Similarly, substituting $\boldsymbol{\theta}_i^{t_2} = \boldsymbol{\theta}_i^{t+1}, \boldsymbol{\theta}_i^{t_1} = \boldsymbol{\theta}_i^{t+\frac{1}{2}}, \boldsymbol{\theta}_i^{t_0} = \boldsymbol{\theta}_i^t$ into Eq. (47), we have that

$$\boldsymbol{\theta}_i^{t+1} \in \underset{\boldsymbol{\theta}_i \in \mathbb{R}_{\geq 1}^{|A_i|}}{\arg\min} \{\langle -\boldsymbol{F}_i(\boldsymbol{\theta}^{t+\frac{1}{2}}), \boldsymbol{\theta}_i \rangle + \frac{1}{\eta} D_\psi(\boldsymbol{\theta}_i, \boldsymbol{\theta}_i^t)\}, \ \boldsymbol{x}_i^{t+1} = \frac{\boldsymbol{\theta}_i^{t+1}}{\|\boldsymbol{\theta}_i^{t+1}\|_1},$$

$$\boldsymbol{F}_i(\boldsymbol{\theta}^{t+\frac{1}{2}}) = \langle \frac{\boldsymbol{\theta}_i^{t+\frac{1}{2}}}{\|\boldsymbol{\theta}_i^{t+\frac{1}{2}}\|_1}, \boldsymbol{\ell}_i^{t+\frac{1}{2}} \rangle \mathbf{1} - \boldsymbol{\ell}_i^{t+\frac{1}{2}},$$

which is consistent with the second prox-mapping operator in Eq. (50). Therefore, according to the relationship between Eq. (47) and Eq. (48), we have that the second prox-mapping operator in Eq. (50) and $\boldsymbol{x}_i^{t+1} = \boldsymbol{\theta}_i^{t+1}/\|\boldsymbol{\theta}_i^{t+1}\|_1$ can be rewritten as

$$\boldsymbol{x}_i^{t+1} \in \underset{\boldsymbol{x}_i \in \mathcal{X}_i}{\arg\min} \{\langle \boldsymbol{\ell}_i^{t+\frac{1}{2}}, \boldsymbol{x}_i \rangle + q_i^t(\boldsymbol{x}_i) + D_{q_i^{0:t-1}}(\boldsymbol{x}_i, \boldsymbol{x}_i^t)\},$$

$$q_i^{0:t-1}(\boldsymbol{x}_i) = \frac{\|\boldsymbol{\theta}_i^t\|_1}{\eta} \psi(\boldsymbol{x}_i), \quad q_i^{0:t-1}(\boldsymbol{x}_i) + q_i^t(\boldsymbol{x}_i) = \frac{\|\boldsymbol{\theta}_i^{t+1}\|_1}{\eta} \psi(\boldsymbol{x}_i). \tag{52}$$

In this case, $h_i(\boldsymbol{x}_i), f_i(\boldsymbol{x}_i)$ in Eq. (48) are $q_i^{0:t-1}(\boldsymbol{x}_i)$ and $q_i^t(\boldsymbol{x}_i)$, respectively. Combining Eq. (51) with (52), we get Eq. (20).

Consider the update rule of SPRM$^+$ as shown in the following

$$\boldsymbol{\theta}_i^{t+\frac{1}{2}} \in \underset{\boldsymbol{\theta}_i \in \mathbb{R}_{\geq 1}^{|A_i|}}{\arg\min} \{\langle -\boldsymbol{F}_i(\boldsymbol{\theta}^{t-\frac{1}{2}}), \boldsymbol{\theta}_i \rangle + \frac{1}{\eta} D_\psi(\boldsymbol{\theta}_i, \boldsymbol{\theta}_i^t)\}, \ \boldsymbol{x}_i^{t+\frac{1}{2}} = \frac{\boldsymbol{\theta}_i^{t+\frac{1}{2}}}{\|\boldsymbol{\theta}_i^{t+\frac{1}{2}}\|_1},$$

$$\boldsymbol{\theta}_i^{t+1} \in \underset{\boldsymbol{\theta}_i \in \mathbb{R}_{\geq 1}^{|A_i|}}{\arg\min} \{\langle -\boldsymbol{F}_i(\boldsymbol{\theta}^{t+\frac{1}{2}}), \boldsymbol{\theta}_i \rangle + \frac{1}{\eta} D_\psi(\boldsymbol{\theta}_i, \boldsymbol{\theta}_i^t)\}, \ \boldsymbol{x}_i^{t+1} = \frac{\boldsymbol{\theta}_i^{t+1}}{\|\boldsymbol{\theta}_i^{t+1}\|_1}. \tag{53}$$

Substituting $\boldsymbol{\theta}_i^{t_2} = \boldsymbol{\theta}_i^{t+\frac{1}{2}}, \boldsymbol{\theta}_i^{t_1} = \boldsymbol{\theta}_i^{t-\frac{1}{2}}, \boldsymbol{\theta}_i^{t_0} = \boldsymbol{\theta}_i^t$ into Eq. (47), we have that

$$\boldsymbol{\theta}_i^{t+\frac{1}{2}} \in \underset{\boldsymbol{\theta}_i \in \mathbb{R}_{\geq 1}^{|A_i|}}{\arg\min} \{\langle -\boldsymbol{F}_i(\boldsymbol{\theta}^{t-\frac{1}{2}}), \boldsymbol{\theta}_i \rangle + \frac{1}{\eta} D_\psi(\boldsymbol{\theta}_i, \boldsymbol{\theta}_i^t)\}, \ \boldsymbol{x}_i^{t+\frac{1}{2}} = \frac{\boldsymbol{\theta}_i^{t+\frac{1}{2}}}{\|\boldsymbol{\theta}_i^{t+\frac{1}{2}}\|_1},$$

$$\boldsymbol{F}_i(\boldsymbol{\theta}^{t-\frac{1}{2}}) = \langle \frac{\boldsymbol{\theta}_i^{t-\frac{1}{2}}}{\|\boldsymbol{\theta}_i^{t-\frac{1}{2}}\|_1}, \boldsymbol{\ell}_i^{t-\frac{1}{2}} \rangle \mathbf{1} - \boldsymbol{\ell}_i^{t-\frac{1}{2}},$$

which is consistent with the first prox-mapping operator in Eq. (53). Therefore, according to the relationship between Eq. (47) and Eq. (48), we have that the first prox-mapping operator in Eq. (53)

and $x_i^{t+\frac{1}{2}} = \theta_i^{t+\frac{1}{2}}/\|\theta_i^{t+\frac{1}{2}}\|_1$ can be rewritten as

$$x_i^{t+\frac{1}{2}} \in \underset{x_i \in \mathcal{X}_i}{\arg\min}\{\langle \ell_i^{t-\frac{1}{2}}, x_i\rangle + q_i^{t-\frac{1}{2}}(x_i) + D_{q_i^{0:t-1}}(x_i, x_i^t)\},$$

$$q_i^{0:t-1}(x_i) = \frac{\|\theta_i^t\|_1}{\eta}\psi(x_i), \quad q_i^{0:t-1}(x_i) + q_i^{t-\frac{1}{2}}(x_i) = \frac{\|\theta_i^{t+\frac{1}{2}}\|_1}{\eta}\psi(x_i). \tag{54}$$

In this case, $h_i(x_i)$, $f_i(x_i)$ in Eq. (48) are $q_i^{0:t-1}(x_i)$ and $q_i^{t-\frac{1}{2}}(x_i)$, respectively. Similarly, substituting $\theta_i^{t_2} = \theta_i^{t+1}, \theta_i^{t_1} = \theta_i^{t+\frac{1}{2}}, \theta_i^{t_0} = \theta_i^t$ into Eq. (47), we have that

$$\theta_i^{t+1} \in \underset{\theta_i \in \mathbb{R}_{\geq 1}^{|A_i|}}{\arg\min}\{\langle -F_i(\theta^{t+\frac{1}{2}}), \theta_i\rangle + \frac{1}{\eta}D_\psi(\theta_i, \theta_i^t)\}, \quad x_i^{t+1} = \frac{\theta_i^{t+1}}{\|\theta_i^{t+1}\|_1},$$

$$F_i(\theta^{t+\frac{1}{2}}) = \langle \frac{\theta_i^{t+\frac{1}{2}}}{\|\theta_i^{t+\frac{1}{2}}\|_1}, \ell_i^{t+\frac{1}{2}}\rangle \mathbf{1} - \ell_i^{t+\frac{1}{2}},$$

which is consistent with the second prox-mapping operator in Eq. (53). Therefore, according to the relationship between Eq. (47) and Eq. (48), we have that the second prox-mapping operator in Eq. (53) and $x_i^{t+1} = \theta_i^{t+1}/\|\theta_i^{t+1}\|_1$ can be rewritten as

$$x_i^{t+1} \in \underset{x_i \in \mathcal{X}_i}{\arg\min}\{\langle \ell_i^{t+\frac{1}{2}}, x_i\rangle + q_i^t(x_i) + D_{q_i^{0:t-1}}(x_i, x_i^t)\},$$

$$q_i^{0:t-1}(x_i) = \frac{\|\theta_i^t\|_1}{\eta}\psi(x_i), \quad q_i^{0:t-1}(x_i) + q_i^t(x_i) = \frac{\|\theta_i^{t+1}\|_1}{\eta}\psi(x_i). \tag{55}$$

In this case, $h_i(x_i)$, $f_i(x_i)$ in Eq. (48) are $q_i^{0:t-1}(x_i)$ and $q_i^t(x_i)$, respectively. Combining Eq. (54) with (55), we get Eq. (21).

## J  Example of Different Game Types

In this section, we provide examples of smooth games that satisfy the MVI and weak MVI, respectively. We do not provide the example of smooth games satisfying monotonicity as any two-player zero-sum matrix game is a smooth game and satisfies monotonicity. Note that in this section, we focus on two-player normal-form game, whose utility function is convex and represented by payoff matrices. Any two-player normal-form game is a smooth game. For each two-player normal-form game, the utility functions of player $0$ and $1$ are presented by payoff matrices $A$ and $B$, respectively. Formally, $u_0(x) = x_0^\mathrm{T} A x_1$ and $u_1(x) = x_1^\mathrm{T} B^\mathrm{T} x_0$, which implies $\ell_0^x = -A x_1$ and $\ell_1^x = -B^\mathrm{T} x_0$.

### J.1  Example of Games Satisfying the MVI

The example is defined as following

$$A = \begin{pmatrix} 2 & 0 \\ 0 & 0 \end{pmatrix}, \quad B = \begin{pmatrix} 2 & 0 \\ 0 & 0 \end{pmatrix}.$$

This game violates monotonicity when

$$x_0 = \begin{pmatrix} 0 \\ 1 \end{pmatrix}, \quad x_1 = \begin{pmatrix} 0 \\ 1 \end{pmatrix}, \quad x_0' = \begin{pmatrix} 0.1 \\ 0.9 \end{pmatrix}, \quad x_1' = \begin{pmatrix} 0.1 \\ 0.9 \end{pmatrix}.$$

Formally, in this case, we have

$$\ell_0^x = \begin{pmatrix} 0 \\ 0 \end{pmatrix}, \quad \ell_1^x = \begin{pmatrix} -2 \\ 0 \end{pmatrix}, \quad \ell_0^{x'} = \begin{pmatrix} 0 \\ 0 \end{pmatrix}, \quad \ell_1^{x'} = \begin{pmatrix} -2 \\ 0 \end{pmatrix}.$$

$$\langle \ell^x - \ell^{x'}, x - x'\rangle = \begin{pmatrix} 2 \\ 0 \end{pmatrix} \cdot \begin{pmatrix} -0.1 \\ 0.1 \end{pmatrix} + \begin{pmatrix} 2 \\ 0 \end{pmatrix} \cdot \begin{pmatrix} -0.1 \\ 0.1 \end{pmatrix} = -0.4 < 0$$

which violates monotonicity.

Now, we show that the provided example satisfies the MVI. An NE of this game (learned by "Nashpy" [Knight and Campbell, 2018]) is

$$\boldsymbol{x}_0^* = \begin{pmatrix} 1 \\ 0 \end{pmatrix}, \quad \boldsymbol{x}_1^* = \begin{pmatrix} 1 \\ 0 \end{pmatrix}.$$

We define the strategies of players as following

$$\boldsymbol{x}_0 = \begin{pmatrix} a \\ 1-a \end{pmatrix}, \quad \boldsymbol{x}_1 = \begin{pmatrix} b \\ 1-b \end{pmatrix},$$

where $0 \le a \le 1$ and $0 \le b \le 1$. The loss gradient $\boldsymbol{\ell}_i^{\boldsymbol{x}}$ of player $i$ is

$$\boldsymbol{\ell}_0^{\boldsymbol{x}} = -\boldsymbol{A}\boldsymbol{x}_1, \quad \boldsymbol{\ell}_1^{\boldsymbol{x}} = -\boldsymbol{B}^{\mathrm{T}}\boldsymbol{x}_0.$$

Formally, for player 0, we have

$$\boldsymbol{A}\boldsymbol{x}_1 = \begin{pmatrix} 2 & 0 \\ 0 & 0 \end{pmatrix} \begin{pmatrix} b \\ 1-b \end{pmatrix} = \begin{pmatrix} 2b \\ 0 \end{pmatrix},$$

$$\boldsymbol{\ell}_0^{\boldsymbol{x}} = -\boldsymbol{A}\boldsymbol{x}_1 = \begin{pmatrix} -2b \\ 0 \end{pmatrix}.$$

Similarly, for player 1, we have

$$\boldsymbol{B}^{\mathrm{T}} = \begin{pmatrix} 2 & 0 \\ 0 & 0 \end{pmatrix},$$

$$\boldsymbol{B}^{\mathrm{T}}\boldsymbol{x}_0 = \begin{pmatrix} 2 & 0 \\ 0 & 0 \end{pmatrix} \begin{pmatrix} a \\ 1-a \end{pmatrix} = \begin{pmatrix} -2a \\ 0 \end{pmatrix},$$

$$\boldsymbol{\ell}_1^{\boldsymbol{x}} = -\boldsymbol{B}^{\mathrm{T}}\boldsymbol{x}_0 = \begin{pmatrix} -2a \\ 0 \end{pmatrix}.$$

In this case, we have

$$\langle \boldsymbol{\ell}^{\boldsymbol{x}}, \boldsymbol{x} - \boldsymbol{x}^* \rangle = \begin{pmatrix} -2b \\ 0 \end{pmatrix} \cdot \begin{pmatrix} a-1 \\ 1-a \end{pmatrix} + \begin{pmatrix} -2a \\ 0 \end{pmatrix} \cdot \begin{pmatrix} b-1 \\ 1-b \end{pmatrix} = -4ab + 2a + 2b = (-4a+2)b + 2a.$$

We can find that $(-4a+2)b+2a$ is linear function w.r.t $b$ given fixed $a$. If $1 \ge a \ge \frac{1}{2}$, $(-4a+2)b+2a$ decreases as $b$ increases. Therefore, given $1 \ge a \ge \frac{1}{2}$, $\min_{0 \le b \le 1}(-4a+2)b+2a = (-4a+2)+2a = 2 - 2a \ge 0$. Similarly, if $0 \le a < \frac{1}{2}$ $(-4a+2)b+2a$ decreases as $b$ decreases. Therefore, given $0 \le a < \frac{1}{2}$, $\min_{0 \le b \le 1}(-4a+2)b+2a = 2a \ge 0$. Hence, we get $-4ab + 2b + 2a \ge 0$, which implies $-4ab + 2a + 2b \ge 0$. Therefore, we get

$$\langle \boldsymbol{\ell}^{\boldsymbol{x}}, \boldsymbol{x} - \boldsymbol{x}^* \rangle = -4ab + 2a + 2b \ge 0.$$

Then, we have $\langle \boldsymbol{\ell}^{\boldsymbol{x}}, \boldsymbol{x} - \boldsymbol{x}^* \rangle \ge 0, \forall \boldsymbol{x} \in \mathcal{X}$ and $\exists \boldsymbol{x}^* \in \mathcal{X}^*$, which means the MVI holds in this game.

## J.2   Example of Games Satisfying the Weak MVI

The example is defined as following

$$\boldsymbol{A} = \begin{pmatrix} 1 & 0 \\ -1 & 1 \end{pmatrix}, \quad \boldsymbol{B} = \begin{pmatrix} 0 & 1 \\ -1 & 1 \end{pmatrix}.$$

The unique NE of this game (learned by "Nashpy" [Knight and Campbell, 2018]) is

$$\boldsymbol{x}_0^* = \begin{pmatrix} 0 \\ 1 \end{pmatrix}, \quad \boldsymbol{x}_1^* = \begin{pmatrix} 0 \\ 1 \end{pmatrix}.$$

This game violates the MVI when

$$\boldsymbol{x}_0 = \begin{pmatrix} 0.7 \\ 0.3 \end{pmatrix}, \quad \boldsymbol{x}_1 = \begin{pmatrix} 0.9 \\ 0.1 \end{pmatrix}.$$

Formally, in this case, we have

$$\langle \boldsymbol{\ell^x}, \boldsymbol{x} - \boldsymbol{x^*} \rangle = \begin{pmatrix} -0.9 \\ 0.8 \end{pmatrix} \cdot \begin{pmatrix} 0.7 - 0 \\ 0.3 - 1 \end{pmatrix} + \begin{pmatrix} 0.3 \\ -1 \end{pmatrix} \cdot \begin{pmatrix} 0.9 - 0 \\ 0.1 - 1 \end{pmatrix} = -0.02 < 0,$$

which violates the MVI.

Now, we show that the provided example satisfies the weak MVI. From Meng et al. [2025], for any smooth game, we have

$$r^{dg}(\boldsymbol{x}) = \max_{\boldsymbol{x'} \in \mathcal{X}} \langle \boldsymbol{\ell^x}, \boldsymbol{x} - \boldsymbol{x'} \rangle \leq C_1 r^{tan}(\boldsymbol{x}) = C_1 \min_{\boldsymbol{z} \in \mathcal{N}_{\mathcal{X}}(\boldsymbol{x})} \|\boldsymbol{\ell^x} + \boldsymbol{z}\|_2,$$

where $C_1$ is a game-dependent constant. Recall the definition of the weak MVI

$$\langle \boldsymbol{\ell^x} + \boldsymbol{z}, \boldsymbol{x} - \boldsymbol{x^*} \rangle \geq \rho \|\boldsymbol{\ell^x} + \boldsymbol{z}\|_2^2, \forall \boldsymbol{z} \in \mathcal{N}_{\mathcal{X}}(\boldsymbol{x}).$$

Therefore, if we can show that

$$\langle \boldsymbol{\ell^x}, \boldsymbol{x} - \boldsymbol{x^*} \rangle \geq -(r^{dg}(\boldsymbol{x}))^2,$$

we can always find a $\rho = -C_1^2 < 0$ to ensure the weak MVI holds since $\forall \boldsymbol{z} \in \mathcal{N}_{\mathcal{X}}(\boldsymbol{x})$,

$$\langle \boldsymbol{\ell^x}, \boldsymbol{x} - \boldsymbol{x^*} \rangle \geq -(r^{dg}(\boldsymbol{x}))^2 = -(\max_{\boldsymbol{x'} \in \mathcal{X}} \langle \boldsymbol{\ell^x}, \boldsymbol{x} - \boldsymbol{x'} \rangle)^2 \geq -(C_1 r^{tan}(\boldsymbol{x}))^2 = -C_1^2 \min_{\boldsymbol{z'} \in \mathcal{N}_{\mathcal{X}}(\boldsymbol{x})} \|\boldsymbol{\ell^x} + \boldsymbol{z'}\|_2^2 \geq -C_1^2 \|\boldsymbol{\ell^x} + \boldsymbol{z}\|_2^2,$$

and

$$\langle \boldsymbol{z}, \boldsymbol{x} - \boldsymbol{x^*} \rangle \geq 0.$$

Now, we show that $\langle \boldsymbol{\ell^x}, \boldsymbol{x} - \boldsymbol{x^*} \rangle \geq -(r^{dg}(\boldsymbol{x}))^2$ holds in this game. We define the strategies of players as following

$$\boldsymbol{x}_0 = \begin{pmatrix} a \\ 1 - a \end{pmatrix}, \quad \boldsymbol{x}_1 = \begin{pmatrix} b \\ 1 - b \end{pmatrix},$$

where $0 \leq a \leq 1$ and $0 \leq b \leq 1$. The loss gradient $\boldsymbol{\ell_i^x}$ of player $i$ is

$$\boldsymbol{\ell_0^x} = -\boldsymbol{A}\boldsymbol{x}_1, \quad \boldsymbol{\ell_1^x} = -\boldsymbol{B}^{\mathrm{T}}\boldsymbol{x}_0.$$

Formally, for player 0, we have

$$\boldsymbol{A}\boldsymbol{x}_1 = \begin{pmatrix} 1 & 0 \\ -1 & 1 \end{pmatrix} \begin{pmatrix} b \\ 1 - b \end{pmatrix} = \begin{pmatrix} b \\ -b + 1 - b \end{pmatrix} = \begin{pmatrix} b \\ 1 - 2b \end{pmatrix},$$

$$\boldsymbol{\ell_0^x} = -\boldsymbol{A}\boldsymbol{x}_1 = \begin{pmatrix} -b \\ -(1 - 2b) \end{pmatrix} = \begin{pmatrix} -b \\ 2b - 1 \end{pmatrix}.$$

Similarly, for player 1, we have

$$\boldsymbol{B}^{\mathrm{T}} = \begin{pmatrix} 0 & -1 \\ 1 & 1 \end{pmatrix},$$

$$\boldsymbol{B}^{\mathrm{T}}\boldsymbol{x}_0 = \begin{pmatrix} 0 & -1 \\ 1 & 1 \end{pmatrix} \begin{pmatrix} a \\ 1 - a \end{pmatrix} = \begin{pmatrix} -(1 - a) \\ a + (1 - a) \end{pmatrix} = \begin{pmatrix} a - 1 \\ 1 \end{pmatrix},$$

$$\boldsymbol{\ell_1^x} = -\boldsymbol{B}^{\mathrm{T}}\boldsymbol{x}_0 = \begin{pmatrix} 1 - a \\ -1 \end{pmatrix}.$$

Now, we show $\langle \boldsymbol{\ell^x}, \boldsymbol{x} - \boldsymbol{x^*} \rangle \geq -(r^{dg}(\boldsymbol{x}))^2$ by showing $\langle \boldsymbol{\ell^x}, \boldsymbol{x} - \boldsymbol{x^*} \rangle + (r^{dg}(\boldsymbol{x}))^2 \geq 0$ holds. We first compute $\langle \boldsymbol{\ell^x}, \boldsymbol{x} - \boldsymbol{x^*} \rangle$. Formally, we get

$$\boldsymbol{x}_0 - \boldsymbol{x}_0^* = \begin{pmatrix} a \\ 1 - a \end{pmatrix} - \begin{pmatrix} 0 \\ 1 \end{pmatrix} = \begin{pmatrix} a \\ -a \end{pmatrix},$$

$$\boldsymbol{x}_1 - \boldsymbol{x}_1^* = \begin{pmatrix} b \\ 1 - b \end{pmatrix} - \begin{pmatrix} 0 \\ 1 \end{pmatrix} = \begin{pmatrix} b \\ -b \end{pmatrix}.$$

Next, calculate the dot products

$$\langle \boldsymbol{\ell_0^x}, \boldsymbol{x}_0 - \boldsymbol{x}_0^* \rangle = \begin{pmatrix} -b \\ 2b - 1 \end{pmatrix} \cdot \begin{pmatrix} a \\ -a \end{pmatrix} = -ab - a(2b - 1) = -3ab + a,$$

$$\langle \boldsymbol{\ell}_1^{\boldsymbol{x}}, \boldsymbol{x}_1 - \boldsymbol{x}_1^* \rangle = \begin{pmatrix} 1-a \\ -1 \end{pmatrix} \cdot \begin{pmatrix} b \\ -b \end{pmatrix} = b(1-a) + b = b(1 - a + 1) = b(2-a).$$

Combine the results

$$\langle \boldsymbol{\ell}^{\boldsymbol{x}}, \boldsymbol{x} - \boldsymbol{x}^* \rangle = \langle \boldsymbol{\ell}_0^{\boldsymbol{x}}, \boldsymbol{x}_0 - \boldsymbol{x}_0^* \rangle + \langle \boldsymbol{\ell}_1^{\boldsymbol{x}}, \boldsymbol{x}_1 - \boldsymbol{x}_1^* \rangle = -3ab + a + b(2-a).$$

This simplifies to:

$$\langle \boldsymbol{\ell}^{\boldsymbol{x}}, \boldsymbol{x} - \boldsymbol{x}^* \rangle = -3ab + a + 2b - ab = -4ab + 2b + a.$$

Similarly, for $r^{dg}(\boldsymbol{x}) = \max_{\boldsymbol{x}' \in \mathcal{X}} \langle \boldsymbol{\ell}^{\boldsymbol{x}}, \boldsymbol{x} - \boldsymbol{x}' \rangle$, we get

$$\max_{\boldsymbol{x}' \in \mathcal{X}} \langle \boldsymbol{\ell}^{\boldsymbol{x}}, \boldsymbol{x} - \boldsymbol{x}' \rangle = \langle \boldsymbol{\ell}_0^{\boldsymbol{x}}, \boldsymbol{x}_0 \rangle - \min(\boldsymbol{\ell}_0^{\boldsymbol{x}}[0], \boldsymbol{\ell}_0^{\boldsymbol{x}}[1]) + \langle \boldsymbol{\ell}_1^{\boldsymbol{x}}, \boldsymbol{x}_1 \rangle - \min(\boldsymbol{\ell}_1^{\boldsymbol{x}}[0], \boldsymbol{\ell}_1^{\boldsymbol{x}}[1]),$$

which results in

$$\max_{\boldsymbol{x}' \in \mathcal{X}} \langle \boldsymbol{\ell}^{\boldsymbol{x}}, \boldsymbol{x} - \boldsymbol{x}' \rangle = -4ab + 4b - 2 + a - \min(-b, 2b-1) - \min(1-a, -1) = -4ab + 4b - 1 + a - \min(-b, 2b-1)$$

**Case 1:** If $0 \le b \le \frac{1}{3}$,

$$\langle \boldsymbol{\ell}^{\boldsymbol{x}}, \boldsymbol{x} - \boldsymbol{x}^* \rangle + (r^{dg}(\boldsymbol{x}))^2 = -4ab + 2b + a + (-4ab + 4b - 1 + a - 2b + 1)^2 = -4ab + 2b + a + (-4ab + 2b + a)^2.$$

It is obviously if $-4ab + 2b + a \ge 0$, $-4ab + 2b + a + (-4ab + 4b - 1 + a - 2b + 1)^2 = -4ab + 2b + a + (-4ab + 2b + a)^2 \ge 0$. Now, we show $-4ab + 2b + a \ge 0$. Formally, we get

$$-4ab + 2b + a = (-4a + 2)b + a.$$

We can find that $(-4a + 2)b + a$ is linear function w.r.t $b$ given fixed $a$. If $1 \ge a \ge \frac{1}{2}$, $(-4a + 2)b + a$ decreases as $b$ increases. Therefore, given $1 \ge a \ge \frac{1}{2}$, $\min_{0 \le b \le \frac{1}{3}} (-4a + 2)b + a = (-4a + 2)\frac{1}{3} + a = \frac{2}{3} - \frac{a}{3} \ge \frac{1}{3}$. Similarly, if $0 \le a < \frac{1}{2}$, $(-4a + 2)b + a$ decreases as $b$ decreases. Therefore, given $0 \le a < \frac{1}{2}$, $\min_{0 \le b \le \frac{1}{3}} (-4a + 2)b + a = a \ge 0$. Hence, we get $-4ab + 2b + a \ge 0$, which implies $-4ab + 2b + a + (-4ab + 4b - 1 + a - 2b + 1)^2 = -4ab + 2b + a + (-4ab + 2b + a)^2 \ge 0$.

**Case 2:** If $\frac{1}{3} \le b \le 1$,

$$\langle \boldsymbol{\ell}^{\boldsymbol{x}}, \boldsymbol{x} - \boldsymbol{x}^* \rangle + (r^{dg}(\boldsymbol{x}))^2 = -4ab + 2b + a + (-4ab + 4b - 1 + a + b)^2 = -4ab + 2b + a + (-4ab + 5b + a - 1)^2.$$

Now, we simplify the expression

$$-4ab + 2b + a + (-4ab + 5b + a - 1)^2.$$

Then,

$(-4ab + 5b + a - 1)^2$
$= (-4ab + 5b + a - 1)(-4ab + 5b + a - 1)$
$= (-4ab)^2 + (5b)^2 + a^2 + (-1)^2 + 2(-4ab \cdot 5b) + 2(-4ab \cdot a) + 2(-4ab \cdot -1) + 2(5b \cdot a) + 2(5b \cdot -1) + 2(a \cdot -1)$
$= 16a^2b^2 + 25b^2 + a^2 + 1 - 40ab^2 - 8a^2b + 8ab + 10ab - 10b - 2a$
$= 16a^2b^2 + 25b^2 + a^2 - 40ab^2 - 8a^2b + 18ab - 10b - 2a + 1.$

So the full expression is

$$-4ab + 2b + a + 16a^2b^2 + 25b^2 + a^2 - 40ab^2 - 8a^2b + 18ab - 10b - 2a + 1.$$

Therefore, we define

$$f(a) = (16b^2 - 8b + 1)a^2 + (-40b^2 + 14b - 1)a + 25b^2 - 8b + 1.$$

For $f(a)$, given a fixed $b$, it is a quadratic function with respect to $a$. For the term $32b^2 - 16b + 2$, as it takes the minimum value when $b = \frac{16}{64} = \frac{1}{4}$, we have that the value of $32b^2 - 16b + 2$ increases as $b$ increases when $\frac{1}{3} \le b \le 1$. Therefore, the minimum and maximum values of $32b^2 - 16b + 2$ when $\frac{1}{3} \le b \le 1$ are $32\frac{1}{9} - \frac{16}{3} + 2 = \frac{2}{9}$ and $32 - 16 + 2 = 18$, respectively. As $32b^2 - 16b + 2 > 0$, for $f(a)$, given a fixed $b$, so it takes the minimum value in the following case

$$a = \frac{40b^2 - 14b + 1}{32b^2 - 16b + 2} = \frac{32b^2 - 16b + 2 + 8b^2 + 2b - 1}{32b^2 - 16b + 2} = 1 + \frac{8b^2 + 2b - 1}{32b^2 - 16b + 2}.$$

**Algorithm 1** SExRM$^+$

---

**Require:** Step size $\eta \in \left(0, \frac{1}{DL_u}\right)$.

1: Initialize: $\boldsymbol{\theta}_i^1 \leftarrow \mathbf{1}/|A_i|, \forall i \in \mathcal{N}$
2: **for** $t = 1, 2, \ldots$ **do**
3:     **for** $i \in \mathcal{N}$ **do**
4:         $\boldsymbol{\theta}_i^{t+\frac{1}{2}} \in \arg\min_{\boldsymbol{\theta}_i \in \mathbb{R}_{\geq 1}^{|A_i|}} \{\langle -\boldsymbol{F}_i(\boldsymbol{\theta}^t), \boldsymbol{\theta}_i\rangle + \frac{1}{\eta} D_\psi(\boldsymbol{\theta}_i, \boldsymbol{\theta}_i^t)\}, \; \boldsymbol{x}_i^{t+\frac{1}{2}} = \frac{\boldsymbol{\theta}_i^{t+\frac{1}{2}}}{\|\boldsymbol{\theta}_i^{t+\frac{1}{2}}\|_1}$
5:     **end for**
6:     **for** $i \in \mathcal{N}$ **do**
7:         $\boldsymbol{\theta}_i^{t+1} \in \arg\min_{\boldsymbol{\theta}_i \in \mathbb{R}_{\geq 1}^{|A_i|}} \{\langle -\boldsymbol{F}_i(\boldsymbol{\theta}^{t+\frac{1}{2}}), \boldsymbol{\theta}_i\rangle + \frac{1}{\eta} D_\psi(\boldsymbol{\theta}_i, \boldsymbol{\theta}_i^t)\}, \; \boldsymbol{x}_i^{t+1} = \frac{\boldsymbol{\theta}_i^{t+1}}{\|\boldsymbol{\theta}_i^{t+1}\|_1}$
8:     **end for**
9: **end for**

---

For the term $8b^2 + 2b - 1$, as it takes the minimum value when $b = \frac{-2}{16} \leq 0$, we have that the value of $8b^2 + 2b - 1$ increases as $b$ increases when $\frac{1}{3} \leq b \leq 1$. Therefore, the minimum value of $8b^2 + 2b - 1$ when $\frac{1}{3} \leq b \leq 1$ is $8\frac{1}{9} + \frac{2}{3} - 1 = \frac{5}{9}$. Combining $8b^2 + 2b - 1 \geq \frac{5}{9}$ and $18 \geq 32b^2 - 16b + 2 \geq \frac{2}{9}$, we have

$$1 + \frac{8b^2 + 2b - 1}{32b^2 - 16b + 2} \geq 1.$$

Therefore, given a fixed $b$, $f(a)$ takes the minimum value when $a = 1$. Therefore, we get

$$f(1) = 16b^2 - 8b + 1 - 40b^2 + 14b - 1 + 25b^2 - 8b + 1 = b^2 - 2b + 1 \geq 0, \forall \frac{1}{3} \leq b \leq 1.$$

**Conclusion:** Combining the results in **Case 1** and **Case 2**, we have

$$\langle \boldsymbol{\ell}^{\boldsymbol{x}}, \boldsymbol{x} - \boldsymbol{x}^*\rangle + (r^{dg}(\boldsymbol{x}))^2 \geq 0.$$

Therefore, we get $\forall \boldsymbol{z} \in \mathcal{N}_{\boldsymbol{\mathcal{X}}}(\boldsymbol{x})$,

$$\langle \boldsymbol{\ell}^{\boldsymbol{x}}, \boldsymbol{x} - \boldsymbol{x}^*\rangle \geq -(r^{dg}(\boldsymbol{x}))^2 = -(\max_{\boldsymbol{x}' \in \boldsymbol{\mathcal{X}}} \langle \boldsymbol{\ell}^{\boldsymbol{x}}, \boldsymbol{x} - \boldsymbol{x}'\rangle)^2 \geq -(C_1 r^{tan}(\boldsymbol{x}))^2 = -C_1^2 \min_{\boldsymbol{z}' \in \mathcal{N}_{\boldsymbol{\mathcal{X}}}(\boldsymbol{x})} \|\boldsymbol{\ell}^{\boldsymbol{x}} + \boldsymbol{z}'\|_2^2$$
$$\geq -C_1^2 \|\boldsymbol{\ell}^{\boldsymbol{x}} + \boldsymbol{z}\|_2^2.$$

In addition, from the definition of the normal cone, we have

$$\langle \boldsymbol{z}, \boldsymbol{x} - \boldsymbol{x}^*\rangle \geq 0, \forall \boldsymbol{z} \in \mathcal{N}_{\boldsymbol{\mathcal{X}}}(\boldsymbol{x}).$$

Combining the above results, we obtain

$$\langle \boldsymbol{\ell}^{\boldsymbol{x}} + \boldsymbol{z}, \boldsymbol{x} - \boldsymbol{x}^*\rangle \geq -(r^{dg}(\boldsymbol{x}))^2 \geq -C_1^2 \|\boldsymbol{\ell}^{\boldsymbol{x}} + \boldsymbol{z}\|_2^2, \forall \boldsymbol{z} \in \mathcal{N}_{\boldsymbol{\mathcal{X}}}(\boldsymbol{x}),$$

which means the weak MVI holds in this game with $\rho = -C_1^2$.

# K  Pseudocode of RM$^+$ Variants Mentioned in This Paper

Now, we provide the pseudocode of RM$^+$ variants mentioned in this paper. Specifically, the pseudocode of SExRM$^+$, SPRM$^+$, and SOGRM$^+$ are shown in Algorithms 1, 2, and 3, respectively.

# L  Experiments

**Configurations.** We conduct experiments on (i) randomly generated two-player zero-sum matrix games with sizes $[10, 20, 50]$, (ii) the normal-form representation of two extensive-form games, Kuhn Poker and Goofspiel, (iii) randomly generated three-player zero-sum polymatrix games with sizes $[10, 20, 50]$, (iv) the games presented in Appendix J.1 and J.2, as well as (v) two Leduc Poker variants. The first three game types satisfy monotonicity [Pérolat et al., 2021], while the games discussed in Appendix J.1 and J.2 satisfy MVI and weak MVI, respectively. The normal-form representations of the two extensive-form games are derived from the open-source code provided by

---

**Algorithm 2** SPRM$^+$

---

**Require:** Step size $\eta \in \left(0, \frac{1}{8DL_u}\right)$.

1: Initialize: $\boldsymbol{\theta}_i^{\frac{1}{2}} \leftarrow \mathbf{1}/|A_i|$, $\boldsymbol{\theta}_i^1 \leftarrow \mathbf{1}/|A_i|$, $\forall i \in \mathcal{N}$
2: **for** $t = 1, 2, \ldots$ **do**
3:     **for** $i \in \mathcal{N}$ **do**
4:         $\boldsymbol{\theta}_i^{t+\frac{1}{2}} \in \arg\min_{\boldsymbol{\theta}_i \in \mathbb{R}_{\geq 1}^{|A_i|}} \{\langle -\boldsymbol{F}_i(\boldsymbol{\theta}^{t-\frac{1}{2}}), \boldsymbol{\theta}_i \rangle + \frac{1}{\eta} D_\psi(\boldsymbol{\theta}_i, \boldsymbol{\theta}_i^t)\}$, $\boldsymbol{x}_i^{t+\frac{1}{2}} = \frac{\boldsymbol{\theta}_i^{t+\frac{1}{2}}}{\|\boldsymbol{\theta}_i^{t+\frac{1}{2}}\|_1}$
5:     **end for**
6:     **for** $i \in \mathcal{N}$ **do**
7:         $\boldsymbol{\theta}_i^{t+1} \in \arg\min_{\boldsymbol{\theta}_i \in \mathbb{R}_{\geq 1}^{|A_i|}} \{\langle -\boldsymbol{F}_i(\boldsymbol{\theta}^{t+\frac{1}{2}}), \boldsymbol{\theta}_i \rangle + \frac{1}{\eta} D_\psi(\boldsymbol{\theta}_i, \boldsymbol{\theta}_i^t)\}$, $\boldsymbol{x}_i^{t+1} = \frac{\boldsymbol{\theta}_i^{t+1}}{\|\boldsymbol{\theta}_i^{t+1}\|_1}$
8:     **end for**
9: **end for**

---

---

**Algorithm 3** SOGRM$^+$

---

**Require:** Step size $\eta \in \left(0, \frac{1}{2DL_u}\right)$ such that satisfies $\frac{1}{2} + \frac{2\rho}{\eta} - 2\eta^2 D^2 L_u^2 > 0$.

1: Initialize: $\boldsymbol{\theta}_i^{\frac{1}{2}} \leftarrow \mathbf{1}/|A_i|$, $\boldsymbol{\theta}_i^1 \leftarrow \mathbf{1}/|A_i|$, $\forall i \in \mathcal{N}$
2: **for** $t = 1, 2, \ldots$ **do**
3:     **for** $i \in \mathcal{N}$ **do**
4:         $\boldsymbol{\theta}_i^{t+\frac{1}{2}} \in \arg\min_{\boldsymbol{\theta}_i \in \mathbb{R}_{\geq 1}^{|A_i|}} \{\langle -\boldsymbol{F}_i(\boldsymbol{\theta}^{t-\frac{1}{2}}), \boldsymbol{\theta}_i \rangle + \frac{1}{\eta} D_\psi(\boldsymbol{\theta}_i, \boldsymbol{\theta}_i^t)\}$, $\boldsymbol{x}_i^{t+\frac{1}{2}} = \frac{\boldsymbol{\theta}_i^{t+\frac{1}{2}}}{\|\boldsymbol{\theta}_i^{t+\frac{1}{2}}\|_1}$
5:     **end for**
6:     **for** $i \in \mathcal{N}$ **do**
7:         $\boldsymbol{\theta}_i^{t+1} = \boldsymbol{\theta}_i^{t+\frac{1}{2}} - \eta \boldsymbol{F}_i(\boldsymbol{\theta}^{t-\frac{1}{2}}) + \eta \boldsymbol{F}_i(\boldsymbol{\theta}^{t+\frac{1}{2}})$
8:     **end for**
9: **end for**

---

Cai et al. [2025] (https://openreview.net/forum?id=LWeVVPuIx0&noteId=4vbVJryMNi&referrer=%5BTasks%5D(%2Ftasks)). The payoff matrices for Kuhn Poker and Goofspiel are of sizes [27, 64] and [72, 7808], respectively. The final game type is utilized to explore the performance of smooth RM$^+$ variants in combination with counterfactual regret minimization (CFR) for solving extensive-form games (EFGs). We conduct the evaluation on Leduc Poker variants because PCFR$^+$, the combination of PRM$^+$ with the CFR framework, is found to perform suboptimally in poker games such as Leduc Poker [Farina et al., 2021].

In the randomly generated three-player zero-sum, the payoff matrix for each pair of players is a diagonal matrix, with each diagonal element sampled from a standard normal distribution. For randomly generated two-player zero-sum matrix and three-player zero-sum polymatrix games, each element of the payoff matrix is uniformly sampled from $[-1, 1]$. For each game size, we generate 20 instances and report the average duality gaps with variances. The duality gap, $r^{dg}(\boldsymbol{x})$, is used to evaluate the distance to NE, defined as $r^{dg}(\boldsymbol{x}) = \sum_{i \in \mathcal{N}} \max_{\boldsymbol{x}_i'} \langle \boldsymbol{\ell}_i^{\boldsymbol{x}}, \boldsymbol{x}_i - \boldsymbol{x}_i' \rangle$. As analyzed in Meng et al. [2025], the duality gap involves a lower bound of the tangent residual in smooth games, $r^{dg}(\boldsymbol{x}) \leq C_1 r^{tan}(\boldsymbol{x})$, where $C_1$ is a game-dependent constant. Thus, if the tangent residual converges to $0$, the duality gap also converges to $0$. Due to the difficulty in precisely calculating the tangent residual, we do not use it as the metric. We compare smooth RM$^+$ variants (SExRM$^+$, SPRM$^+$, and SOGRM$^+$) with existing RM$^+$ variants (ExRM$^+$, PRM$^+$, and RM$^+$), as well as traditional last-iterate convergence OMD based algorithms—OGDA, EG, and OG[4]. For initialization, we set $\boldsymbol{\theta}_i$ to $\mathbf{1}_{|\boldsymbol{\mathcal{X}}_i|}/|\boldsymbol{\mathcal{X}}_i|$ and $\mathbf{0}$ for smooth and other RM$^+$ variants, respectively. For OGDA, EG, and OG, the initial strategy is the uniform strategy. For all tested algorithm, we use simultaneous updates since to our knowledge, the theoretical analysis of last-iterate convergence is based on simultaneous updates. All experiments are performed on a machine with an i9-13900K CPU and 128 GB of memory.

---

[4]OGDA and OG are different algorithms. OGDA is an instance of Optimistic OMD [Rakhlin and Sridharan, 2013] where the regularizer is the Squared L2 norm. The meaning of "optimistic" varies in different papers. Using the terminology in Hsieh et al. [2019], OGDA is PEG and OG is OG.

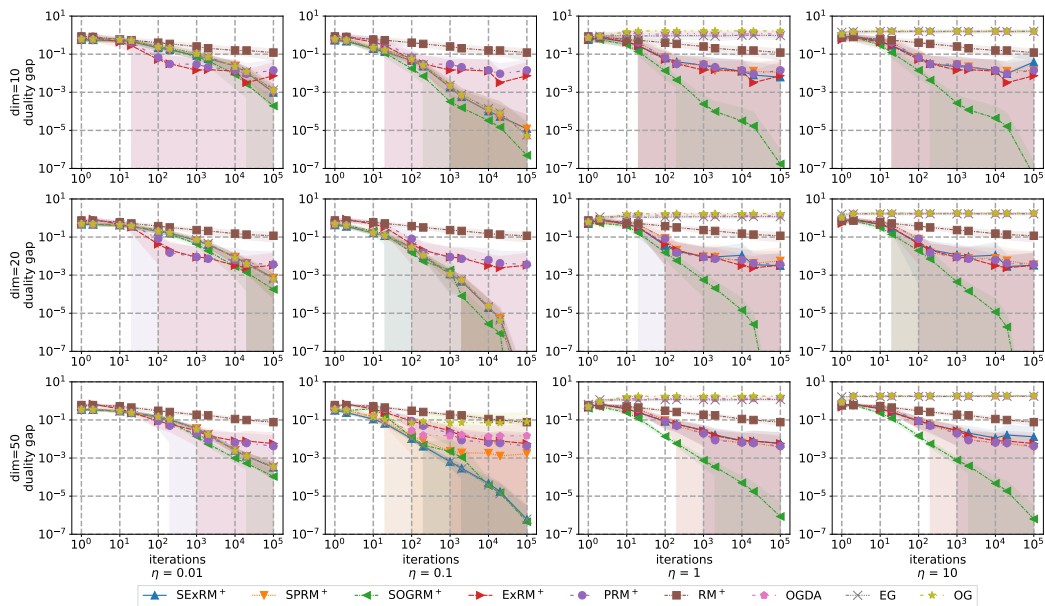

Figure 1: Performance of different algorithms on $10 \times 10$ (top), $20 \times 20$ (middle), $50 \times 50$ (bottom) randomly generated two-player zero-sum matrix games.

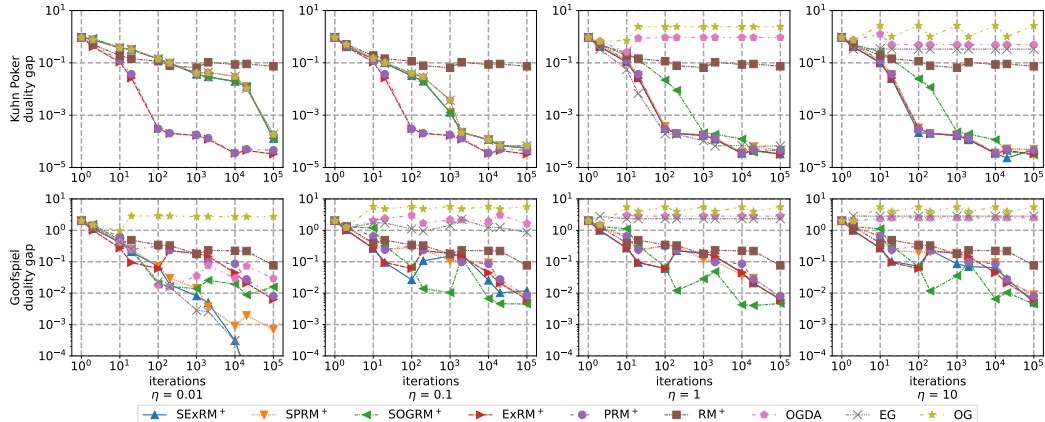

Figure 2: Performance of different algorithms on Kuhn Poker (top) and Goofspiel (bottom).

**Convergence performance on randomly generated two-player zero-sum games.** The results are shown in Figure 1, smooth RM$^+$ variants generally achieve at least similar performance compared to other algorithms. Specifically, OGDA, EG, and OG underperform relative to their smooth RM$^+$ counterparts (SPRM$^+$, SExRM$^+$, and SOGRM$^+$, respectively) and are more sensitive to parameters. For larger $\eta$ values ($\eta = 1$ and $\eta = 10$), OGDA, EG, and OG consistently diverge, while smooth RM$^+$ variants maintain last-iterate convergence. Additionally, we observe that SPRM$^+$ and SExRM$^+$ consistently achieve comparable performance to their corresponding non-smooth RM$^+$ variants, namely PRM$^+$ and ExRM$^+$, respectively. Under optimal parameter settings, SPRM$^+$ and SExRM$^+$ significantly outperform PRM$^+$ and ExRM$^+$, respectively. More importantly, we find that our algorithm, SOGRM$^+$, exhibits the fastest convergence rate and shows the least sensitivity to parameter changes. Moreover, for the reason why SOGRM$^+$ allows large $\eta$ compared to other RM$^+$ variants, we hypothesize that it arises because our proposed algorithm, SOGRM$^+$, performs only a single prox-mapping operator per update step, unlike other smooth RM$^+$ algorithms, which involve two prox-mapping operations at each iteration (the prox-mapping operator is introduced in Section 3). The prox-mapping operator in smooth RM$^+$ variants involves a projection onto the simplex at some time (not always), which may lead to significant changes in $\boldsymbol{\theta}$ depending on the choice of $\eta$. See more details in Appendix M.

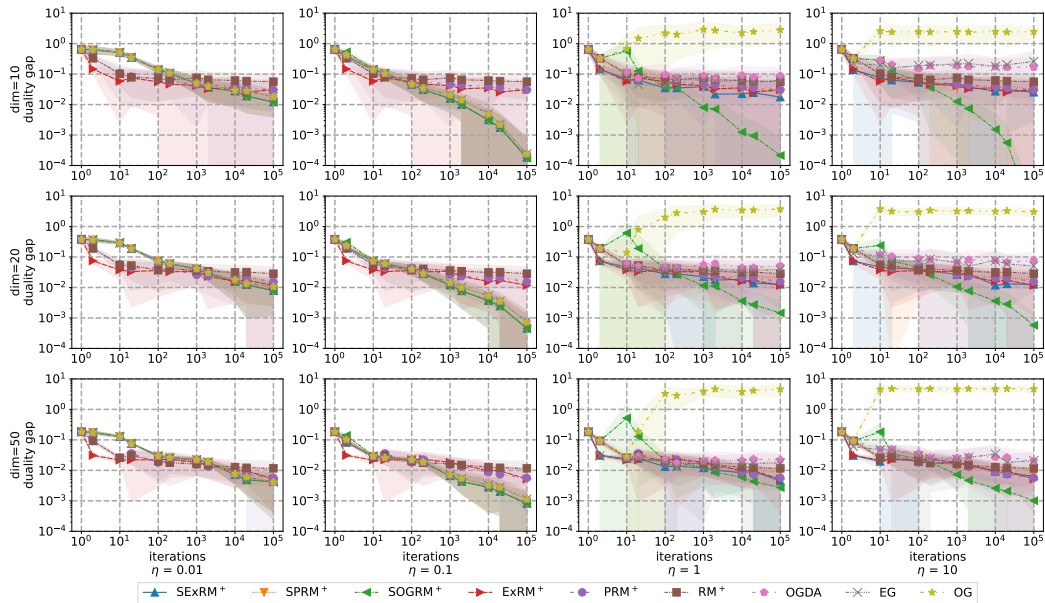

Figure 3: Performance of different algorithms on $10 \times 10$ (top), $20 \times 20$ (middle), $50 \times 50$ (bottom) randomly generated three-player zero-sum polymatrix games.

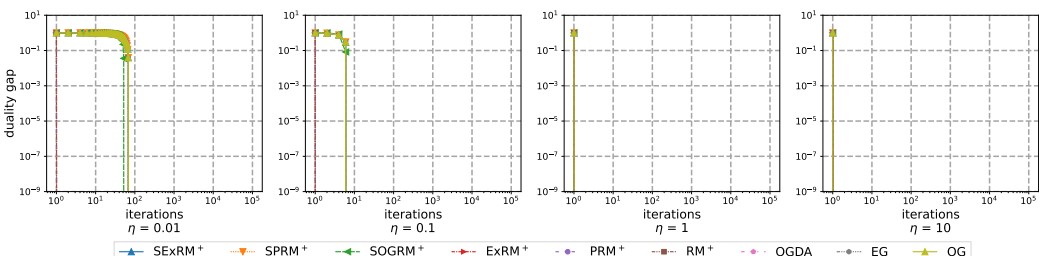

Figure 4: Performance of different algorithms on the game satisfying the MVI provided in Appendix J.1. Notably, this games is very small, *e.g.*, only two actions for each player.

**Convergence performance on the normal-form representation of Kuhn Poker and Goofspiel.** The results are shown in Figure 2. OGDA, EG, and OG exhibit poorer convergence performance and higher sensitivity to hyperparameters compared to their corresponding smooth RM$^+$ variants (SPRM$^+$, SExRM$^+$, and SOGRM$^+$, respectively). Moreover, we observe that OG fails to converge in Goofspiel for any set of parameters. We hypothesize that this is due to the significantly larger scale of Goofspiel compared to the other games tested, requiring OG to use a much smaller learning rate $\eta$ for convergence. In contrast, SOGRM$^+$ demonstrates lower sensitivity to hyperparameters, consistently exhibiting convergence across all parameter settings.

**Convergence performance on randomly generated three-player zero-sum polymatrix games.** The experimental results are shown in Figure 3. Consistent with the results in Figure 1 and Figure 2, the smooth RM$^+$ variants generally exhibit superior convergence performance and reduced sensitivity to hyperparameters compared to their corresponding OMD algorithms. However, we also observe that the OG tends to diverge significantly when $\eta \geq 1$. In contrast, SOGRM$^+$, consistent with previous experimental findings, demonstrates low sensitivity to parameters and retains strong convergence even for $\eta \geq 1$.

**Convergence performance on the games presented in Appendix J.1 and J.2.** The experimental results are shown in Figures 4 and 5. Specifically, Figures 4 and 5 are related to the performance on the games that satisfy the MVI and weak MVI, respectively. Since these games are too small, all algorithms converge quickly. Note that we cannot test on larger games that only satisfy the MVI or weak MVI. This is because we need to prove that the game satisfies the MVI or weak MVI while failing to meet monotonicity. As shown in Appendix J, even for games with only two actions for each player, this proof is quite challenging.

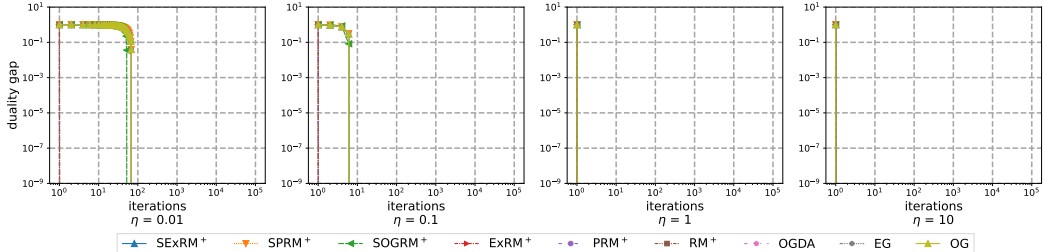

Figure 5: Performance of different algorithms on the game satisfying the weak MVI provided in Appendix J.2. Notably, this games is very small, *e.g.*, only two actions for each player.

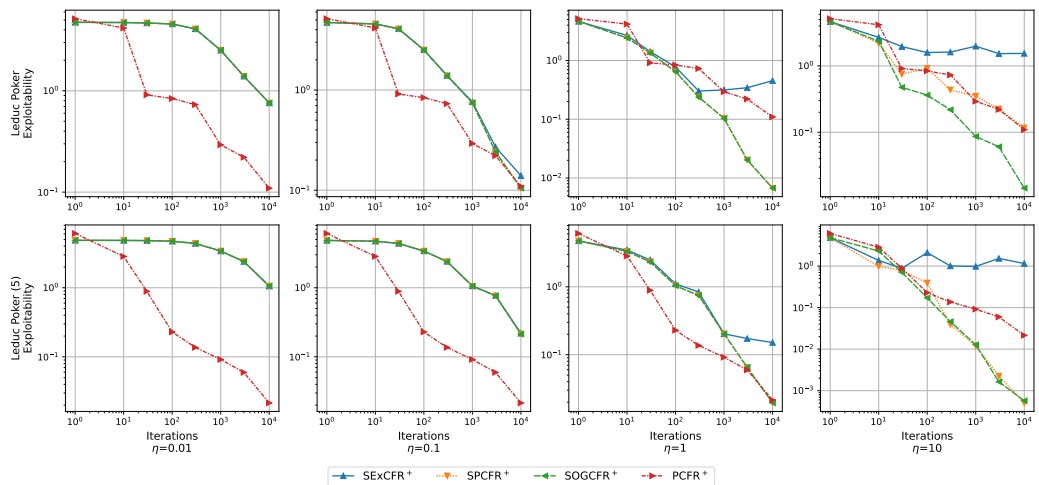

Figure 6: Performance of different algorithms on Leduc Poker and Leduc Poker (5).

**Convergence performance on Leduc Poker variants.** $RM^+$ variants are commonly integrated within the CFR framework to tackle EFGs. Consequently, we evaluate the performance of our algorithm, $SOGRM^+$, when combined with the CFR framework. This integrated method is referred to as $SOGCFR^+$. To validate its efficiency, $SOGCFR^+$ is compared against algorithms that $PCFR^+$, $SPCFR^+$, and $SExCFR^+$, the combinations of CFR framework with $PRM^+$, $SPRM^+$, and $SExRM^+$, respectively. The Leduc Poker variants we tested are Leduc Poker and Leduc Poker (5). In Leduc Poker (5), the number of ranks is 5, whereas in Leduc Poker, it is 3. The experimental results are shown in Figure 6. We observe that $SOGCFR^+$ demonstrates superior robustness to varying step sizes compared to $SPCFR^+$ and $SExCFR^+$.

## M Discussion of the Reason Why $SOGRM^+$ Allows Large $\eta$ Compared to Other $RM^+$ Variants

For the reason why $SOGRM^+$ allows large $\eta$ compared to other $RM^+$ variants, we hypothesize that it arises because our proposed algorithm, $SOGRM^+$, performs only a single prox-mapping operator per update step, unlike other smooth $RM^+$ algorithms, which involve two prox-mapping operations at each iteration (the first occurrence of the prox-mapping operator is in the introduction of OMD, Section 3).

Specifically, the prox-mapping operator in smooth $RM^+$ variants (such as $SExRM^+$, $SPRM^+$, and $SOGRM^+$) involves a projection onto the simplex at some time [Farina et al., 2023] (not always as in OMD algorithms), which may lead to significant changes in $\theta$ depending on the choice of $\eta$. In contrast, the update rule of $SOGRM^+$ (in the second line) omits this prox-mapping operator and instead relies solely on simple addition and subtraction operations. As a result, the initial parameter $\theta_0$ may become negligible compared to the term $\eta F_i(\theta)$. Thus, the values of $\theta_i$ in

Table 3: The sequence of strategies generated by SExRM$^+$.

```
eta=0.01------------------------------------------------------
iteration 1
[0.100000, 0.100000, 0.100000, 0.100000, 0.100000, 0.100000, 0.100000, 0.100000, 0.100000, 0.100000]
iteration 10
[0.098491, 0.100056, 0.109717, 0.106483, 0.096661, 0.091065, 0.094224, 0.101697, 0.107110, 0.094496]
iteration 100
[0.115888, 0.047996, 0.172965, 0.159585, 0.078764, 0.017972, 0.063801, 0.116942, 0.204308, 0.021778]
iteration 1000
[0.247108, 0.000000, 0.114474, 0.070481, 0.000000, 0.000000, 0.082859, 0.185139, 0.299941, 0.000000]
iteration 10000
[0.269960, 0.000000, 0.144571, 0.049592, 0.024670, 0.000000, 0.080797, 0.145601, 0.284809, 0.000000]
iteration 100000
[0.277022, 0.000000, 0.141075, 0.056758, 0.011324, 0.000000, 0.081395, 0.152016, 0.280411, 0.000000]
eta=0.1-------------------------------------------------------
iteration 1
[0.100000, 0.100000, 0.100000, 0.100000, 0.100000, 0.100000, 0.100000, 0.100000, 0.100000, 0.100000]
iteration 10
[0.115382, 0.045286, 0.170913, 0.159142, 0.080181, 0.018597, 0.065633, 0.117780, 0.205713, 0.021372]
iteration 100
[0.244384, 0.000000, 0.121762, 0.065640, 0.000000, 0.000000, 0.088998, 0.180417, 0.298800, 0.000000]
iteration 1000
[0.271157, 0.000000, 0.144291, 0.052641, 0.017717, 0.000000, 0.084207, 0.145175, 0.284811, 0.000000]
iteration 10000
[0.271794, 0.000000, 0.142475, 0.054832, 0.012438, 0.000000, 0.083277, 0.150720, 0.284463, 0.000000]
iteration 100000
[0.271736, 0.000000, 0.142474, 0.054823, 0.012477, 0.000000, 0.083266, 0.150734, 0.284490, 0.000000]
eta=1---------------------------------------------------------
iteration 1
[0.100000, 0.100000, 0.100000, 0.100000, 0.100000, 0.100000, 0.100000, 0.100000, 0.100000, 0.100000]
iteration 10
[0.250238, 0.000000, 0.141786, 0.047503, 0.000000, 0.000000, 0.071202, 0.174680, 0.314590, 0.000000]
iteration 100
[0.272172, 0.000000, 0.107236, 0.031325, 0.012185, 0.000000, 0.098432, 0.173866, 0.304784, 0.000000]
iteration 1000
[0.271736, 0.000000, 0.149928, 0.056319, 0.012477, 0.000000, 0.078212, 0.149291, 0.282037, 0.000000]
iteration 10000
[0.271736, 0.000000, 0.151644, 0.057013, 0.012477, 0.000000, 0.076993, 0.148728, 0.281409, 0.000000]
iteration 100000
[0.271736, 0.000000, 0.142474, 0.054823, 0.012477, 0.000000, 0.083266, 0.150734, 0.284490, 0.000000]
eta=10--------------------------------------------------------
iteration 1
[0.100000, 0.100000, 0.100000, 0.100000, 0.100000, 0.100000, 0.100000, 0.100000, 0.100000, 0.100000]
iteration 10
[0.269329, 0.000000, 0.000000, 0.040992, 0.028389, 0.000000, 0.153818, 0.197411, 0.310061, 0.000000]
iteration 100
[0.267860, 0.000000, 0.142353, 0.054205, 0.019992, 0.000000, 0.066032, 0.178721, 0.270838, 0.000000]
iteration 1000
[0.282710, 0.000000, 0.142474, 0.054823, 0.006415, 0.000000, 0.095782, 0.118767, 0.299029, 0.000000]
iteration 10000
[0.270484, 0.000000, 0.142474, 0.054823, 0.013252, 0.000000, 0.082598, 0.153161, 0.283207, 0.000000]
iteration 100000
[0.280316, 0.000000, 0.142474, 0.054823, 0.006891, 0.000000, 0.095685, 0.118429, 0.301383, 0.000000]
```

SOGRM$^+$ are likely to vary in direct proportion to $\eta$, and the resulting strategy $x_i = \theta_i/\|\theta_i\|_1$ will exhibit a more stable behavior with respect to changes in $\eta$. Therefore, for different values of $\eta$,the sequence of strategies generated by SOGRM$^+$ exhibits small differences. Moreover, when $\eta$ is small, Theorem 6.1 guarantees thatthe sequence of strategies produced by SOGRM$^+$ converges to the set of NE. Consequently, SOGRM$^+$ permits the use of larger $\eta$ values compared to other algorithms.

To validate our statement, as demonstrated in Appendix L, we conducted evaluations on 20 randomly generated 10-dimensional two-player zero-sum matrix games. Specifically, we analyzed the strategies of Player 0 output by SExRM$^+$, SPRM$^+$, and SOGRM$^+$ at iterations 1, 10, 100, 1000, and 10,000. To mitigate randomness, we averaged the strategies across the 20 instances. The results clearly show that for different values of $\eta$,the sequence of strategies generated by SOGRM$^+$ exhibits minimal variation. Notably, when $\eta \geq 1$ and the number of iterations $\geq 1000$, the strategies produced by SOGRM$^+$ are nearly identical across different values of $\eta$. This behavior is not observed in the other two RM$^+$ variants.

Table 4: The sequence of strategies generated by SPRM$^+$.

```
eta=0.01-------------------------------------------------------
iteration 1
[0.100000, 0.100000, 0.100000, 0.100000, 0.100000, 0.100000, 0.100000, 0.100000, 0.100000, 0.100000]
iteration 10
[0.098491, 0.100056, 0.109718, 0.106484, 0.096661, 0.091065, 0.094224, 0.101697, 0.107110, 0.094496]
iteration 100
[0.115888, 0.047996, 0.172965, 0.159585, 0.078764, 0.017972, 0.063801, 0.116942, 0.204308, 0.021778]
iteration 1000
[0.247107, 0.000000, 0.114474, 0.070481, 0.000000, 0.000000, 0.082859, 0.185139, 0.299941, 0.000000]
iteration 10000
[0.269960, 0.000000, 0.144571, 0.049592, 0.024670, 0.000000, 0.080797, 0.145601, 0.284810, 0.000000]
iteration 100000
[0.277022, 0.000000, 0.141075, 0.056758, 0.011324, 0.000000, 0.081394, 0.152016, 0.280411, 0.000000]
eta=0.1-------------------------------------------------------
iteration 1
[0.100000, 0.100000, 0.100000, 0.100000, 0.100000, 0.100000, 0.100000, 0.100000, 0.100000, 0.100000]
iteration 10
[0.115372, 0.045123, 0.170807, 0.159184, 0.080205, 0.018616, 0.065781, 0.117766, 0.205911, 0.021235]
iteration 100
[0.244321, 0.000000, 0.121870, 0.065578, 0.000000, 0.000000, 0.089146, 0.180253, 0.298833, 0.000000]
iteration 1000
[0.271172, 0.000000, 0.144289, 0.052646, 0.017681, 0.000000, 0.084196, 0.145199, 0.284817, 0.000000]
iteration 10000
[0.271796, 0.000000, 0.142475, 0.054831, 0.012437, 0.000000, 0.083275, 0.150721, 0.284465, 0.000000]
iteration 100000
[0.271736, 0.000000, 0.142474, 0.054823, 0.012477, 0.000000, 0.083266, 0.150734, 0.284490, 0.000000]
eta=1-------------------------------------------------------
iteration 1
[0.100000, 0.100000, 0.100000, 0.100000, 0.100000, 0.100000, 0.100000, 0.100000, 0.100000, 0.100000]
iteration 10
[0.237655, 0.000000, 0.000000, 0.000000, 0.000000, 0.000000, 0.236226, 0.203216, 0.322903, 0.000000]
iteration 100
[0.275397, 0.000000, 0.137955, 0.053723, 0.009448, 0.000000, 0.085151, 0.122804, 0.315521, 0.000000]
iteration 1000
[0.261815, 0.000000, 0.142474, 0.054823, 0.019802, 0.000000, 0.068201, 0.186105, 0.266779, 0.000000]
iteration 10000
[0.265631, 0.000000, 0.142474, 0.054823, 0.016390, 0.000000, 0.069668, 0.180080, 0.270934, 0.000000]
iteration 100000
[0.266516, 0.000000, 0.142474, 0.054823, 0.015538, 0.000000, 0.072873, 0.173694, 0.274082, 0.000000]
eta=10-------------------------------------------------------
iteration 1
[0.100000, 0.100000, 0.100000, 0.100000, 0.100000, 0.100000, 0.100000, 0.100000, 0.100000, 0.100000]
iteration 10
[0.227614, 0.000000, 0.000000, 0.000000, 0.035267, 0.000000, 0.172873, 0.288558, 0.275688, 0.000000]
iteration 100
[0.271127, 0.000000, 0.138072, 0.053705, 0.000000, 0.000000, 0.098440, 0.122633, 0.316024, 0.000000]
iteration 1000
[0.272132, 0.000000, 0.142474, 0.054823, 0.013019, 0.000000, 0.083662, 0.145232, 0.288658, 0.000000]
iteration 10000
[0.265599, 0.000000, 0.142474, 0.054823, 0.016434, 0.000000, 0.069693, 0.180184, 0.270793, 0.000000]
iteration 100000
[0.267444, 0.000000, 0.142474, 0.054823, 0.015058, 0.000000, 0.079408, 0.161766, 0.279027, 0.000000]
```

Table 5: The sequence of strategies generated by SOGRM$^+$.

```
eta=0.01---------------------------------------------------
iteration 1
[0.100000, 0.100000, 0.100000, 0.100000, 0.100000, 0.100000, 0.100000, 0.100000, 0.100000, 0.100000]
iteration 10
[0.098649, 0.100078, 0.108744, 0.105828, 0.096983, 0.091963, 0.094790, 0.101526, 0.106359, 0.095079]
iteration 100
[0.115373, 0.048574, 0.172920, 0.159312, 0.079077, 0.018314, 0.063757, 0.117009, 0.203262, 0.022401]
iteration 1000
[0.247102, 0.000000, 0.114400, 0.070508, 0.000000, 0.000000, 0.082918, 0.185222, 0.299849, 0.000000]
iteration 10000
[0.269951, 0.000000, 0.144571, 0.049631, 0.024661, 0.000000, 0.080760, 0.145596, 0.284830, 0.000000]
iteration 100000
[0.277013, 0.000000, 0.141074, 0.056758, 0.011325, 0.000000, 0.081401, 0.152020, 0.280409, 0.000000]
eta=0.1----------------------------------------------------
iteration 1
[0.100000, 0.100000, 0.100000, 0.100000, 0.100000, 0.100000, 0.100000, 0.100000, 0.100000, 0.100000]
iteration 10
[0.109695, 0.054811, 0.168579, 0.154984, 0.082590, 0.023255, 0.067513, 0.117496, 0.194750, 0.026325]
iteration 100
[0.244066, 0.000000, 0.121463, 0.065806, 0.000000, 0.000000, 0.089626, 0.180912, 0.298127, 0.000000]
iteration 1000
[0.271160, 0.000000, 0.144287, 0.052636, 0.017740, 0.000000, 0.084047, 0.145153, 0.284977, 0.000000]
iteration 10000
[0.271809, 0.000000, 0.142475, 0.054831, 0.012428, 0.000000, 0.083271, 0.150721, 0.284465, 0.000000]
iteration 100000
[0.271736, 0.000000, 0.142474, 0.054823, 0.012477, 0.000000, 0.083266, 0.150734, 0.284490, 0.000000]
eta=1------------------------------------------------------
iteration 1
[0.100000, 0.100000, 0.100000, 0.100000, 0.100000, 0.100000, 0.100000, 0.100000, 0.100000, 0.100000]
iteration 10
[0.143554, 0.000000, 0.000000, 0.019365, 0.081302, 0.059547, 0.165077, 0.274010, 0.246277, 0.010868]
iteration 100
[0.272928, 0.000000, 0.140918, 0.055096, 0.011850, 0.000000, 0.079330, 0.152495, 0.287383, 0.000000]
iteration 1000
[0.271736, 0.000000, 0.142474, 0.054823, 0.012477, 0.000000, 0.083267, 0.150733, 0.284489, 0.000000]
iteration 10000
[0.271736, 0.000000, 0.142474, 0.054823, 0.012477, 0.000000, 0.083266, 0.150734, 0.284490, 0.000000]
iteration 100000
[0.271736, 0.000000, 0.142474, 0.054823, 0.012477, 0.000000, 0.083266, 0.150734, 0.284490, 0.000000]
eta=10-----------------------------------------------------
iteration 1
[0.100000, 0.100000, 0.100000, 0.100000, 0.100000, 0.100000, 0.100000, 0.100000, 0.100000, 0.100000]
iteration 10
[0.139452, 0.013191, 0.089558, 0.134699, 0.099862, 0.016938, 0.047250, 0.174526, 0.271009, 0.013515]
iteration 100
[0.267804, 0.000000, 0.142767, 0.053852, 0.014696, 0.000000, 0.083433, 0.150929, 0.286519, 0.000000]
iteration 1000
[0.271736, 0.000000, 0.142474, 0.054823, 0.012477, 0.000000, 0.083266, 0.150734, 0.284490, 0.000000]
iteration 10000
[0.271736, 0.000000, 0.142474, 0.054823, 0.012477, 0.000000, 0.083266, 0.150734, 0.284490, 0.000000]
iteration 100000
[0.271736, 0.000000, 0.142474, 0.054823, 0.012477, 0.000000, 0.083266, 0.150734, 0.284490, 0.000000]
```

