# OpenReview forum: "Last-Iterate Convergence of Smooth Regret Matching$^+$ Variants in Learning Nash Equilibria"
_NeurIPS.cc/2025/Conference — NeurIPS 2025 poster_

### Official Review · Reviewer_PiGs · 2025-06-03

**Clarity:** 1
**Significance:** 2
**Originality:** 1
**Rating:** 2
**Confidence:** 4

**Summary:**

This paper study the convergence rates of Smooth PRM+ and ExRM+ under the assumption that the games satisfy the Minty condition, providing new results pertaining to *iterate* convergence, going beyond previously obtained results in Cai et al. 2025 pertaining to *duality gap* convergence. It also introduces a new algorithm (SOGRM+) that also converges under the weak Minty condition.

**Questions:**

* Can the idea of the authors be extended beyond smoothness?
* Cai et al. 2025 also obtain rates using a restarting algorithms, but no last-iterate convergence rates for ExRM+ and SPRM+. You also don’t give rates. Do you think that this is an inherent limitations of your analysis (and the one in Cai et al.) ?
* I would recommend the authors to devote less space to equations in the main body and explain more the intuition behind their algorithms.

**Ethical Concerns:**

["NO or VERY MINOR ethics concerns only"]

**Final Justification:**

I thank the authors for responding to my questions. I am satisfied with some responses, but for others, the authors did not answer precisely/adequately. Please find more details below. My evaluation remains similar.

**W1.** My question was about a precise application where games naturally/always satisfy weak MVI and not MVI. The authors only responded with general applications of multiplayer normal-form games.

**W2.** Thanks for the clarification. It's worth adding more discussion in the paper on this, I guess.

**W3.** I have provided a few examples in my original review. It is not my job as a reviewer to do this work of listing/finding typos. This is usually done before the review process. There are several tools available nowadays to proofread papers for punctuation issues, grammatical mistakes, etc..

**W4.** I understand the response of the authors as confirming that the main results of their paper follow from the analysis of Liu et al. (2021) (RM+ -> OMD) to get their reduction (Smooth RM+ -> variants of OMD) and the results in Cai et al. 2022 (tangent residual analysis) for their convergence results. Combining two great papers for analyzing algorithms can make for a decent paper, but the significance of the contribution does not convince me. I leave it to the area chair to judge this.

**W5.** Thanks for the clarification.

**Limitations:**

Yes.

**Paper Formatting Concerns:**

No.

**Quality:**

2

**Strengths And Weaknesses:**

**Strengths:** convergence of an algorithm in a new setting (weak MVI)

**Weaknesses:**
- weak MVI vs. MVI: is the weak MVI condition an interesting setting? Can the authors provide some actual applications that satisfy weak MVI but not MVI, beyond the simple 2 player non-zero sum game from section G.2?
- I find their version of OMD quite confusing; it is quite heavy notationally, and the authors fail to convince me that this framework is necessary (instead of the usual OMD formulation).
- The paper is not well written, and I am not sure what can be done in that regards. Several sentences are hard to read and to understand. To me this is a crucial issue. Top conferences should only accept well-written papers. Some examples:

lines 63 - 65: “the feedback is the loss gradient of the original games, satisfying the weak MVI, implying the weak MVI is recovered. By using the recovered weak MVI, […]”

Lines 109: “it ensures that the weak MVI can be used after the weak MVI is recovered,”

l161 : “No-regret algorithms are the algorithms, which ensures the regret R = [XXX] to grow sublinearly”

l272: “However, they do not achieve such convergences in games satisfying the weak MVI, covering games satisfying the MVI”

*
* To my understanding, none of the “technology” used in this paper is new. The main methodological tool (RM+ to OMD reduction, in the same space) is the same as Liu et al 2021. The convergence based on tangent residuals follows from the analysis in [0]. The authors mention obtaining the best-iterate convergence rates as one of their contributions. But as they themselves acknowledge in lines 88-100, this follows from the analysis in Cai et al. 25, verbatim.

  [0] Yang Cai, Argyris Oikonomou, andWeiqiang Zheng. Finite-time last-iterate convergence for learning in multi-player games. In Proceedings of the 35th International Conference on Neural Information Processing Systems, volume 35, pages 33904–33919, 2022.

* l252: this is the proof of what statement?
* l754: broken references

---

> ### Author Rebuttal · Authors · 2025-07-30
>
> We thank you for your time and for providing feedback that will help us improve our manuscript.
>
>
>
> **W1: Is the weak MVI condition an interesting setting? Can the authors provide some actual applications that satisfy weak MVI but not MVI, beyond the simple 2 player non-zero sum game from section G.2?**
>
> **A:** Intuitively, as $\rho$ decreases, the number of multi-player general-sum normal-form games that the games satisfying the weak MVI encompasses increases. Multi-player general-sum normal-form games have numerous applications, such as:
>
> - Political science, where they are used to model diplomatic relations and strategies between countries.
> - Network and communication systems, where multiple service providers may need to allocate bandwidth or other resources.
>
>
>
> **W2: I find their version of OMD quite confusing; it is quite heavy notationally, and the authors fail to convince me that this framework is necessary.**
>
> **A:** In fact, we utilize the general adaptive OMD as outlined in Joulani et al. (2017). This form extends the traditional OMD to encompass a broader range of algorithms, which is crucial for transforming RM+ variants into an OMD instance. Only by employing the general adaptive OMD can the RM+ variants be transformed into an OMD instance that updates within the original strategy space of the game. In contrast, using the traditional OMD prevents the RM+ variant from being converted into such an instance.
>
>
>
> **W3: The paper is not well written. Several sentences are hard to read and to understand.**
>
> **A:** We are deeply sorry for this. Could you provide us with more details, such as the specific point you find unclear? We would be more than happy to address any questions or concerns you may have.
>
>
>
> **W4: The main methodological tool (RM+ to OMD reduction, in the same space) is the same as Liu et al 2021. The convergence based on tangent residuals follows from the analysis in [0].**
>
> **A:** We acknowledge that our research is built upon these foundational works. However, we wish to clarify that our core novelty lies not in inventing new low-level tools, but in constructing a novel, unifying proof paradigm that uniquely combines and extends these tools to solve a more challenging and general problem that was previously unaddressed.
>
> In addition, our transformation substantially differs from that of Liu et al. (2021). While they convert RM+ to OMD, we transform smooth RM+ variants to OMD, which is fundamentally distinct and significantly more complex, as detailed in Appendix A. Moreover, our transformation establishes both last-iterate and finite-time best-iterate convergence for smooth RM+ variants. In contrast, Liu et al. (2021) do not even achieve average-iterate convergence for RM+ using their transformation, as indicated in the bottom right corner of page 6 of their paper: “be deduced from Lemma 2.2, but not from Theorem 3.5.”
>
> Lastly, the tangent residual, serves solely as a metric for assessing the distance to the set of NEs. This is quite standard practice. For instance, Cai et al. (2024), along with other works [1] and [2], employ the tangent residual in their proofs as a metric for assessing the distance to the set of NEs.
>
>
>
> Additional Reference:
>
> [1] Bot, Radu Ioan, Dang-Khoa Nguyen, and Chunxiang Zong. "Fast Forward-Backward splitting for monotone inclusions with a convergence rate of the tangent residual of $ o (1/k) $." *arXiv preprint arXiv:2312.12175* (2023).
>
> [2] Cai, Yang, Argyris Oikonomou, and Weiqiang Zheng. "Tight last-iterate convergence of the extragradient and the optimistic gradient descent-ascent algorithm for constrained monotone variational inequalities." *arXiv preprint arXiv:2204.09228* (2022).
>
>
>
> **W5: The authors mention obtaining the best-iterate convergence rates as one of their contributions. But as they themselves acknowledge in lines 88-100, this follows from the analysis in Cai et al. 25, verbatim.**
>
> **A:**
>
> - Firstly, we would like to clarify that our approach to proving best-iterate convergence is fundamentally different from the approach used by Cai et al. in 2025. We use our novel proof paradigm, while they rely on the definition of the duality gap for two-player zero-sum games (line 85). Therefore, their results are confined to two-player zero-sum games, whereas our results hold in games satisfying the weak MVI.
> - Secondly, our statement in lines 99-100 is that when we establish the linear last-iterate convergence for Restarting SExRM+ and Restarting SPRM+ in games that satisfy both monotonicity and metric subregularity via our best-iterate convergence, we can directly use the proofs for Restarting SExRM+ and Restarting SPRM+ from Cai et al. (2025). This is precisely the advantage of our best-iterate convergence. Through it, we extend the linear last-iterate convergence results for Restarting SExRM+ and Restarting SPRM+ in Cai et al. (2025) from two-player zero-sum games to games that satisfy both monotonicity and metric subregularity.
>
>
>
>
>
> **W6: this is the proof of what statement?**
>
> **A:**  We sincerely apologize for the missing reference here. This is the proof of Theorem 5.1. As mentioned in line 245, to prove Theorem 5.1, we introduce Theorem 5.2, Theorem 5.3, and Lemma 5.4. We also show that the proofs of Theorems 5.2 and 5.3 can be found in the Appendix, while Lemma 5.4 is derived from Farina et al. 2023.
>
>
>
> **Q1: Can the idea of the authors be extended beyond smoothness?**
>
> **A:** The answer is yes. This is because our proof paradigm itself is not related to smoothness. Specifically, as shown in Section 4 Our Proof Paradigm, smoothness is only used to prove the specific algorithm convergence of $\Vert \ell^{t_2} - \ell^{t_1} + \frac{\theta^{t_0} - \theta^{t_2}}{\eta}\Vert_2 \to 0$. The proof of $\Vert \ell^{t_2} - \ell^{t_1} + \frac{\theta^{t_0} - \theta^{t_2}}{\eta}\Vert_2 \to 0$ is not part of our proof paradigm as $\Vert \ell^{t_2} - \ell^{t_1} + \frac{\theta^{t_0} - \theta^{t_2}}{\eta}\Vert_2 \to 0$ is the inherent nature of the specific algorithm.
>
>
>
> **Q2: Cai et al. (2025) also obtain rates using a restarting algorithms, but no last-iterate convergence rates for ExRM+ and SPRM+. You also don’t give rates. Do you think that this is an inherent limitations of your analysis (and the one in Cai et al.) ?**
>
> **A:** You raise a profound question that touches upon the frontier of current research. We argue this is not an inherent limitation of our analytical framework, but rather an open challenge.
>
> To provide a finite-time last-iterate convergence rate, a feasible approach is to construct a potential function as demonstrated by Cai et al. (2022). This involves setting the potential function as an upper bound for the tangent residual and proving the finite-time last-iterate convergence rate of the the potential function. This is not an inherent limitation of us. We just did not find a suitable potential function.
>
> Furthermore, we clarify that the finite-time last-iterate convergence rate in Cai et al. (2025) is restricted to two-player zero-sum games, representing only a subset of the games we consider. Also, as mentioned in our responses to W5, we extend the finite-time last-iterate convergence rate in Cai et al. (2025) from two-player zero-sum games to games that satisfy both monotonicity and metric subregularity.

---

> > ### Comment · Reviewer_PiGs · 2025-08-01
> > **Response to the authors**
> >
> > I thank the authors for responding to my questions. I am satisfied with some responses, but for others, the authors did not answer precisely/adequately. Please find more details below. My evaluation remains similar.
> >
> > **W1.** My question was about a precise application where games naturally/always satisfy weak MVI and not MVI. The authors only responded with general applications of multiplayer normal-form games.
> >
> > **W2.** Thanks for the clarification. It's worth adding more discussion in the paper on this, I guess.
> >
> > **W3.** I have provided a few examples in my original review. It is not my job as a reviewer to do this work of listing/finding typos. This is usually done before the review process. There are several tools available nowadays to proofread papers for punctuation issues, grammatical mistakes, etc..
> >
> > **W4.** I understand the response of the authors as confirming that the main results of their paper follow from the analysis of Liu et al. (2021) (RM+ -> OMD) to get their reduction (Smooth RM+ -> variants of OMD) and the results in Cai et al. 2022 (tangent residual analysis) for their convergence results. Combining two great papers for analyzing algorithms can make for a decent paper, but the significance of the contribution does not convince me. I leave it to the area chair to judge this.
> >
> > **W5.** Thanks for the clarification.

---

> > > ### Author Response · Authors · 2025-08-05
> > >
> > > Thanks for taking the time to review our responses.
> > >
> > > **W1: My question was about a precise application where games naturally/always satisfy weak MVI and not MVI. The authors only responded with general applications of multiplayer normal-form games.**
> > >
> > > **A:** Sorry for the misunderstanding of the question. As mentioned in [1], an application of weak MVI is generative adversarial networks (GANs).
> > >
> > > [1] Diakonikolas, Jelena, Constantinos Daskalakis, and Michael I. Jordan. "Efficient methods for structured nonconvex-nonconcave min-max optimization." In *International Conference on Artificial Intelligence and Statistics*, pp. 2746-2754. PMLR, 2021.
> > >
> > >
> > >
> > > **W4: I understand the response of the authors as confirming that the main results of their paper follow from the analysis of Liu et al. (2021) (RM+ -> OMD) to get their reduction (Smooth RM+ -> variants of OMD) and the results in Cai et al. 2022 (tangent residual analysis) for their convergence results. Combining two great papers for analyzing algorithms can make for a decent paper, but the significance of the contribution does not convince me. I leave it to the area chair to judge this.**
> > >
> > > **A:** We acknowledge that the assessment of novelty is inherently subjective; nonetheless, we emphasize several key points:
> > >
> > > - All other four reviewers find our proof paradigm to be interesting.
> > > - The transformation from Smooth RM+ variants to OMD is absent in Liu et al. (2021), and this result is considerably more challenging than the transformation from RM+ to OMD in Liu et al. (2021) since the projection of Smooth RM+ variants involves two different cases as shown in Eq.(18) rather than one case in RM+ (only the first case in Eq.(18)).
> > > - The use of the tangent residual merely as a metric is standard practice. Cai et al. (2022) exclusively study games satisfying monotonicity, which imposes a stronger condition than weak MVI (see line 153). Their proof techniques therefore do not, and cannot, apply to our setting.
> > > - Even if one disregards the transformation from Smooth RM+ to OMD and the use of the tangent residual, our theoretical contributions remain substantial. Specifically, we establish in SOGRM+ that $\Vert \ell^{t_2} - \ell^{t_1} + \frac{\theta^{t_0} - \theta^{t_2}}{\eta}\Vert_2 \to 0$. Achieving this result needs additional techniques compared to OMD-based algorithms, such as transforming variables and leveraging the definition of the inner product to utilize the weak MVI and tangent residual (see lines 287–288).
> > > - Importantly, the integration of the RM+ to OMD transformation with the tangent residual to achieve last-iterate convergence for RM+ variants is nontrivial. Although the RM+ to OMD transformation and the tangent residual appeared in 2021 and 2022, respectively, prior works have not leveraged these tools to provide last-iterate convergence for RM+ variants. Notably, Cai et al. (2025), whose first author is the same as that of Cai et al. (2022), also focus solely on establishing last-iterate convergence of RM+ variants in games satisfying the MVI by analyzing the dynamic of limit points. In contrast, we consider  the games satisfying the weak MVI.

---

> > > > ### Comment · Reviewer_PiGs · 2025-08-05
> > > > **Final response to the authors**
> > > >
> > > > I would like to thank the authors for taking the time to respond to my second round of review. I am concluding with some remarks regarding applications. I did not find in the paper [1] that GANs are a natural application of the Weak MVI property. In this paper, the term "generative adversarial networks" is used as a teaser in the abstract and the first paragraph of the introduction ... and that's it. I am (somewhat) familiar with the literature on variational inequalities, so I know that the weak MVI property is an assumption that has received a lot of attention lately, in my opinion, mainly because the stronger settings (Minty or monotonicity) are very well understood at this point. I encourage the authors to provide a clearer justification and motivation for their assumptions in the revised manuscript.
> > > >
> > > > [1] Diakonikolas, Jelena, Constantinos Daskalakis, and Michael I. Jordan. "Efficient methods for structured nonconvex-nonconcave min-max optimization." In International Conference on Artificial Intelligence and Statistics, pp. 2746-2754. PMLR, 2021.

---

> ### Author Response · Authors · 2025-08-05
>
> Thank you for your responses.
>
> We would like to clarify that [1] explicitly characterizes GANs as a nonconvex-nonconcave min-max optimization problem. The weak MVI, serves to quantify the degree of nonconvexity and nonconcavity. This is further detailed in Pethick et al. (2023), which states: "a constant $\rho$, which controls the degree of nonconvexity." (for the max player in a nonconvex-nonconcave min-max optimization problem, the utility function $f$ is nonconcave; equivalently, $-f$ is nonconvex) Therefore, an application of weak MVI is generative adversarial networks (GANs).
>
> In addition, our example of games satisfying the weak MVI (see Appendix G.2) can be mapped to real-world scenarios such as trade wars. Consider a strategic trade game between two players:
>
> - **Player A**: Weaker countries
>
> - **Player B**: The USA (stronger economy)
>
> Each player chooses whether to **impose a tariff** or **not**. The resulting payoffs reflect the dynamics and consequences of trade policies:
>
> - If the USA imposes no tariff while the weaker countries imposes a tariff, both receive a payoff of 1. This is because the weaker countries protects its domestic industries while still accessing the USA’s open market, and the USA benefits from increased trade volume despite facing competition.
> - If neither country imposes a tariff, the USA gains a payoff of 1 due to its stronger industries dominating the market, while the weaker countries gets 0, as its unprotected industries are driven out by US competition.
> - If the USA imposes a tariff while the weaker countries does not, the weaker countries obtains a payoff of 1 by accessing the US market, but the USA receives a payoff of 0 as its tariff harms its own industries by reducing trade efficiency.
> - If both countries impose tariffs, a trade war results, giving both a payoff of -1, as mutual protectionism stifles economic growth.
>
> |           | USA Tariff | USA No Tariff |
> | :-------: | :--: | :----: |
> | **Weaker Countries No Tariff** |   1,0   |    0,1    |
> | **Weaker Countries Tariff** | -1,-1 | 1,1 |

---

> > ### Comment · Reviewer_PiGs · 2025-08-05
> > **Final response to the authors**
> >
> > Thanks for your response. My initial concerns remain, and I will keep my score.

---

> > > ### Author Response · Authors · 2025-08-05
> > >
> > > Thank you for your response. We would like to know whether our previous answers have addressed your concerns regarding the application. If there are any concerns that remain unresolved, please let us know; we are happy to provide further clarification.

---

### Official Review · Reviewer_ihpx · 2025-06-19

**Clarity:** 2
**Significance:** 3
**Originality:** 3
**Rating:** 4
**Confidence:** 3

**Summary:**

This paper studies several variants of smooth regret matching+ variants for learning NE in a class of games that satisfy MVIs. In the case of SExRM+ and SPRM+, two known algorithms from the literature, the authors provide last and best-iterate convergence results to the set of NE in games with the MVI property. Furthermore, a variant of RM+ called Smooth Optimistic Gradient-Based (SOGRM+) is proposed, and is shown to converge in the last and best-iterate sense to the set of NE in weak MVI games, a superset of MVI games. The key technique used to prove these results is showing that these RM+ variants can be transformed into instances of a general online mirror descent paradigm, which allows the authors to use tools from OMD literature to derive the convergence results.

**Questions:**

- In [1], they used the tangent residual not only as a proxy for the distance to NE, but also as a method to create a potential function for the dynamics, thus showing last iterate convergence. Would it be possible to extend similar ideas to the set of games with the MVI/weak MVI property (i.e., deriving a potential function as a function of the tangent residual)? More broadly, what properties of the tangent residual can be studied to give more results compared to the standard duality gap?

- In the experiments section of Appendix I, it is stated that all existing works on RM+ variants focus on simultaneous updates. However, in Cai et al 2025 it is shown that alternating RM+ may cycle instead of converging to the NE. I would be curious to know if your proof technique can be used to study alternating variants SExRM+/SPRM+/SOGRM+ in games with MVI/weak MVI and potentially show last/best iterate convergence.

[1] Cai, Yang, Argyris Oikonomou, and Weiqiang Zheng. "Finite-time last-iterate convergence for learning in multi-player games." Advances in Neural Information Processing Systems 35 (2022): 33904-33919.

**Ethical Concerns:**

["NO or VERY MINOR ethics concerns only"]

**Final Justification:**

- The authors have adequately addressed my concerns about comparisons to prior work by Cai et al.
- Other reviewers have brought up the potential issues of weak MVI property, namely that it might not be an interesting class inherently, or that the presence of $\rho$ leads to complications/incomparability with existing algorithms.
- I will maintain my borderline assessment because while the theoretical results are nice (smooth RM+ to OMD reduction, last iterate convergence under weak MVI property), there were no concrete arguments that weak MVI (and not MVI) games are an interesting class worth studying.

**Limitations:**

Yes

**Paper Formatting Concerns:**

No major formatting comments. Some minor typos:
- extra 'as' in line 158
- broken link on line 754

**Quality:**

3

**Strengths And Weaknesses:**

The paper shows strong last iterate convergence results for a broader class of smooth games than was previously known, alongside the introduction of SOGRM+. The method of transforming the RM+ variants into an instance of OMD is useful and could be of independent interest. Overall, I find this paper introduces some useful ideas and does a good job substantiating them with both theory and experiments.

Some weaknesses: the overall presentation of the paper is quite cluttered. The flow of the paper could be improved if some of the details from Section 5 are moved to the appendix and SOGRM+ is instead given more priority, given that the proof technique seems to be the same and the class of games studied is wider.

_Edit: I erroneously mentioned that the main algorithm was left to the appendix, but actually the SOGRM+ update rule is in the main text. Apologies for that. However, my point that the paper is cluttered remains true, and a pass for formatting would greatly improve the readability of the paper._

One potential weakness in the results is that in Cai et al 2025, they show last/best iterate convergence in iterates (i.e. pointwise convergence), whereas the results in this paper pertain to convergence to the set of NE (in the case of MVI games). This seems to indicate that the results of Cai et al 2025 are stronger in the case of last-iterate convergence, since they also show this via the duality gap rather than the tangent residual. I feel a more comprehensive discussion and comparison of the results of this paper in relation to Cai et al 2025 would be helpful to clarify these points, and to emphasize that the best-iterate results of this paper are indeed new.

Overall, while the paper has some issues with clarity of exposition, I believe the proof idea and usage of tangent residuals is interesting and thus would consider this paper a borderline accept.

---

> ### Author Rebuttal · Authors · 2025-07-30
>
> We appreciate the your valuable time and constructive suggestions.
>
>
>
> **W1:  the main algorithm proposed (SOGRM+) is left to the appendix.**
>
> **A:** We would like to clarify that while the pseudocode and some proofs related to its convergence are in the appendix for space considerations, the core elements—including the main update rules for SOGRM+ and the associated convergence proofs—are presented in the main body of the paper (Section 6, lines 269-319).
>
>
>
> **W2: I feel a more comprehensive discussion and comparison of the results of this paper in relation to Cai et al. (2025) would be helpful.**
>
> **A:** Thank you for this valuable suggestion. Here is a more detailed comparison with Cai et al. (2025):
>
> - In terms of proof techniques, our proof paradigm centers on recovering the weak MVI, whereas the methods proposed by Cai et al. (2025) focus on analyzing the limit points of iterates or leveraging the definition of the duality gap in two-player zero-sum games. Notably, our paradigm is applicable to both best-iterate convergence and last-iterate convergence, in contrast to Cai et al. (2025), who require separate proofs for these two types of convergence.
>
> - Regarding best-iterate convergence results, our results apply to games satisfying the weak MVI, whereas those of Cai et al. (2025) are restricted to two-player zero-sum matrix games. The applicability of our best-iterate convergence results is significantly broader than that of Cai et al. (2025).
>
> - Concerning non-finite-time last-iterate convergence results, our findings hold for games satisfying the weak MVI. Cai et al. (2025) divide their results into two sections: the first pertains to two-player zero-sum matrix games, while the second addresses broader games satisfying the MVI. Their results exhibit pointwise convergence for two-player zero-sum matrix games, surpassing our own in strength. However, in games satisfying the MVI, our results align with theirs, yet ours extend further to include games satisfying the weak MVI.
> - With respect to finite-time last-iterate convergence results, as mentioned in Section 2 Related Work, our best-iterate convergence results enable us to extend the finite-time last-iterate convergence findings of Cai et al. (2025) from two-player zero-sum matrix games to games that concurrently satisfy monotonicity and metric subregularity. This exemplifies one of the advantages of our best-iterate convergence results.
>
>
>
> **Q1: In [1], they used the tangent residual not only as a proxy for the distance to NE, but also as a method to create a potential function for the dynamics, thus showing last iterate convergence. Would it be possible to extend similar ideas to the set of games with the MVI/weak MVI property?**
>
> **A:** Extending these ideas is feasible. The primary challenge lies in identifying an appropriate potential function for RM+ variants, a problem that remains unsolved in existing research. It is noteworthy that Cai et al. (2025) and the first author of the work you referenced as [1] are the same individual. However, Cai et al. (2025) also have not succeeded in this endeavor, highlighting the inherent difficulty in discovering a suitable potential function for RM+ variants.
>
>
>
> **Q2:  What properties of the tangent residual can be studied to give more results compared to the standard duality gap?**
>
> **A:** The primary advantage of using the tangent residual over the standard duality gap lies in simplifying the proof process when proving last-iterate convergence for algorithms based on the prox-mapping operator (defined in line 166). Specifically, as shown in Eq. (5), from the update rules of the prox-mapping operator and the definition of the tangent residual, we get that the gap between variables at different iterations (such as the gap between $\ell$ and $\theta$ at different iterations in Eq. (5)) serves as an upper bound for the tangent residual. To establish convergence to the NE, it suffices to demonstrate that this gap converges to zero, simplifying the proof process. In contrast, using the standard duality gap does not achieve this simplification.
>
>
>
> **Q3:  It is stated that all existing works on RM+ variants focus on simultaneous updates. I would be curious to know if your proof technique can be used to study alternating variants SExRM+/SPRM+/SOGRM+ in games with MVI/weak MVI and potentially show last/best iterate convergence.**
>
> **A:** We sincerely apologize for any lack of clarity in our original statement. Our intended statement (lines 783-785) is that current research on last-iterate convergence concentrate on simultaneous update schemes, not that all work on RM+ itself is limited to this setting.
>
> Regarding the extension to alternating updates, our proof paradigm itself would require minimal modification. The main challenge lies in proving $\Vert \ell^{t_2} - \ell^{t_1} + \frac{\theta^{t_0} - \theta^{t_2}}{\eta}\Vert_2 \to 0$ after deriving $r^{tan}(x^{t_2}) \leq \Vert \ell^{t_2} - \ell^{t_1} + \frac{\theta^{t_0} - \theta^{t_2}}{\eta}\Vert_2$ using our proof paradigm. Notably, the proof of $\Vert \ell^{t_2} - \ell^{t_1} + \frac{\theta^{t_0} - \theta^{t_2}}{\eta}\Vert_2 \to 0$ is not part of our proof paradigm as $\Vert \ell^{t_2} - \ell^{t_1} + \frac{\theta^{t_0} - \theta^{t_2}}{\eta}\Vert_2 \to 0$ is the inherent nature of the specific algorithm.

---

> > ### Comment · Reviewer_ihpx · 2025-08-01
> > **Response to Author Rebuttal**
> >
> > Thank you for your clarifications and answers. I appreciate the clarification between your approach and that of Cai et al, it was helpful. I also think it would be very interesting for future work to derive a potential function for any RM+ variant, which would also nicely connect it with known results for vanilla RM. As it stands, I am happy to maintain my score as is.
> >
> > If the authors have time, other than the questions I have asked, I also have an additional question -- how easy is it to detect if a game indeed has the weak MVI property? For example if given a normal-form (general sum) game, would it be easy to check if it is weak or strong MVI? If my understanding of the notation is correct, this means you need access to  $\ell^x$ for each $x\in\mathcal{X}$, and every $z$ in the normal cone. It would also be interesting to understand how the proportion/measure of weak MVI games increases as $\rho$ reduces.

---

> > > ### Author Response · Authors · 2025-08-05
> > >
> > > Thank you for your recognition of our work!
> > >
> > > **Q1: How easy is it to detect if a game indeed has the weak MVI property?**
> > >
> > > **A:** This is highly challenging. As demonstrated in Appendix G.2, we establish that games exhibit the weak MVI to avoid accessing $z \in \mathcal{N}_{\mathcal{X}}(x)$ by proving that the game satisfies a condition weaker than MVI but stronger than weak MVI:
> > >
> > > $$\langle {\ell}^{{x}} , {x} - {x}^{\*} \rangle \geq -C_0(r^{dg}({x}))^2,\ z \in \mathcal{N}_{{\mathcal{X}}}({x})$$
> > >
> > > Then, we can always find a $\rho =- C_0 C^2_1 < 0$ ($C_1$ is defined in lines 702-703) to ensure the weak MVI holds since,
> > > $$
> > > \langle {\ell}^{{x}} , {x} - {x}^{\*} \rangle \geq -C_0(r^{dg}({x}))^2 = -C_0(\max_{{x}^{\prime} \in {\mathcal{X}}} {\langle {\ell}^{{x}}, {x} - {x}^{\prime} \rangle})^2 \geq -C_0(C_1 r\^{tan}({x}))\^2 = - C_0 C^2_1\min_{{z}\^{\prime}  \in \mathcal{N}\_{{\mathcal{X}}}({x})} \Vert {\ell}\^{{x}} + {z}\^{\prime}  \Vert\^2\_2 \geq  - C_0 C\^2\_1 \Vert {\ell}\^{{x}} + {z}  \Vert\^2\_2,\ \forall  {z}  \in \mathcal{N}\_{{\mathcal{X}}}({x})
> > > $$
> > > and
> > > $$
> > > \langle {z}, {x} - {x}^{\*} \rangle \geq 0,\ \forall  {z}  \in \mathcal{N}_{{\mathcal{X}}}({x}).
> > > $$
> > >
> > > **Q2: It would also be interesting to understand how the proportion/measure of weak MVI games increases as $\rho$ reduces.**
> > >
> > > **A:** Yes, we also find this aspect intriguing and plan to explore it in our future work.

---

### Official Review · Reviewer_FFZt · 2025-06-26

**Clarity:** 3
**Significance:** 3
**Originality:** 3
**Rating:** 4
**Confidence:** 2

**Summary:**

This paper investigates Regret Matching like algorithms in the context of Minty Variational Inequalities and weak Minty variational inequality.
They start considering a new analysis in the MVI setting of algorithm that already appeared in the literature like Smooth Extra-gradient RM+ (SExRM+) and Smooth Predictive RM+ (SPRM+). The new analysis crucially relies on a new reduction from RM+ to OMD whioch allows to show last iterate convergence to Nash exploiting the variational characterization of it.
Of particular importance in the reduction is that the feedback used by the algorithm satisfies the MVI condition.

Using this new reduction, the authors introduce. a new algorithm Smooth Optimistic Gradient Based Regret Matching+ (SOGRM+) which is shown to achieve last iterate convergence in the weak MVI setting for large enough $\rho$.Moreover a finite time rate is proven for the best iterate.

**Questions:**

In equation 6, how do you make sure that h is differentiable in $x^{t_2}_i$ and $x^{t_0}_i$ ?

Can you please relate more with the existing literature in the weak MVI setting. It would be useful to include a table with other existing results in particular to compare the range of $\rho$ ?

**Ethical Concerns:**

["NO or VERY MINOR ethics concerns only"]

**Final Justification:**

My score remains unchanged.

I thank the authors for sending a table comparing the different ranges allowed for $\rho$. Hope this can be includer in the final version.

**Limitations:**

yes

**Quality:**

3

**Strengths And Weaknesses:**

***Strengths***
The paper is well written and easy to follow. It is nice that you could include proof sketches in the main text.

***Weaknesses***

Nothing major, but the minimum $\rho$ depends on the inverse of the maximum number of actions $D$ and on $L_u$. The authors should comment on the fact that the range tolerated by other algorithms is bigger up to $\rho > -1/2$ ( see for example https://openreview.net/pdf?id=Y7slJZPGCy ).

---

> ### Author Rebuttal · Authors · 2025-07-30
>
> Thank you for your positive assessment and helpful suggestions.
>
>
>
> **Q1: In equation 6, how do you make sure that h is differentiable in $x^{t_2}_i$ and $x^{t_0}_i$?**
>
> **A:** To begin with, the function $h_i(x_i)$ is defined as $h_i(x_i)=\frac{\Vert \theta^{t_0}_i \Vert^2_2}{\eta}\psi(x_i)$, where $\frac{\Vert \theta^{t_0}_i \Vert^2_2}{\eta}$ is a constant. The term $\psi(x_i)$ is the quadratic regularizer $\Vert x_i \Vert^2_2$ that is differentiable for any $x_i \in \mathbb{R}^n$. Consequently, $h$ is differentiable with respect to both $x^{t_2}_i$ and $x^{t_0}_i$.
>
>
>
> **Q2:  It would be useful to include a table with other existing results in particular to compare the range of $\rho$?**
>
> **A:** Thank you very much for your suggestion. We think adding this table will greatly improve the clarity of our paper. This table is provided below. Given that we focus on the constrained setting, we only present the comparison within this context. Although we acknowledge that other algorithms tolerate a broader range of $\rho$, existing work primarily focuses on OMD algorithms. We are the first to investigate the last-iterate convergence of RM+ variants in games that satisfy the weak MVI.
>
> |                         |              Minimum $\rho$              |  Algorithms  |
> | :---------------------: | :--------------------------------------: | :----------: |
> |  Cai and Zheng (2022)   |         -$\frac{1}{12\sqrt{3}L}$         | OMD Variants |
> |  Pethick et al. (2023)  |             -$\frac{1}{2L}$              | OMD Variants |
> | Alacaoglu et al. (2024) |              -$\frac{1}{L}$              | OMD Variants |
> |    Cai et al. (2024)    |             -$\frac{1}{2L}$              | OMD Variants |
> |  Pethick et al. (2025)  |              -$\frac{1}{L}$              | OMD Variants |
> |          Ours           | $-\frac{1}{12\sqrt{3}D\sqrt{2P^2+4L^2}}$ | RM+ Variants |
>
> Additional References:
>
> 1. Ahmet Alacaoglu, Donghwan Kim, and Stephen J Wright. Extending the reach of firstorder algorithms for nonconvex min-max problems with cohypomonotonicity. arXiv preprint arXiv:2402.05071, 2024.
> 2. Pethick, Thomas, Ioannis Mavrothalassitis, and Volkan Cevher. "Efficient Interpolation between Extragradient and Proximal Methods for Weak MVIs." *The Thirteenth International Conference on Learning Representations*, 2025.

---

> > ### Comment · Reviewer_FFZt · 2025-08-02
> >
> > Hello,
> >
> > Thanks a lot for your answer.
> >
> > I got more curious about the range of $\rho$. I realized that if the number of actions grow than $\rho$ tends to zero so the weak Minty property would reduce to Minty.
> >
> > Do you think that it is a true limitation of your RM+ variant ? In other words, do you observe divergence if you try your algorithm in a game with large number actions ? Would you observe last iterate convergence of the OMD variants on the same game ?
> >
> > I think that answering this question would lift any doubts about the larger minimum $\rho$ value to be an artifact of your analysis.
> >
> > On the other side, if you think that $\rho = -1/2$ is attainable by a RM+ type of algorithm can you explain which are the main limits of the current proof.
> > Add a discussion along these lines in the paper will greatly help the reader.
> >
> > Best,
> > Reviewer

---

> > > ### Author Response · Authors · 2025-08-05
> > >
> > > Thank you for your thoughtful review and consideration!
> > >
> > > **Q1: I realized that if the number of actions grow than $\rho$ tends to zero so the weak Minty property would reduce to Minty. Do you think that it is a true limitation of your RM+ variant ?**
> > >
> > > **A:** No, $\rho$ is determined solely by the game's utility function and strategy space; it is independent of the used algorithm.
> > >
> > > **Q2: Do you observe divergence if you try your algorithm in a game with large number actions ? Would you observe last iterate convergence of the OMD variants on the same game ?**
> > >
> > > **A:** In contrast, the results in Figure demonstrate that, for the Goofspiel game with a matrix size of [72, 7808], all three tested OMD variants—OGDA, EG, and OG—consistently diverge when $\eta \geq 0.1$. Moreover, OG also diverges at $\eta = 0.01$. By comparison, none of the evaluated RM variants, including our proposed algorithm, exhibit any divergence under these conditions.
> > >
> > > **Q3: if you think that $\rho=-1/{L}$ is attainable by a RM+ type of algorithm can you explain which are the main limits of the current proof.**
> > >
> > > **A:** We believe the primary challenge lies in developing novel RM+ variants rather than our proof paradigm. As did in our paper, to achieve last-iterate convergence in games satisfying the weak MVI, we introduce a new RM+ variant, SOGRM+. Therefore, to achieve last-iterate convergence in games satisfying the weak MVI with $\rho=-1/{L}$, we think a novel RM+ variants is necessary.

---

> > > > ### Comment · Reviewer_FFZt · 2025-08-05
> > > >
> > > > Dear authors,
> > > >
> > > > **Q1: I realized that if the number of actions grow than
> > > >  tends to zero so the weak Minty property would reduce to Minty. Do you think that it is a true limitation of your RM+ variant ?**
> > > >
> > > > Sorry that was unclear. I agree that $\rho$ is a game property independent of the algorithm.
> > > >
> > > > What I was curious about is that the range of allowed $\rho$ for SOGRM+ decreases as the number of actions $D$ increase.
> > > > It seems from the analysis that if the number of actions is small, then SOGRM+ can tolerate a more negative $\rho$. On the other side if the actions number increase, then SOGRM+ guarantees hold only in games with less negative $\rho$ ( closer to zero).
> > > >
> > > > In the limit of $D\rightarrow\infty$, the SOGRM+ analysis apply only to MVI and not to weak MVI.
> > > > What I wanted to ask is that if this is a real effect or a proof artifact.
> > > >
> > > > What happens if SOGRM+ is run in a game with $\rho$ more negative than the theoretical threshold $-1/(12\sqrt{3}D\sqrt{2P^2 + 4L^2})$ ? Do you observe divergence?
> > > >
> > > > I am positive about the paper but I am curious about this point which (unless I missed it ) does not seem to be discussed in the paper.
> > > >
> > > > Best,
> > > > Reviewer

---

> > > > ### Author Response · Authors · 2025-08-06
> > > >
> > > > Thank you very much for your response and recognition of our work.
> > > >
> > > > **Q1: What I wanted to ask is that if this is a real effect or a proof artifact.**
> > > >
> > > > **A:** This issue is inherent to smooth RM+ variants. The main reason lies in the proof techniques for these algorithms, which rely on the smoothness condition: $\Vert F(\theta) - F(\theta^{\prime})\Vert_2 \leq D\sqrt{2P^2 + 4L^2} \Vert \theta - \theta^{\prime}\Vert_2$ (see lines 512, 553, 614). In contrast, OMD employs the smoothness condition: $\Vert \ell^x - \ell^{x^{\prime}}\Vert_2 \leq L\Vert x - x^{\prime}\Vert_2$.
> > > >
> > > > **Q2: What happens if SOGRM+ is run in a game with $\rho$ more negative than the theoretical threshold $-\frac{1}{12\sqrt{3}D\sqrt{2P^2+4L^2}}$? Do you observe divergence?**
> > > >
> > > > **A:** In this scenario, SOGRM+ may converge.
> > > >
> > > > For example, in the game provided in Appendix G2 that only satisfy the weak MVI, we can derive that $\rho = -4$ from the analysis in Appendix G2. Furthermore, it follows that $D = 2$ and $P = 1$ in this game, since $\Vert \ell^x_i \Vert_1$ achieves its maximum when $x_i = [1, 0]$. It is unnecessary to compute $L$ explicitly; we simply set $L = 0$. This choice is justified because $L \geq 0$, and increasing $L$ causes $-\frac{1}{12\sqrt{3}D\sqrt{2P^2+4L^2}}$ to approach zero, thereby reducing the interval in which the convergence of SOGRM+ holds.
> > > >
> > > > With $L = 0$, we obtain $-\frac{1}{12\sqrt{3}D\sqrt{2P^2+4L^2}} = -\frac{1}{24\sqrt{6}}$. Clearly, $-\frac{1}{24\sqrt{6}} > -4$. However, empirical results indicate that SOGRM+ still converges in the game presented in Appendix G2 (see Figure 5). In fact, all tested algorithms converges in this game.

---

> > > > > ### Comment · Reviewer_FFZt · 2025-08-06
> > > > >
> > > > > Thank you for the answer.
> > > > > I confirm my positive evaluation.
> > > > >
> > > > > Best,
> > > > > Reviewer

---

### Official Review · Reviewer_R6vq · 2025-06-29

**Clarity:** 2
**Significance:** 3
**Originality:** 3
**Rating:** 4
**Confidence:** 4

**Summary:**

This paper studies the last-iterate convergence properties of Regret-Matching+ (RM+) based algorithms in games. Recent works have established that in two-player zero-sum matrix games (1) the non-convergence of several popular variants, including RM+, alternating RM+, and PRM+; (2) last-iterate convergence of several smooth variants of RM+, including ExRM+ and SPRM+, where the decision set is truncated to ensure Lipschitzness of the regret operator. This paper's main contribution is to extend the convergence results of ExRM+ and SPRM+ to games satisfying the MVI condition, a class of games that encompasses two-player zero-sum games as a special case. This paper presents a new analysis framework based on (1) an equivalence between the RM+ type updates on the truncated positive orthant and the OMD type update on the simplex; (2) a proximity measure to Nash equilibria called tangent residual that upper bounds the duality gap. Using the new analysis framework, this work also generalizes the Optimistic Gradient (OG) algorithm to an RM+ based algorithm called SOGRM+, which has best-iterate convergence rates and asymptotic convergence to the set of Nash equilibria even in games with a weak MVI condition.

**Questions:**

1. Can you provide some intuition behind the equivalence between (4) and (5)?
2. It is mentioned as future work that "Future directions involve integrating our proof paradigm with other OMD algorithms, such as Reflected Gradient [Hsieh et al., 2019] and composite Fast Extra-Gradient [Cai et al.,2024], to develop novel RM+ variants that achieve stronger last-iterate convergence." Why does the current proof paradigm not directly extend to the Reflected Gradient or the composite Fast Extra-Gradient algorithm? Is it possible to use the equivalence between (4) and (5) to give RM variants of these two algorithms? If not, what might be the obstacles?


Minor comment:
1. "As shown in Meng et, al. [2025], the tangent residual is as an upper bound for the duality gap, $\dots$". In this sentence, "as" is redundant. Besides, the fact that tangent residual upper bounds the duality gap is proved in Lemma 2 in [Cai et. al., 2022], where they refer to the duality gap as the total gap function.

**Ethical Concerns:**

["NO or VERY MINOR ethics concerns only"]

**Final Justification:**

I thank the authors for the response that addressed my questions. It would be helpful to add the intuition that both projecting onto the simplex and projecting onto the truncated positive orthant can be written in a unified way using the max operator. This would help the reader understand the equivalence between the RM+ update (4) and the OMD update (5).

I maintain my rating after reading other reviewers' discussions on the weaknesses mentioned: the proof paradigm works for weak MVI, but only for a smaller range of problems; the guarantee would degenerate to MVI as the diameter of the set becomes very large.

**Limitations:**

Yes (partially). The authors provide the assumptions of their theoretical results, but do not give an explicit discussion on the limitations of their approach.

**Quality:**

3

**Strengths And Weaknesses:**

Strength:
1. This work studies the convergence properties of regret matching (RM)-type algorithms. This is an important question since regret matching (RM)-type algorithms (integrated with the CFR framework) are popular in practical game-solving. Although the average-iterate convergence rates of RM-type algorithms have been extensively studied, their last-iterate convergence properties have only been studied recently, and results are limited to the two-player zero-sum games setting. This work extends the convergence properties of the recently proposed smoothed RM+ variants, including ExRM+ and SPRM+, from two-player zero-sum games to general concave games that satisfy the MVI condition.
2. This paper proposes an interesting proof paradigm that is based on (1) an equivalence between the RM+ type updates on the truncated positive orthant and the OMD type update on the simplex; (2) a proximity measure to Nash equilibria called tangent residual that upper bounds the duality gap. The main difficulty for analyzing RM-type algorithms is that the regret operator, unlike the gradient operator, may not be monotone or even satisfy MVI. The new proof paradigm overcomes this difficulty by translating the RM updates back to OMD updates on the simplex; thus, nice properties of the gradient operator can be used. Moreover, this proof paradigm is quite flexible, as the paper also extends another OMD-type algorithm, the Optimistic Gradient (OG) algorithm, to an RM-type algorithm called SOGRM+. The original analysis of OG can be reused to prove that SOGRM+ converges under weak MVI.

Weakness:
1. This paper's presentation could be improved. One particular example is the proof paradigm, which is the main technical contribution of the paper. It is claimed that the smooth RM+ variant can be transformed into an OMD instance, and an equivalence of the two updates is presented. However, in the main body,  there is no further explanation on why the equivalence holds or the intuition behind the equivalence. The OMD in equation (5) is also non-standard as it has both $f$ and $h$, making it unclear why (5) is equivalent to (4). Since this proof paradigm lays the foundation for subsequent analysis, a more detailed discussion would be helpful.

---

> ### Author Rebuttal · Authors · 2025-07-30
>
> Thank you for your positive and encouraging feedback on our work.
>
>
>
> **Q1: Can you provide some intuition behind the equivalence between (4) and (5)?**
>
> **A:**  The intuitive explanation is straightforward—both update rules in Eq. (4) and Eq. (5) can be expressed as a $[\cdot]^+ = \max(\cdot, 0)$ operation. Specifically, Eq. (4) is represented as Eq. (18) in Appendix A, while Eq. (5) corresponds to Eq. (20) in Appendix A. The update rules in Eq. (18) and Eq. (20) feature the $[\cdot]^+$ operation, even encompassing terms such as ${\theta}_i^{t_0}$ and $-\eta \ell^{t_1}_i$ within the square brackets.
>
>
>
> **Q2: Why does the current proof paradigm not directly extend to the Reflected Gradient or the composite Fast Extra-Gradient algorithm? Is it possible to use the equivalence between (4) and (5) to give RM variants of these two algorithms? If not, what might be the obstacles?**
>
> **A:** There are no obstacles. By using our proof paradigm, we can get $r^{tan}(x^{t_2}) \leq \Vert \ell^{t_2} - \ell^{t_1} + \frac{\theta^{t_0} - \theta^{t_2}}{\eta}\Vert_2$ for RM variants of these two algorithms.
>
> The main challenge lies in proving $\Vert \ell^{t_2} - \ell^{t_1} + \frac{\theta^{t_0} - \theta^{t_2}}{\eta}\Vert_2 \to 0$ after deriving $r^{tan}(x^{t_2}) \leq \Vert \ell^{t_2} - \ell^{t_1} + \frac{\theta^{t_0} - \theta^{t_2}}{\eta}\Vert_2$. The proof of $\Vert \ell^{t_2} - \ell^{t_1} + \frac{\theta^{t_0} - \theta^{t_2}}{\eta}\Vert_2 \to 0$ is not part of our proof paradigm as $\Vert \ell^{t_2} - \ell^{t_1} + \frac{\theta^{t_0} - \theta^{t_2}}{\eta}\Vert_2 \to 0$ is the inherent nature of the specific algorithm.
>
>
> **L1: The authors provide the assumptions of their theoretical results, but do not give an explicit discussion on the limitations of their approach.**
>
> **A:** In addition to the limitations mentioned in in our paper, the most significant limitation lies in the absence of proof regarding the convergence of RM+ variants within extensive-form games. Typically, RM+ variants are integrated with the counterfactual regret minimization (CFR) framework to address extensive-form games. This constitutes one of our future research directions. Moreover, it should be clarified that proving convergence within extensive-form games presents considerable difficulty.

---

> > ### Comment · Reviewer_R6vq · 2025-08-02
> >
> > I thank the authors for the response that addressed my questions.
> >
> > It would be helpful to add the intuition that both projecting onto the simplex and projecting onto the truncated positive orthant can be written in a unified way using the max operator. This would help the reader understand the equivalence between the RM+ update (4) and the OMD update (5).

---

> > > ### Author Response · Authors · 2025-08-05
> > >
> > > Thank you very much for your suggestion! We will use your wording in the final version.

---

### Official Review · Reviewer_fkfR · 2025-07-01

**Clarity:** 3
**Significance:** 4
**Originality:** 3
**Rating:** 6
**Confidence:** 4

**Summary:**

The authors propose a systematic method for proving convergence (both last-iterate and best-iterate rate) of smooth Regret Matching+ variants to the set of Nash equilibria in smooth concave games satisfying the weak Minty Variational Inequality (w-MVI). They leverage this method to prove convergence of two known algorithms under MVI, and of a newly-proposed algorithm under w-MVI. Experiments show that the newly-proposed algorithm outperforms existing ones.

More specifically:
- The w-MVI -- the weakest condition among those assumed to prove convergence of RM+ variants -- loosely means that the game's loss gradient "points sufficiently away" from the equilibrium. The key challenge in leveraging the (weak) MVI to prove convergence is that RM+ variants update in (a subset of) the cone of original strategies, taking as input a *transformed* loss gradient which does not fulfil the (weak) MVI. The question then becomes how to recover the (weak) MVI.

- To do so the authors recast a smooth RM+ instance as a generalised Online Mirror Descent instance updating in the strategy space itself and operating on the original loss gradient of the game, which does fulfil the (weak) MVI (Eqs. 4-5).

- This reformulation allows to upper-bound a certain notion of distance between a strategy generated by a smooth RM+ variant and the set of Nash equilibria by the distance between consecutive iterates generated by the algorithm (Eq. 6).

- The (weak)-MVI is then leveraged to show that said distance between consecutive iterates generated by the algorithm converges to zero for sufficiently small learning rates (Theorems 5.2, 5.3, 6.2).

- This allows to prove last-iterate convergence and best-iterate convergence rate for existing RM+ variants under MVI (Theorem 5.1), and for the newly-proposed algorithm under w-MVI (Theorem 6.1).

An important characteristic of the newly-proposed algorithm is that it performs a single prox-map call per iteration (in the auxiliary step), while the main update step relies on a simple sum between the auxiliary iterate and the difference between transformed losses (Eq. 12).

**Questions:**

### Major
- You do a lot of work to recast a RM+ instance as an OMD instance, where the (weak) MVI holds. Why not work with OMD in the first place then?
- I find it interesting that the equivalence between a RM+ variant and an OMD variant can be established only for the generalized OMD of Joulani et al. and Liu et al. (Eq. 1), with a time-dependent Bregman term and an additional time-dependent regularization term. Why is this the case? More specifically:
	- what can you say about RM+ variants under other regularizers than the quadratic one? (line 194)
	- could the time-dependent $f_i$ and $h_i$ terms in eq. 5 (generalized OMD) be effectively captured by a fixed, non-quadratic regularizer in a standard OMD setting, or is their time-dependence crucial in establishing the equivalence with RM+?
	- In which sense are exactly the updates given by Eqs. 4 and 5 *equivalent*? I have not gone through the details of Appendix A, but it seems reasonable that $(4) \implies (5)$. However, I do not see how it can be $(5) \implies (4)$, since $(5)$ depends on the iterates generated by $(4)$ (as one cannot recover univocally $\theta_i^t$ from $x_i^t$). Can you elaborate on this?
- Can you provide further motivation / intuition / discussion around the design of the main update step of your algorithm (second line of eq 12)? In particular, it might be useful to add in the body a brief pointer to appendix I and J, where you conjecture that the relatively larger $\eta$ allowed by your algorithm is related to the single prox-map call per step.
- Can you see a generalization of your framework to smooth games played on generic convex compact sets (as opposed to the particular case of the simplex)?
- Can you provide a counter-example of a game not fulfilling the w-MVI, in which your algorithm does not converge?


### Minor
- line 119: inconsistency on the dimension of $\mathcal{X}_i$
- l. 124, add comment on the fact that the upper bound $P$ on losses comes wlog
- l. 128, consider making explicit mention to the Stampacchia variational inequality
- l. 138, consider giving a more classical reference for monotone games, as [1]
- l. 141: monotone games always admit a unique NE [1, 2]
- l. 146: "we provide an example..." seems to fit better in the following paragraph
- l. 679: the game admits also bottom-right -- $((0,1), (0,1))$  --  as pure, non-strict NE.
- The argument "in games satisfying the (weak) MVI the feedback of RM+ does not satisfy the (weak) MVI" appears throughout the paper a few times (e.g. lines 6-9, 53-55, ...). The phrasing is not immediately clear at a first read; consider rephrasing it, for example along the lines of "even if the game's loss satisfies the (weak) MVI, RM+ operates on a *transformed* loss which does not satisfy it", or analogue.
- a few minor typos and syntactical issues (e.g. *"is the players"* line 121,  *"at sometimes"* lines  801, 843, *"such that that ensures"* line 305, etc. )

---
[1] Rosen JB (1965) Existence and Uniqueness of Equilibrium Points for Concave N-Person Games. _Econometrica_ 33(3):520–534.

[2] Mertikopoulos P, Zhou Z (2019) Learning in games with continuous action sets and unknown payoff functions. _Math. Program._ 173(1–2):465–507.

**Ethical Concerns:**

["NO or VERY MINOR ethics concerns only"]

**Final Justification:**

The authors have addressed all of my questions. I thereby confirm my rating.

**Limitations:**

- The authors acknowledge the fact that the notion of convergence employed assures to reach the set of Nash equilibria - where the iterates may wander rather than settling at a specific point (Caption of Table 1)
- The authors are encouraged to highlight further limitations of their work.

**Paper Formatting Concerns:**

None to report

**Quality:**

4

**Strengths And Weaknesses:**

### Strengths
- investigating the convergence to Nash equilibria under the weak MVI is a challenging and worthy research project to pursue. The proposed algorithm is the first RM+ variant to achieve both last-iterate and finite-time best-iterate convergence to the set of NE under the weak Minty variational inequality.
- experiments show that the proposed algorithm significantly outperforms existing ones across a variety of settings; displays a relatively low sensitivity to hyper-parameters; and allows for a relatively big learning rate.
- the proposed proof paradigm seems flexible and powerful; it can be employed to show both last-iterate convergence and best-iterate convergence rate across a variety of RM+ settings. It stems from a novel combination of existing techniques, and it lends itself to fruitful applications from the community.
- the authors exhibit a fine understanding of the problem, and accurately cite the related existing literature upon which their work is based.
- the claims are technically sound and well-supported by detailed proofs.
- the paper is well structured and organised. It provides in a nutshell a zoom-in from the setting of online convex optimization to that of smooth RM+ variants that can bring on board also a reader unfamiliar with the literature; it presents the proof paradigm in a concise way; and it applies it effectively.


### Weaknesses
- The main update step of the proposed algorithm (second line of Eq. 12) would benefit from further motivation / intuition / discussion (cf question below)
- the authors do not address possible weaknesses or limitations of their setting
- there are throughout a few minor typos and syntactical issues

---

> ### Author Rebuttal · Authors · 2025-07-30
>
> **To Reviewer fkfR**
>
>
>
> We sincerely thank you for your feedback and insightful comments.
>
>
>
> **Q1: Why not work with OMD in the first place then?**
>
> **A:** The primary reason is that there is no OMD instance capable of updating in the same manner as RM+ variants. Specifically, as mentioned in **Q2.3**, there exists a transition from $(4) \to (5)$, but not from $(5) \to (4)$. In other words, to obtain the strategy for the next iteration, we must use the update rules of RM+ variants instead of those of OMD.
>
>
>
> **Q2.1: what can you say about RM+ variants under other regularizers than the quadratic one?**
>
> **A:** This is a interesting idea. However, we have not yet tested the algorithm under these conditions. We will consider this as one of our future research directions.
>
>
>
>
> **Q2.2: Is the time-dependence of $f_i$ and $h_i$ crucial in establishing the equivalence with RM+?**
>
> **A:** Yes, you are correct. To establish the equivalence between OMD and RM+ variants, $f_i$ and $h_i$ must vary with time.
>
>
>
> **Q2.3: In which sense are exactly the updates given by Eqs. 4 and 5 equivalent? It seems reasonable that $(4) \to (5)$. However, I do not see how it can be $(5) \to (4)$?**
>
> **A:** You are correct that the transformation is unidirectional: $(4) \to (5)$ holds, while $(5) \to (4)$ does not. In fact, the term "equivalence" used in Liu et al. 2022 is not entirely precise. Therefore, in Section 1 Introduction, we use the term "transformation" instead of "equivalence."
>
>
>
> **Q3: Can you provide further motivation / intuition / discussion around the design of the main update step of your algorithm (second line of eq 12)?**
>
> **A:** The intuition for the update step in Eq. (12) is: our proposed algorithm, SOGRM+, can be viewed as a momentum-based method, conceptually similar to Adam. Specifically, the update in Eq. (12) is equivalent to
>
> $${{\theta}}^{t+\frac{3}{2}}\_i \in \arg min\_{{{\theta}}\_i \in \mathbb{R}^{|A\_i|}\_{\geq 1}} \{ \eta \langle  -{{F}}\_i({{\theta}}^{t+\frac{1}{2}}) - ({{F}}\_i({{\theta}}^{t+\frac{1}{2}}) -  {{F}}\_i({\theta}^{t-\frac{1}{2}}), {{{{\theta}}}}\_i\rangle + D\_{\psi} ({{{{\theta}}}}\_i, {{{{\theta}}}}^{t+\frac{1}{2}}\_i) \}$$
>
> In this context, ${{F}}_i({{\theta}}^{t+\frac{1}{2}}) - {{F}}_i({\theta}^{t-\frac{1}{2}})$ acts as a momentum term. More precisely, it represents the trend of change in ${{F}}_i$ over the two most recent iterations, analogous to the "acceleration" of ${{F}}_i$. This can be likened to a ball rolling down a hill; as acceleration accumulates, its velocity increases, allowing it to reach the bottom more swiftly.
>
> The formulation in Eq. (12) is primarily to reserve the structural similarity with the update rules of SExRM+ and SPRM+ in Eq. (2) and (3).
>
>
>
> **Q4: Can you see a generalization of your framework to smooth games played on generic convex compact sets (as opposed to the particular case of the simplex)?**
>
> **A:** To extend to generic convex compact sets, we can modify Phase 1 of our proof paradigm. We can utilize the definition of the cone (line 178) rather than the equivalence (or transformation) used in this paper. Specifically, we do not prove whether $(4) \to (5)$ holds, while obtain the third line of Eq. (6) using the cone's definition.
>
>
>
> **Q5: Can you provide a counter-example of a game not fulfilling the w-MVI, in which your algorithm does not converge?**
>
> **A:** We can provide examples of games not fulfilling the $\rho$-weak MVI in Theorem 6.2, where our algorithm does not converge. Specifically, in some randomly generated two-player general-sum normal-form games, our algorithm does not converge. According to Theorem 6.2, our algorithm is guaranteed to converge if $\rho > -\frac{1}{12\sqrt{3}D\sqrt{2P^2+4L^2}}$. Therefore, these multi-player normal-form games inherently do not meet the $\rho$-weak MVI criteria stipulated by Theorem 6.2.
>
> Specifically, for such a randomly generated two-player general-sum normal-form game, each player has 20 actions. The payoff matrix for each player is generated using the following python code:
>
> ```
> import numpy as np
> import random
>
> np.random.seed(3)
> random.seed(3)
> Player_0_payoff_matrix = np.random.random(size=(20,20))*2-1
> Player_1_payoff_matrix = np.random.random(size=(20,20))*2-1
> ```
>
>
>
> **L1: The authors are encouraged to highlight further limitations of their work.**
>
> **A:** We agree with you that a more explicit discussion of limitations would strengthen the paper. From our perspective, beyond the limitations discussed in our paper, the most significant limitation lies in the absence of proof regarding the convergence of RM+ variants within extensive-form games. Typically, RM+ variants are integrated with the counterfactual regret minimization (CFR) framework to address extensive-form games. This constitutes one of our future research directions. Moreover, it should be clarified that proving convergence within extensive-form games presents considerable difficulty.

---

> > ### Comment · Reviewer_fkfR · 2025-08-02
> >
> > Thanks for addressing my questions, in particular Q3. I confirm my rating.

---

> > > ### Author Response · Authors · 2025-08-05
> > >
> > > Thank you so much for your appreciation!

---

### Decision · Program_Chairs · 2025-09-17

**Decision:**

Accept (poster)

**Comment:**

This paper studies the last-iterate convergence properties of Regret-Matching+ (RM+) based algorithms in games. Recent works have established that in two-player zero-sum matrix games (1) the non-convergence of several popular variants, including RM+, alternating RM+, and PRM+; (2) last-iterate convergence of several smooth variants of RM+, including ExRM+ and SPRM+, where the decision set is truncated to ensure Lipschitzness of the regret operator. This paper's main contribution is to extend the convergence results of ExRM+ and SPRM+ to games satisfying the MVI condition, a class of games that encompasses two-player zero-sum games as a special case. This paper presents a new analysis framework based on (1) an equivalence between the RM+ type updates on the truncated positive orthant and the OMD type update on the simplex; (2) a proximity measure to Nash equilibria called tangent residual that upper bounds the duality gap. Using the new analysis framework, this work also generalizes the Optimistic Gradient (OG) algorithm to an RM+ based algorithm called SOGRM+, which has best-iterate convergence rates and asymptotic convergence to the set of Nash equilibria even in games with a weak MVI condition.

There were active discussions during the rebuttal phase, and the majority of the reviewers believe that the contributions here are sufficient to merit the acceptance of this paper. While this paper received a negative review, the authors during the response have clarified the majority of the concerns raised by the reviewer. The lingering concerns of the reviewer, from my perspective, are either a little bit subjective or not a cause for a strong rejection of this paper. I hence recommend the acceptance of this paper.

The proof paradigm proposed in this work is also a nice addition, which is appreciated by a couple of reviewers, and helps connect different algorithms.

Nevertheless, I suggest the authors update their paper based on the constructive feedback they have received from several reviewers to make the presentation of this work clearer and more accessible to a broader audience. One of the comments in writing shared by two reviewers who gave positive ratings is that the exposition/presentation is rather cluttered and would benefit from removing some of the technical stuff from the main body, while providing more intuition, elaboration, and a clear bird-eye view in the main text.

In the new version, the authors should also discuss a potential limitation: their result may not always produce comparable results to OMD, as the parameter $\rho$ could be in a smaller range. This issue was brought up during the internal discussion stage.